# On the Convergence and Sample Complexity Analysis of Deep Q-Networks with $\varepsilon$-Greedy Exploration

**Shuai Zhang**
New Jersey Institute of Technology

**Hongkang Li**
Rensselaer Polytechnic Institute

**Meng Wang**
Rensselaer Polytechnic Institute

**Miao Liu**
IBM Research

**Pin-Yu Chen**
IBM Research

**Songtao Lu**
IBM Research

**Sijia Liu**
Michigan State University

**Keerthiram Murugesan**
IBM Research

**Subhajit Chaudhury**
IBM Research

## Abstract

This paper provides a theoretical understanding of Deep Q-Network (DQN) with the $\varepsilon$-greedy exploration in deep reinforcement learning. Despite the tremendous empirical achievement of the DQN, its theoretical characterization remains underexplored. First, the exploration strategy is either impractical or ignored in the existing analysis. Second, in contrast to conventional Q-learning algorithms, the DQN employs the target network and experience replay to acquire an unbiased estimation of the mean-square Bellman error (MSBE) utilized in training the Q-network. However, the existing theoretical analysis of DQNs lacks convergence analysis or bypasses the technical challenges by deploying a significantly over-parameterized neural network, which is not computationally efficient. This paper provides the first theoretical convergence and sample complexity analysis of the practical setting of DQNs with $\varepsilon$-greedy policy. We prove an iterative procedure with decaying $\varepsilon$ converges to the optimal Q-value function geometrically. Moreover, a higher level of $\varepsilon$ values enlarges the region of convergence but slows down the convergence, while the opposite holds for a lower level of $\varepsilon$ values. Experiments justify our established theoretical insights on DQNs.

## 1 Introduction

Reinforcement learning (RL) is a sequential decision-making process for a learning agent taking actions in the environment. RL has found important applications in autonomous control [37, 36], healthcare [16], Internet of Things [40, 33], and natural language processing [70]. The environment of an RL problem is modeled as a Markov decision process (MDP) with an underlying state transition probability matrix. The goal of the problem is to find an optimal policy to select the best actions to maximize the immediate and future rewards. Q-learning [77] has been recognized as one of the most promising and efficient learning algorithms for seeking the optimal policy because it does not require the knowledge of the model of the environment (namely, "model free") and can learn from data generated from a non-optimal policy (namely, "off-policy"). Traditional Q-learning approaches are centered on tabular methods [82, 41] or linear function approximations [65, 13] to estimate the optimal action-value (Q-value) function. However, tabular methods require sample complexity scaling in the order of state space, which is impractical for modern RL problems involving large or even infinite state space [82]. Q-learning with linear function approximation is limited to applications only when the transition matrix admits a linear feature representation [23].

37th Conference on Neural Information Processing Systems (NeurIPS 2023).

Table 1: Comparison among some representative works of Q-learning with function approximation.

| Work | Neural Approximation | Convergence of MSBE | $\varepsilon$-greedy |
|---|:---:|:---:|:---:|
| Yang & Wang (2019) | ✗ | ✓ | ✗ |
| Xu & Gu (2020) | ✓ | ✓ | ✗ |
| Fan et al. (2020) | ✓ | ✗ | ✗ |
| Liu et al. (2022) | ✓ | ✗ | ✓ |
| This work | ✓ | ✓ | ✓ |

Due to the remarkable advancements in deep neural networks (DNNs), the Deep Q-network (DQN) framework has emerged as a powerful approach that leverages the expressive power of non-linear functions and the ability to generalize to unknown states. DQN has proven to be a pioneering solution that addresses challenges encountered by traditional approaches. In the DQN framework [51], the Q-value function is approximated using a DNN, and the algorithm iteratively updates the Q-value function by collecting data following the $\varepsilon$-greedy policy. The $\varepsilon$-greedy policy is the simplest approach to balance exploration and exploitation. Namely, with a probability of $\varepsilon$, we select a random action (*exploration*), and with a probability of $1 - \varepsilon$, we choose the best action according to the current estimated Q-value function (*exploitation*). $\varepsilon$-greedy is myopic compared with other strategic explorations, e.g., Thompson sampling-based [71, 55, 25] and optimism in the face of the uncertainty (OFU)-based [30, 31] ones. However, implementing these strategic explorations is not computationally efficient in DQNs [17], while DQNs equipped with $\varepsilon$-greedy policy have shown empirical success in diversified applications, e.g., the game of Go [63], Atari games [51], robotics [34], and autonomous vehicles [61, 12, 60].

Despite the numerical success, the theoretical understanding of DQN remains elusive, which could prevent its applications in domains requiring *reliable and safe* decision-making. First, updating the Q-value function involves minimizing a mean-squared Bellman error (MSBE) function. However, existing convergence analysis and statistical properties are predominantly limited to linear models, failing to capture the complexities present in non-linear neural networks like DQN. Second, the sample complexity required for the convergence of MSBE is still not well comprehended. Achieving the desired accuracy in this context often demands a sample complexity that grows exponentially with the input dimension [46], rendering it impractical and inefficient in real-world scenarios. Third, the optimal selection of the $\varepsilon$ value in DQN remains a gap in existing research. The hyperparameter tuning in algorithms involved with neural networks can be arduous and time-consuming. For instance, without a well-designed hyperparameter configuration, only a small fraction (e.g., 1%) of the possible combinations yield satisfactory results in neural network training [76].

**Contributions.** To the best of our knowledge, this paper presents the first theoretical study with convergence analysis for Deep Q-Networks (DQNs) utilizing the $\varepsilon$-greedy policy. A comparison with existing works can be found in Table 1. The paper focuses on the Q-value function approximated by a DNN. It offers a comprehensive analysis of the convergence of DQNs and provides insights into the estimation error of the learned Q-value function, accompanied by an analysis of the sample complexity. The key contributions of this study are as follows:

**1. The convergence analysis of DQN with bounded estimation errors.** Assuming the existence of a DNN with unknown weights $W^\star$ that matches the optimal Q-value function, this paper proves that the learned model via DQNs equipped with $\varepsilon$-greedy policy through the (accelerated) gradient descent algorithm converges linearly to $W^\star$ up to some characterizable estimation error.

**2. The theoretical characterization for a wide selection range of $\varepsilon$ over iterations.** This paper provides lower and upper bounds of $\varepsilon$ at each iteration for the convergence of DQNs and, in particular, characterizes the sample complexity, estimation error, and convergence rate of the DQN equipped with $\varepsilon$-greedy with decreasing $\varepsilon$. Moreover, this paper proves that a higher level of $\varepsilon$ values leads to an enlarged region of convergence, which relaxes the requirement on the initialization, while a lower level of $\varepsilon$ values leads to faster convergence.

**3. The sample complexity analysis for learning a desired Q-value function.** We quantify the required sample complexity, depending on the neural network parameters and distribution shift of the collected data, for the learned model to converge to the desired accuracy. Typically, the estimation error of the converged model scales in the order of $1/\sqrt{N_s}$, where $N_s$ is the number of samples.

## 2 Related Works.

**Q-learning with linear function approximation.** In the setting of linear function approximation, the Q-function is assumed to be a linear function of either the feature mapping [83, 32, 92] or a mixture of some basis kernels [91, 52]. Early works mainly focus on the algorithm design [6, 48, 2] and convergence analysis [38, 47, 67, 14, 69] but lacks theoretical guarantees with polynomial sample complexity. Assuming the underlying Q-function can be exactly represented as a linear function of the feature mapping with some unknown parameters, several sample-efficient algorithms are proposed to find the ground-truth mapping with finite-sample guarantee [11, 80], and the sample complexity depends linearly on the feature dimension [80].

**Q-learning with non-linear function approximation.** Recent approaches with non-linear function approximation mainly fall into the frameworks of the Bellman Eluder (BE) dimension [30, 18, 58, 27], Neural Tangent Kernel (NTK) [81, 10, 79, 10, 20, 53], and Besov regularity [68, 29, 54, 46, 24]. The Eluder dimension is at least in an exponential order even for a two-layer neural network [19], leading to uncharacterizable sample complexity. The NTK framework linearizes deep neural networks to tackle convergence in non-linear models. However, it requires using computationally inefficient extremely wide neural networks [81]. Moreover, the sample complexity can exponentially increase with the input feature dimension, necessitating a substantial number of samples for accurate estimation. Furthermore, the NTK approach fails to explain the advantages of non-linear neural networks over linear function approximation [46, 24]. Besov space requires the neural networks to be sparse and makes an unpractical assumption that the algorithm can find the global optimal of the non-convex objective function [29, 54, 46, 24]. To the best of our knowledge, only [46] considers $\varepsilon$-greedy policy with theoretical analysis applicable to DQNs. However, the model is limited to sparse neural networks, and it cannot characterize the case of varying $\varepsilon$.

**Supervised learning with neural networks.** Compared with Q-learning in RL, where the label of the sampled data depends on the currently estimated Q function, analyzing supervised learning is less challenging, where sampled data label is known and fixed across the training. Existing theoretical results for supervised learning are largely built upon NTK [28, 21, 39, 15, 43, 45], model recovery [90, 26, 3, 64, 88, 85, 87], and structured data [44, 62, 8, 1, 35, 78, 84, 42]. Due to the high non-convexity of neural networks [59], the one-hidden-layer neural network is still a state-of-the-art practice for convergence analysis and generalization guarantees. Additional assumptions, e.g., Gaussian distribution [89, 7], linear separable data [75, 9] on the data distribution, are needed for finite-sample analysis.

## 3 Problem Formulation: Notation, Background, and Algorithm

**The Markov Decision Process and Q-learning.** A discounted Markov decision process (MDP) is defined as $(\mathcal{S}, \mathcal{A}, \mathcal{P}, r, \gamma)$, where $\mathcal{S}$ is the state space, $\mathcal{A}$ is the action set. $\mathcal{P} : \mathcal{S} \times \mathcal{A} \longrightarrow \Delta(\mathcal{S})$ is the transition operator, and $p_{\boldsymbol{s},\boldsymbol{s}'}^a := \mathcal{P}(\boldsymbol{s}'|\boldsymbol{s}, a)$ denotes the transition probability from current state $\boldsymbol{s}$ and action $a$ to the next state $\boldsymbol{s}'$. In addition, $r : \mathcal{S} \times \mathcal{A} \longrightarrow [-R_{\max}, R_{\max}]$ is the reward function, and $\gamma \in (0, 1)$ is the discount factor.

At a state $\boldsymbol{s} \in \mathcal{S}$, the agent takes action $a \in \mathcal{A}$ according to some behavior policy $\pi$, denoted as $a \sim \pi(\boldsymbol{s})$ (or $a = \pi(\boldsymbol{s})$ for deterministic policy). Then, the system moves to the next state $\boldsymbol{s}'$ following the transition probability $p_{\boldsymbol{s},\boldsymbol{s}'}^a$. Meanwhile, the agent receives an immediate reward $r(\boldsymbol{s}, a)$ from the environment. Let $\{\boldsymbol{s}_i, a_i\}_{i=0}^{\infty}$ be the generated sequential data given the behavior policy $\pi$ and transition probability $\mathcal{P}$. We define the state-value function $V_\pi$ at state $s$ as

$$V^\pi(\boldsymbol{s}) = \mathbb{E}_{\pi, \mathcal{P}} \big[ \sum_{i=0}^{\infty} \gamma^i r(\boldsymbol{s}_i, a_i) \mid \boldsymbol{s}_0 = \boldsymbol{s} \big], \qquad (1)$$

which is the expected total discounted reward starting from the state $\boldsymbol{s}$. For any state-action $(\boldsymbol{s}, a)$, the corresponding Q-value (or action-value) function $Q_\pi$ of a policy $\pi$ is defined as

$$Q^\pi(\boldsymbol{s}, a) = \mathbb{E}_{\pi, \mathcal{P}} \big[ \sum_{i=0}^{\infty} \gamma^i r(\boldsymbol{s}_i, a_i) \mid \boldsymbol{s}_0 = \boldsymbol{s}, a_0 = a \big]. \qquad (2)$$

Then, the goal of the agent is to find an optimal policy $\pi^\star$ that maximizes the state-value function in (1) for all states, which is equivalent to

$$Q^\star(\boldsymbol{s}, a) := \max_\pi Q^\pi(\boldsymbol{s}, a) = r(\boldsymbol{s}, a) + \gamma \cdot \mathbb{E}_{\boldsymbol{s}'|\boldsymbol{s},a} \max_{a'} Q^\star(\boldsymbol{s}', a'), \qquad (3)$$

where (3) is known as the Bellman equation. With the optimal Q-value function $Q^\star$, the optimal policy can be derived via $\pi^\star(\boldsymbol{s}) = \operatorname{argmax}_a Q^\star(\boldsymbol{s}, a)$ [77, 66].

**The Deep Neural Network Model.** The DQN utilizes a deep neural network (DNN) $H : \mathbb{R}^d \longrightarrow \mathbb{R}$ to approximate the optimal Q-value function $Q^\star$ in (3). Specifically, given input $\boldsymbol{x} \in \mathbb{R}^d$, the output of the $L$-hidden-layer DNN with $K$ neurons in each hidden layer is defined as

$$H(\boldsymbol{W}; \boldsymbol{x}) := \mathbf{1}^\top / K \cdot \phi(\boldsymbol{W}_L^\top \cdots \phi(\boldsymbol{W}_1^\top \boldsymbol{x})), \tag{4}$$

where $\boldsymbol{W}_1 \in \mathbb{R}^{d \times K}$, $\boldsymbol{W}_l \in \mathbb{R}^{K \times K}$ with $l = 2, \cdots, L$, and $\boldsymbol{W} = [\operatorname{vec}(\boldsymbol{W}_1)^\top, \cdots, \operatorname{vec}(\boldsymbol{W}_L)^\top]^\top$ is the concatenation of the vectorization of all parameter matrices. $\phi(\cdot)$ is the nonlinear activation function, and we consider the ReLU activation function, i.e., $\phi(z) = \max\{0, z\}$. Then, the Q-value function $Q(\boldsymbol{s}, a)$ is parameterized using the DNN as

$$Q(\boldsymbol{W}; \boldsymbol{s}, a) = H(\boldsymbol{W}; \boldsymbol{x}(\boldsymbol{s}, a)), \tag{5}$$

where $\boldsymbol{x} : \mathcal{S} \times \mathcal{A} \longrightarrow \mathbb{R}^d$ is the feature mapping of the state-action pair. Without loss of generality, we assume $|\boldsymbol{x}(\boldsymbol{s}, a)| \leq 1$. Then, the goal of DQN-based Q-learning is to minimize the mean squared Bellman error (MSBE) as

$$\min_{\boldsymbol{W}} : f(\boldsymbol{W}) := \mathbb{E}_{(\boldsymbol{s},a) \sim \pi^\star} \big( Q(\boldsymbol{W}; \boldsymbol{s}, a) - r(\boldsymbol{s}, a) - \gamma \cdot \mathbb{E}_{\boldsymbol{s}'|\boldsymbol{s},a} \max_{a'} Q(\boldsymbol{W}; \boldsymbol{s}', a') \big)^2, \tag{6}$$

where $\mu$ is the distribution of $(\boldsymbol{s}, a)$ following the optimal policy $\pi^\star$.

## 3.1 The Deep Q-Network Algorithm

---
**Algorithm 1** Deep Q-Network

---
1: **Input**: Number of iterations $T \times M$, and experience replay buffer size $N$, exploration probability $\{\varepsilon_t\}_{t=1}^T$, step size $\eta$, and momentum parameter $\beta$.
2: Initialize the Q-network with weights $\boldsymbol{W}^{(0,0)}$.
3: **for** $t = 0, 1, 2, \cdots, T - 1$ **do**
4:     Select the initial weights $\boldsymbol{W}^{(t,0)}$.
5:     **for** $m = 0, 1, 2, \cdots, M - 1$ **do**
6:         Sample data and store in the experience replay buffer $\mathcal{D}_t$ following $\varepsilon_t$-greedy policy, namely, at state $\boldsymbol{s}_n$, with probability $\varepsilon_t$, select a random action $a_n$, otherwise select $a_n = \operatorname{argmax}_a Q(\boldsymbol{W}^{(t,0)}; \boldsymbol{s}_n, a)$.
7:         Sample random mini-batch of transition $\mathcal{D}_t^{(m)}$ from the replay buffer $\mathcal{D}_t$.
8:         Set $y_n = r_n + \gamma \max_a Q(\boldsymbol{W}^{(t,0)}; \boldsymbol{s}_n', a)$ for $n \in 1, 2, \cdots, |\mathcal{D}_t^{(m)}|$.
9:         Perform a gradient descent step

$$\boldsymbol{W}^{(t,m+1)} = \boldsymbol{W}^{(t,m)} - \eta \cdot g_t^{(m)}(\boldsymbol{W}^{(t,m)}) + \beta(\boldsymbol{W}^{(t,m)} - \boldsymbol{W}^{(t,m-1)}).$$

10:     **end for**
11:     Set $\boldsymbol{W}^{(t+1,0)} = \boldsymbol{W}^{(t,M)}$.
12: **end for**

---

The learning problem (6) is solved via the DQN equipped with $\varepsilon$-greedy exploration, as summarized in Algorithm 1. In $t$-th outer loop, we initialize the Q-value function using currently estimated weights $\boldsymbol{W}^{(t,0)}$ as $Q(\boldsymbol{W}^{(t,0)})$ (line 4). Then, for each inner loop, the agent selects and executes actions according to the $\varepsilon$-greedy policy (line 6), namely, with probability $\varepsilon$, we select a random action, and with probability $1 - \varepsilon$, we select the action based on the greedy policy with respect to $Q(\boldsymbol{W}^{(t,0)})$. The data are stored in a replay buffer with size $N$ (line 7). Then, we sample a mini-batch of independent samples from $\mathcal{D}_t$, denoted as $\mathcal{D}_t^{(m)}$ for the $m$-th inner loop (line 8). Next, we update the current weights using a mini-batch (accelerated) gradient descent algorithm (line 9). The gradient descent (GD) direction in this step at point $\boldsymbol{W}$ is represented as

$$g_t^{(m)}(\boldsymbol{W}) = \sum_{n \in \mathcal{D}_t^{(m)}} \big( Q(\boldsymbol{W}; \boldsymbol{s}_n, a_n) - y_n^{(t)} \big) \cdot \nabla_{\boldsymbol{W}} Q(\boldsymbol{W}; \boldsymbol{s}_n, a_n), \tag{7}$$

where

$$y_n^{(t)} = r_n + \gamma \cdot \max_{a' \in \mathcal{A}} Q(\boldsymbol{W}^{(t,0)}; \boldsymbol{s}_n', a'). \tag{8}$$

Note that (7) can be viewed as the gradient of

$$\mathbb{E}_{(\boldsymbol{s},a) \sim \mu_t} \big( Q(\boldsymbol{W}; \boldsymbol{s}, a) - r - \mathbb{E}_{\boldsymbol{s}'|\boldsymbol{s},a} \max_{a'} Q(\boldsymbol{W}^{(t,0)}; \boldsymbol{s}', a') \big)^2, \tag{9}$$

which is the approximation to (6) via fixing the $\max_a Q(\boldsymbol{W})$ as $\max_a Q(\boldsymbol{W}^{(t,0)})$. After moving along the GD direction, accelerated gradient descent (AGD) adds a momentum term, denoted by $\beta(\boldsymbol{W}^{(t,m)} - \boldsymbol{W}^{(t,m-1)})$ to accelerate the convergence rate [57]. Vanilla SGD can be viewed as a special case of AGD by letting $\beta = 0$. After updating neuron weights in the inner loop, we set the network as the currently estimated Q-value function $Q(\boldsymbol{W}^{(t,0)})$ (line 11) and repeat the steps above.

## 4 Theoretical Results

### 4.1 Takeaways of the Theoretical Findings

We consider the general setup of DQNs with $\varepsilon$-greedy under some commonly used assumptions. To the best of our knowledge, we provide the first theoretical characterization of both the convergence and sample complexity analysis for DQNs with $\varepsilon$-greedy. The major notations are summarized in Table 2. We first briefly introduce the key takeaways of our results, and the formal theoretical results are introduced in Section 4.3.

Table 2: Some Important Notations

| | | | |
|---|---|---|---|
| $K$ | Number of neurons in each hidden layer. | $L$ | Number of the hidden layers. |
| $d$ | Dimension of the feature mapping of $(\boldsymbol{s}, a)$. | $N_s$ | The sample complexity for $\delta$-optimal policy. |
| $\boldsymbol{W}^\star$ | The global optimal to (6). | $e_t$ | The value of $\|\boldsymbol{W}^{(t,0)} - \boldsymbol{W}^\star\|_F$. |
| $c_\varepsilon$ | A small positive constant with a linear dependence on $\varepsilon_t$. | $C_t$ | The fraction of actions with data at iteration $t$ such that $\mathrm{argmax}_a Q(\boldsymbol{W}^{(t,0)}; \boldsymbol{s}, a) \neq \mathrm{argmax}_a Q(\boldsymbol{W}^\star; \boldsymbol{s}, a)$. |

**(T1) Theoretical characterizations of $\{\varepsilon_t\}_{t=1}^T$ for convergence.** We prove that for a wide selection of $\{\varepsilon_t\}$ values that decrease over time, Algorithm 1 converges to $Q^\star$ linearly up to some estimation error. Let $c_\varepsilon$ measure the value level of $\varepsilon_t$'s. A higher level of $\varepsilon$ values (i.e., a larger $c_\varepsilon$) leads to an enlarged region of convergence (in the order of $c_\varepsilon$), measured by the distance from the initialization $\boldsymbol{W}^{(0,0)}$ to $\boldsymbol{W}^\star$. Thus, larger $\varepsilon$ values relax the requirements on $\boldsymbol{W}^{(0,0)}$. A lower level of $\varepsilon$ values (i.e., a smaller $c_\varepsilon$) leads to faster convergence with a rate in the order of $c_\varepsilon$. Our findings explain the intuition that the agent tends to explore more at the beginning and exploit more after gaining enough knowledge during the exploration.

**(T2) Convergence to the optimal Q-value function $Q^\star$ with geometric decay.** The learned models converge to the ground truth model $Q^\star$ with a geometric decay up to some bounded estimation error. The convergence rate is upper bounded by $\gamma + c_\varepsilon \cdot (1 - \gamma)$. When $\gamma$ is close to one, the problem emphasizes long-term rewards. While the immediate reward can be observed directly, the future reward needs to be estimated by the learned Q function as shown in (7). Therefore, with a large $\gamma$, the learned Q function needs more iterations to converge, which leads to a slow convergence rate.

**(T3) Sample complexity for achieving a desired estimation error of the optimal Q-value function.** With the proper selection of $\{\varepsilon_t\}_{t=1}^T$, the estimation error of the learned model scales in the order of $\frac{C_T}{(1-\gamma)^2 \sqrt{N_s}}$. $C_T$ is the fraction of actions following the current greedy policy that differ from the ones following the optimal policy. $N_s$ is the number of samples. With a smaller discounted factor $\gamma$, the problem focuses more on the immediate reward, which can be observed directly, making $Q^\star$ easier to learn. The learned model achieves a small estimation error given a small distribution shift, a large sample size, or a small discounted factor.

### 4.2 Assumptions

We propose assumptions commonly used in existing RL and neural network learning theories and notations to simplify the presentation. Assumption 1 assumes the existence of a good approximation of DNN to $Q^\star$, which guarantees (6) is solvable. The assumption is commonly used in both deep learning theories [90, 86] and reinforcement learning theories [46, 24]. Assumption 2 assumes the samples from experience replay are independent and identically distributed (i.i.d.), which follows the assumptions in existing theoretical analysis of DQN [24, 53] and matches the intuition of using experience replay to break temporal dependence among the samples. Specifically, we have

**Assumption 1.** *There exists a DNN with weights $\boldsymbol{W}^\star$ such that minimizes (6) as $f(\boldsymbol{W}^\star) = 0$.*

Assumption 1 assumes that the Q-value function with some unknown ground truth $\boldsymbol{W}^{\star}$ [1] can represent the optimal Q-value function.

**Assumption 2.** *Suppose the mini-batch data are i.i.d. samples from the replay buffer following the distribution $\mu_t$, which is the stationary distribution of the behavior policy at $t$-th outer loop.*

Assumption 2 assumes the mini-batch at the $t$-th outer loop are *i.i.d.* samples following $\mu_t$, where $\mu_t$ is the stationary distribution of $(\boldsymbol{s}, a)$ generated by $\varepsilon_t$-greedy policy at $t$-th outer loop [2]. The mini-batch samples are close to being independent given the experience size in practice is sufficiently large ($\sim$ millions [51]).

In what follows, we define two quantities $C_t$ and $\rho$ to simplify the presentation of theoretical results.

**Definition 1.** $C_t \in [0, 1]$ *is the fraction of non-optimal state-action pair $(\boldsymbol{s}, a)$ in the greedy policy with respect to $Q(\boldsymbol{W}^{(t,0)})$, i.e., the fraction of $(\boldsymbol{s}, a)$ pairs that satisfy*

$$a = \operatorname*{argmax}_{a'} Q(\boldsymbol{W}^{(t,0)}; \boldsymbol{s}, a') \neq \pi^{\star}(\boldsymbol{s}) \tag{10}$$

*among all $(\boldsymbol{s}, a)$ pairs following the greedy policy at $t$-th outer loop as $a = \operatorname{argmax}_{a'} Q(\boldsymbol{W}^{(t,0)}; \boldsymbol{s}, a')$.*

$C_t$ can be viewed as the difference between behavior policy and optimal policy. Theorem 1 shows the general results for any value of $C_t$. Nevertheless, the greedy policy is improved over time, e.g., updating the weights every few steps (line 11 in Algorithm 1). Hence, $C_t$ depends on $\boldsymbol{W}^{(t,0)}$ and is expected to decrease as $\boldsymbol{W}^{(t,0)}$ approaching $\boldsymbol{W}^{\star}$ [56, 92], which will be discussed in Corollary 3.

**Definition 2.** *Let $\rho$ be the value of*

$$\rho := \min_{\|\boldsymbol{\alpha}\|_2 \geq 1} \mathbb{E}_{(\boldsymbol{s},a) \sim \pi^{\star}} \left( \boldsymbol{\alpha}^{\top} \nabla_{\boldsymbol{W}} Q(\boldsymbol{W}^{\star}; \boldsymbol{s}, a) \right)^2. \tag{11}$$

$\rho$ suggests the radius of the local convex region of the objective function. We provide the lower bound for $\rho$ in Lemma 7 (see the proof in Appendix E.2), suggesting a sufficiently large local convex region near $\boldsymbol{W}^{\star}$.

## 4.3 Major Theoretical Results

Lemma 1 characterizes the convergent point when minimizing (9) using the mini-batch gradient descent in the $t$-th outer loop under certain conditions. Specifically, given that the initial weights at the $t$-th outer loop are sufficiently close to the ground truth as shown in (12) and the replay buffer is large enough as shown in (13), the distance between the learned model weights $\boldsymbol{W}^{(t+1,0)}$ and $\boldsymbol{W}^{\star}$ are bounded from above as shown in (14).

**Lemma 1** (Estimation error of $\boldsymbol{W}^{(t+1,0)}$). *Suppose Assumptions 1 & 2 hold and the initial neuron weights at the $t$-th outer loop satisfy*

$$e_t := \|\boldsymbol{W}^{(t,0)} - \boldsymbol{W}^{\star}\|_F \leq \mathcal{O}\left(1 - \frac{1 - \Theta(\varepsilon_t)}{\Theta(\sqrt{N})}\right) \cdot \frac{\rho \cdot \|\boldsymbol{W}^{\star}\|_F}{K}, \tag{12}$$

*The step size $\eta$ is $1/T$, and the size of the replay buffer is*

$$N = \Omega(\rho^{-2} \cdot K^3 \cdot L \cdot d \cdot \log q \cdot T). \tag{13}$$

*Then, with the high probability of at least $1 - q^{-d}$, the neuron weights $\boldsymbol{W}^{(t+1,0)}$ generated from Algorithm 1 satisfy*

$$\|\boldsymbol{W}^{(t+1,0)} - \boldsymbol{W}^{\star}\|_F \leq \left(1 - \Theta(\varepsilon_t)\right) \cdot \gamma \cdot \|\boldsymbol{W}^{(t,0)} - \boldsymbol{W}^{\star}\|_F + \frac{C_t + (1 - C_t)\varepsilon_t}{\Theta(\sqrt{N})} \cdot \frac{|\mathcal{A}| \cdot R_{\max}}{1 - \gamma}. \tag{14}$$

**Remark 1** (Large replay buffer reduces the estimation error and the requirement for $\boldsymbol{W}^{(t,0)}$). From (14), a larger $N$ leads to a reduced distance between $\boldsymbol{W}^{(t+1,0)}$ and $\boldsymbol{W}^{\star}$. Moreover, (12) implies that if we increase the size of replay buffer $N$, the upper bound of $e_t$ increases, indicating the algorithm can tolerant a large range of $\boldsymbol{W}^{(t,0)}$.

---

[1] Note that $\boldsymbol{W}^{\star}$ does not need to be unique. We abuse the notation $\|\boldsymbol{W} - \boldsymbol{W}^{*}\|_2$ to denote the minimum distance of $\boldsymbol{W}$ to any $\boldsymbol{W}^{*}$ satisfying Assumption 1.

[2] The framework can be extended into non i.i.d. samples, see Appendix G.

In the following corollary, we show the upper and lower bound of $\varepsilon_t$ at the $t$-th outer loop. The lower bound guarantees that the RHS of (12) is larger than 0, so we have a sufficiently large radius for convergence. The upper bound ensures (14) be less than $e_t$, indicating an improved estimation of $Q^\star$ across the iterations.

**Corollary 1** (Range of $\varepsilon$). *Given the assumptions and conditions in Theorem 1 hold. To ensure the existence of a good initialization at iteration $t$, $\varepsilon_t$ needs to satisfy*

$$\varepsilon_t \geq 1 - \Theta(\sqrt{N}) \cdot (1 - e_t), \tag{15}$$

*To ensure the estimated learned model is improving over iterations, $\varepsilon_t$ needs to satisfy*

$$\varepsilon_t \leq \frac{(1-\gamma)^2 \cdot \Theta(\sqrt{N}) \cdot e_t}{(1-C_t) \cdot |\mathcal{A}| \cdot R_{\max}} - \frac{C_t}{1-C_t}. \tag{16}$$

**Remark 2** (Reduce $\varepsilon_t$ as $t$ increases). The lower bound can always be smaller than the upper bound given a sufficiently large $N$ as shown in (13). From both (15) and (16), we know that $\varepsilon_t$ needs to decrease as $e_t$ decreases. Specifically, the lower bound of $\varepsilon_t$ is a linear function of $e_t$, and the upper bound of $\varepsilon_t$ is a linear function of $e_t$. Namely, we need a relatively large $\varepsilon_0$ at the beginning since the distance between initial point $\boldsymbol{W}^{(0,0)}$ and $\boldsymbol{W}^\star$ is large. As $t$ increases, $e_t$, which is the distance of learned neuron weights $\boldsymbol{W}^{(t,0)}$ to the ground truth $\boldsymbol{W}^\star$, becomes smaller, and we should decrease $\varepsilon_t$ to guarantee an improved $Q^{(t+1,0)}$ over $Q^{(t,0)}$.

Theorem 1 shows that the learned model from Algorithm 1 converges to the optimal Q-value function $Q^\star$ with geometric decay up to an estimation error shown in (20).

**Theorem 1** (Convergence to $Q^\star$). *Suppose Assumptions 1 and 2 hold, the buffer size $N$ satisfies (13). Let us define $C_{\max}$ be a constant that is larger than $C_t$ for $1 \leq t \leq T$, when $\varepsilon_t$ satisfy*

$$\varepsilon_t = \frac{c_\varepsilon \cdot \Theta(\sqrt{N}) \cdot e_t}{(1-C_{\max}) \cdot |\mathcal{A}| \cdot R_{\max}} - \frac{C_{\max}}{1-C_{\max}} \tag{17}$$

*for a fixed constant $c_\varepsilon \in (0, (1-\gamma)^2]$, and the initialization satisfies*

$$\|\boldsymbol{W}^{(0,0)} - \boldsymbol{W}^\star\|_F \leq \mathcal{O}\Big(1 - \frac{1-c_\varepsilon}{\Theta(\sqrt{N})}\Big) \cdot \frac{\rho \cdot \|\boldsymbol{W}^\star\|_F}{K}. \tag{18}$$

*Then, with the high probability of at least $1 - T \cdot q^{-d}$, we have*

1. *the learned weights decay geometrically with*

$$\|\boldsymbol{W}^{(t+1,0)} - \boldsymbol{W}^\star\|_F \leq \big(\gamma + c_\varepsilon \cdot (1-\gamma)\big) \cdot \|\boldsymbol{W}^{(t,0)} - \boldsymbol{W}^\star\|_F, \quad \forall t \leq \log_\gamma(1/N). \tag{19}$$

2. *the returned model $Q(\boldsymbol{W}^{(T,0)})$ exhibits an estimation error in the order of $1/\sqrt{N}$ with*

$$\sup_{(\boldsymbol{s},a)} \big|Q(\boldsymbol{W}^{(T,0)}) - Q^\star\big| = \Theta\big(\|\boldsymbol{W}^{(T,0)} - \boldsymbol{W}^\star\|_F\big) \leq \frac{C_{\max} \cdot |\mathcal{A}| \cdot R_{\max}}{(1-\gamma)^2 \cdot \Theta(\sqrt{N})}, \tag{20}$$

*where $T \geq \log_\gamma(1/N)$.*

**Remark 3** (Selection of $\varepsilon_t$). The value of $\varepsilon_t$ is influenced by three key factors: $c_\varepsilon$, $C_{\max}$, and the current estimation error bound $e_t$. The constant $c_\varepsilon$ is fixed and controls the magnitude of the values in the sequence $\varepsilon_{t\,t=1}^T$, providing a way to regulate the level of $\varepsilon_t$. Regarding $C_{\max}$, $C_t$ tends to decrease as the iteration index $t$ increases, indicating a progressive improvement in the policy and resulting in a smaller data distribution shift. Hence, to estimate $C_{\max}$, we can leverage $C_0$, which is obtained by collecting data based on the policy $\max_a Q(\boldsymbol{W}^{(0,0)}; \boldsymbol{s}, a)$. Moreover, with $c_\varepsilon$ fixed, the estimation error $e_t$ follows a geometric decay pattern, as depicted in (19). This behavior allows us to use the expression $\big(\gamma + c_\varepsilon \cdot (1-\gamma)\big)^t \cdot e_0$ as an estimate for $e_t$.

**Remark 4** (Geometric decay to $Q^\star$). From (19) and (20), we know that the learned model from the proposed algorithm converges to $Q^\star$ with a geometric decay up to some estimation error. The convergence rate is in the order of $\gamma + c_\varepsilon \cdot (1-\gamma)$, and the estimation error is in the order of $(1-\gamma)^{-2} \cdot C_{\max}/\sqrt{N}$. As we mentioned in the takeaways in Section 4.1, a small discounted factor $\gamma$ leads to a fast convergence rate. We have a reduced estimation with a large buffer with size $N$, a small distribution shift $C_T$, and a small $\gamma$.

**Remark 5** (Larger $\varepsilon$ values for enlarged region of convergence and smaller $\varepsilon$ values for faster convergence). From (17), we know that $c_\varepsilon$ controls the value level of $\varepsilon_t$. From (18), a larger $c_\varepsilon$ (i.e., a higher level of $\varepsilon$ values) increases the upper bound of $\|\boldsymbol{W}^{(0,0)} - \boldsymbol{W}^\star\|_2$ and, thus, enlarges the proper region of $\boldsymbol{W}^{(0,0)}$. (19) indicates that the convergence rate is in the order of $c_\varepsilon$. A smaller $c_\varepsilon$ (i.e., lower level of $\varepsilon$ values) leads to faster convergence.

In the following corollary, we provide the sample complexity for achieving $\delta$ estimation error as shown in (21), where $\widetilde{\Omega}(\cdot)$ omits some $\log$ factors. The corollary can be obtained by letting (20) to be less than a desired accuracy $\delta$.

**Corollary 2** (Sample complexity). *To achieve an estimation error of $\delta$, the required number of samples, referred to as the sample complexity, needs to satisfy*

$$N_s = N \cdot \log \gamma = \widetilde{\Omega}\big((1-\gamma)^4 \cdot C_{\max} \cdot |\mathcal{A}|^2 R_{\max}^2 \cdot K^3 \cdot L \cdot d \cdot T/\delta^2\big). \tag{21}$$

**Remark 6** (Sample complexity). (21) shows that the sample complexity is a linear function of $d$ and $L$, where $d$ is the feature mapping of the state-action pair $(\boldsymbol{s}, a)$ and $L$ is the number of layers. Given the freedom of degree of $\boldsymbol{W}$ is a linear function of $d$ and $L$, respectively, the sample complexity is almost order-wise optimal with respect to $d$ and $L$.

The following corollary presents a tighter bound for the model estimation error compared with (20). This improvement arises from a stronger assumption on $C_t$, which becomes a function of $\|\boldsymbol{W}^{(t,0)} - \boldsymbol{W}^\star\|_2$. Compared with (20), the estimation error bound in (24) and (25) considers the cases that the behavior policy is improved as $\boldsymbol{W}^{(t,0)}$ becomes closer to $\boldsymbol{W}^\star$. As a result, we achieve a more precise and tighter estimation of the error in the model.

**Corollary 3** (Distribution shift and estimation error). *Assume that $C_t$ is Hölder continuous with a factor $\alpha$ as*

$$C_t = \mathcal{O}(\|\boldsymbol{W}^{(t,0)} - \boldsymbol{W}^\star\|_2^\alpha), \tag{22}$$

[3] *for $0 < \alpha \leq 1$ and all $t \in [T]$. When $\varepsilon_t$ satisfy*

$$\varepsilon_t = \frac{c_\varepsilon \cdot \Theta(\sqrt{N}) \cdot e_t}{(1 - C_t) \cdot |\mathcal{A}| \cdot R_{\max}} - \frac{C_t}{1 - C_t}, \tag{23}$$

*the estimation error of the Q-function satisfies*

$$\sup_{(\boldsymbol{s},a)} \left| Q(\boldsymbol{W}^{(T,0)}) - Q^\star \right| = \frac{(|\mathcal{A}| \cdot R_{\max})^{\frac{1}{1-\alpha}}}{(1-\gamma)^{\frac{2}{1-\alpha}} \cdot \Theta\big(N^{\frac{1}{2(1-\alpha)}}\big)}. \tag{24}$$

*In the special case that $C_t = \mathcal{O}(\|\boldsymbol{W}^{(t,0)} - \boldsymbol{W}^\star\|_2)$, then*

$$\sup_{(\boldsymbol{s},a)} \left| Q(\boldsymbol{W}^{(T,0)}) - Q^\star \right| = 0. \tag{25}$$

**Remark 7** (Reduced or zero estimation error when behavior policy is improved over iterations). Recall the definition of $C_t$ in Definition 1. If $e_t$ is zero, i.e., $\boldsymbol{W}_t = \boldsymbol{W}^\star$, the action selected following the greedy policy is always the optimal action, which means that $C_t = 0$. Therefore, it is reasonable to assume that $C_t$ is Hölder continuous as shown in (22). When $\alpha > 0$, we can see that (24) is less than (20), indicating a reduced estimation error and sample complexity. Typically, if $C_t$ has an order of growth less than $e_t$ near $\boldsymbol{W}^\star$, a zero estimation error is achievable.

## 4.4 The Roadmap of Proofs, Comparison with Existing Works, and Limitations

The proof of Theorem 1 draws inspiration from the model estimation framework in the supervised learning setting [90, 88]. The key idea is to use a population risk function (PRF) to characterize the objective function in (9). By satisfying certain conditions such as having sufficient training samples and a bounded data distribution shift, the approximation error between the PRF and objective function can be bounded. This allows for the characterization of the optimization problem in (9) by analyzing the landscape and convergence properties of the PRF.

---

[3] Although this equation depends on the algorithm's trajectory, it can be easily derived from a time-independent equation $|\pi_{\boldsymbol{W}}(\boldsymbol{s}|a) - \pi^\star(\boldsymbol{s}|a) \leq C\|\boldsymbol{W} - \boldsymbol{W}^\star\|_2$. Also, this equation is a weaker condition than (2) in [92], which holds universally across the entire space and model parameter space.

In comparison to existing proofs based on the model estimation frameworks, this paper addresses two additional challenges. Firstly, it extends the proof from one-hidden-layer neural networks to multi-layer neural networks. This extension is achieved by providing new tools for characterizing the Hessian matrix (refer to Lemma 3) and concentration bound (refer to Lemmas 3 and 6). Additionally, this paper characterizes the differences between the two functions caused by the interaction of neuron weights across layers in the gradient and Hessian matrix. Secondly, the paper extends the proof from supervised learning settings to Q-learning settings. This requires characterizing the additional error term caused by the data distribution shift and "noisy" labels (refer to Lemma 3) because the empirical risk function is no longer an expectation of the defined population risk function.

Existing state-of-the-art theoretical results on Q-learning with neural network approximations primarily revolve around the NTK and Besov regularity frameworks. In the NTK framework, the networks are assumed to be extremely over-parameterized, requiring an impractical projection step and resulting in error bounds that cannot be characterized for networks with finite width. In the Besov regularity framework, the neural network needs to be sparse, which does not align with the DQN algorithm. Furthermore, the analysis in the Besov regularity framework relies on the achievability of the global minimizer of the non-convex problem in (9), which cannot be guaranteed using GD algorithms. This paper takes a significant step towards bridging the gap between theoretical understanding and practical applications of DQN by addressing these challenges. However, there still remains a gap between the theoretical results and numerical findings. Future research directions include devising efficient exploration strategies for DQNs to further enhance their performance and extending the theoretical analysis to variants of DQNs and policy gradient-based methods.

## 5 Numerical Experiments

In this section, we provide numerical justification that our theoretical findings are aligned with practical DQNs through the Atari Pong game, which is commonly used for DQNs in [50, 51, 73]. We take the Double DQN (DDQN) [73], one of the most popular variants of DQN, as the backbone in the setup. DDQN differs from DQN only in (8) via changing $y = r + \gamma \cdot Q(\boldsymbol{W}^{(t,0)}; \boldsymbol{s}', a^\star)$ to $y = r + \gamma \cdot Q(\boldsymbol{W}^{(t,m)}; \boldsymbol{s}', a^\star)$, where $a^\star = \mathrm{argmax}_a Q(\boldsymbol{W}^{(t,0)}; \boldsymbol{s}', a)$. DDQN outperforms DQN in the relief of overoptimism and the improvement of stability. Our numerical experiments on DDQN also indicates that our analysis applies to the variants of DQN.

The input to the network is $84 \times 84 \times 4$ images, where the last dimension represents the number of frames in history. The network is a convolutional neural network consisting of three convolutional layers and one fully-connected layer. The algorithm terminates if the average score over the recent 20 episodes does not improve or the algorithm reaches the maximum episode set as 200, which is around $4 \times 10^5$ training steps. The testing score is calculated based on a similar setup as the training process by fixing the maximum memory size $N$ as 2000 and greedy policy, i.e., $\varepsilon = 0$. Each point in the plot is averaged over 10 experiments with an error bar representing the standard deviation.

**Estimation errors with respect to the sample complexity $N$.** As the Q-value function is the estimate of the expected cumulative reward, we use the difference between the reward obtained from the estimated Q-value function and the maximum reward as the estimation error of the learned model to the optimal Q-value function, which is also consistent with the experiments in [50, 51, 73]. Given that the full test score in the Pong game is 21, we set the test error as the value of (*21 - test score*) in each experiment. The $\varepsilon_t$ in $\varepsilon$-greedy policy decreases geometrically from 1 to 0.01. We vary the number of samples in the replay buffer from 400 to 2500. Figure 1 shows that the test error is almost linear in $1/\sqrt{N}$, which is consistent with our characterization in (20). In addition, experiments with a large $N$ have a shorter error bar indicating a more stable learning performance with a large sample complexity as shown in (12).

**Convergence with different selections of $\varepsilon$.** Figure 2 illustrates the convergence rate when $\varepsilon_t$ in the $\varepsilon$-greedy policy changes. For each point, $\varepsilon_0$ is selected as the value in the x-axis, and we decrease $\varepsilon_t$ geometrically as the iteration $t$ increases. Each point is averaged over 5 independent trials. We can see that the convergence rate is a linear function of $c_\varepsilon$, matching our findings in (19).

**Performance with different selections of $\varepsilon$.** We investigate the effect of $\varepsilon$-greedy policy during the training. In Figure 3, we show the test scores using $\varepsilon$-greedy policy with (1) geometrically decreasing $\varepsilon$ from 1 to 0.01, (2) fixed $\varepsilon$ as 0.1, and (3) fixed $\varepsilon$ as 0. Each test score in the curve is averaged over the past 10 episodes to smooth the trend. One can observe that a gradually decreasing

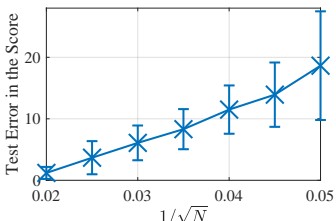
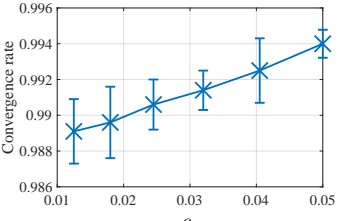
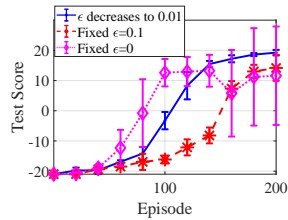

Figure 1: Test error in scores against the number of samples.

Figure 2: The convergence rate against the value of $c_\varepsilon$.

Figure 3: Test scores of the cases of $\varepsilon$-greedy down to different values.

$\varepsilon$ leads to a better score than fixing $\varepsilon$ to be the final value. The test score of $\varepsilon = 0.1$ shares a similar trend to the $\varepsilon$-greedy policy but with a slower speed, matching our findings in (19) that a small $\varepsilon$ leads to a slow convergence rate. The test score of $\varepsilon = 0$ has the fastest convergence rate at the early stage, but the convergent point is the worst and the most unstable, matching our findings in (18) that a small $\varepsilon$ leads to a reduced radius of convergence.

## 6 Conclusions and Discussions

This paper provides the first convergence and sample complexity analysis of the DQN algorithm equipped with $\varepsilon$-greedy exploration. We establish the theoretical guarantee for the convergence of the learned model to the optimal Q-value function $Q^\star$, which can be used to derive the optimal policy. We provide a nearly optimal sample complexity for achieving an arbitrarily small estimation error. We also prove that $\varepsilon$-greedy with decreasing $\varepsilon$ achieves both an enlarged radius of convergence and an improved convergence rate. Future directions include the generalization of the theoretical analysis to the variants of DQNs and the design of efficient exploration strategies for DQNs.

One of the anonymous reviewers raised concerns about (12) regarding its demanding requirements on the initial policy. We would like to clarify that (12) primarily concerns the optimization analysis of the objective function rather than the initial policy. Instead, we impose only a minor assumption on the initial policy, with no specific environmental constraints. In our work, $C_0$ quantifies the initial policy's difference from the optimal policy and is independent of (12) in our primary theoretical results. With a sufficiently large replay buffer, $C_0$ can approach one, except when it equals 1, indicating an extreme divergence from the optimal policy in all states. Thus, our initial policy assumption is minimal. Considering the highly non-convex nature of deep neural network objective functions with countless local minima, (12) represents the state-of-the-art assumption for optimizing deep neural networks.

As mentioned by the anonymous reviewers, the selections of $\varepsilon_t$ in (17) depends on $e_t = \|\boldsymbol{W}^{(t)} - \boldsymbol{W}^\star\|_2$, which is unknown to the agent. Here, we would like to clarify that $e_t$ can be replaced by its upper bound in (19) and lower bound in (20). By plugging (19) and (20) into (17), we have

$$\varepsilon_t = \max\left\{ \frac{c_\varepsilon \cdot \Theta(\sqrt{N}) \cdot \left(\gamma + c_\varepsilon \cdot (1-\gamma)\right)^t e_0}{(1 - C_{\max}) \cdot |\mathcal{A}| \cdot R_{\max}} - \frac{C_{\max}}{1 - C_{\max}}, \quad c_\varepsilon \cdot \frac{C_{\max}}{1 - C_{\max}} \right\}. \quad (26)$$

As mentioned by one of the anonymous reviewers, Corollary 3 relies on an assumption that depends on the algorithm's trajectory, which lacks mathematical rigor. Here, we would like to clarify that although this equation depends on the algorithm's trajectory, it can be easily derived from a time-independent equation

$$|\pi_{\boldsymbol{W}}(\boldsymbol{s}|a) - \pi^\star(\boldsymbol{s}|a)| \le C\|\boldsymbol{W} - \boldsymbol{W}^\star\|_2. \quad (27)$$

Additionally, it is worth mentioning the difference between (22) in this paper and (2) in [92]. Specifically, we want to highlight that the equation above, (27), leads to (22). In contrast, (2) in [92] requires the following condition to hold for for all $\boldsymbol{W}_1$ and $\boldsymbol{W}_2$:

$$|\pi_{\boldsymbol{W}_1}(\boldsymbol{s}|a) - \pi_{\boldsymbol{W}_2}(\boldsymbol{s}|a) \le C\|\boldsymbol{W}_1 - \boldsymbol{W}_2\|_2 \quad (28)$$

As a comparison, (27) only requires $\boldsymbol{W}_2$ to be the ground truth and $\boldsymbol{W}_1$ to be some weights near the ground truth. In other words, (27) is a sufficient condition for (22). While equation (2) in [92] does not hold with epsilon-greedy, (27) can hold with Q-learning using epsilon-greedy, thus ensuring the mathematical rigor of (22).

## Acknowledgment

This work was done when Shuai was a postdoc at Rensselaer Polytechnic Institute (RPI). We thank Dr. Tianyi Chen at RPI for providing remarkable and inspiring insight into the analysis of temporal difference learning. This work was supported by AFOSR FA9550-20-1-0122, ARO W911NF-21-1-0255, NSF 1932196, and the Rensselaer-IBM AI Research Collaboration (http://airc.rpi.edu), part of the IBM AI Horizons Network (http://ibm.biz/AIHorizons). We thank all anonymous reviewers for their constructive comments.

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

# Supplementary Materials for:

On the Convergence and Sample Complexity Analysis of
Deep Q-Network with Epsilon-Greedy Exploration

The structure of the appendix mainly follows the roadmap of the proof described in Section 4.4.

In Appendix A, we define the characterizable population risk function in (31) to approximate the objective function. Also, some notations to simplify the analysis are introduced in Appendix A, and we recommend the readers to refer to Table 3 for the major notations used in the proofs.

In Appendix B, we provide the proof for Lemma 1 and Theorem 1 following the steps as (1) Characterization of the local convex region of population risk function (Lemma 2), (2) Characterization of the distance between the population risk function and the objective function (Lemma 3), (3) Characterization of the convergence of two consecutive iterations $W^{(t,m+1)}$ and $W^{(t,m)}$, and (4) Mathematical induction over the $t$ and $m$ to obtain the error bound between the convergent point $W^{(T,0)}$ and the desired point $W^\star$.

In Appendix C, we provide the preliminary lemmas and the whole proof for Lemma 2, which characterizes the local convex region of the non-convex population risk function.

In Appendix D, we provide the preliminary lemmas and the whole proof for Lemma 3, which characterizes the difference of $g_t$ and the gradient descent of defined population risk function in (31).

In Appendix E, we provide the proofs for the preliminary lemmas in proving Lemmas 2 and 3.

Before moving to the details, we provide an overview of the techniques in the proofs.

**(P1.) The local convex region near $W^\star$.** To characterize the local convex region, we first bound the Hessian matrix of the defined population risk function in (31) at $W^*$. Then, we derive the changes in the Hessian matrix when the neuron weights move around the $W^*$. Specifically, we prove that when neuron weights $W$ are not far away $W^\star$, then the Hessian matrix in this region is always positive-definite, indicating that a local convex region near $W^*$. [90] considers the one-hidden-layer neural network, and the lower bound of the Hessian matrix only holds for Gaussian input. Instead, in this paper, we consider multi-layer cases and need to derive a lower bound for the Hessian matrix for all the layers. Instead, the input of the intermediate layer cannot be proved to be Gaussian but belong to sub-Gaussian distribution. Therefore, we built the proof for the lower bound of the Hessian matrix when the input belongs to the sub-Gaussian distribution. Compared with Gaussian input, Sub-Gaussian does not have a closed form of the probability density function. Instead of directly calculating the lower bound, we convert the problem into proving a series of functions are linearly independent over a Hilbert space (see Lemma 7 and the proof in Appendix E). Instead of directly calculating the distance of the population risk function in different points, we characterize a Gaussian variable such that the distance over the sub-Gaussian distribution can be upper bounded by the one over the Gaussian variable (see Lemma 6 and the proof in Appendix E).

**(P2.) The difference between the gradient $g_t$ and the population risk function.** With the local convex region of the population risk function, we can characterize the convergence of the population risk function. With Lemma 3, we can prove that the distance between the population risk function and $g_t$ is small enough, the behaviors of the iterations via $g_t$ can be described by the ones in the population risk function with some additional error terms. Compared with the proof in [90], We need to address the extension from supervised learning settings to Q learning settings and the extension from the one-hidden-layer neural networks to the multi-layer neural networks. First, similar to challenges in (P.1), we provide a new concentration bound to characterize the distance between the two functions for the intermediate layers (see $I_1$ in the proof of Lemma 3). Second, the distance between the two functions has an additional error term due to the inconsistency of the label defined in (31) and (8) (see $I_2$ in the proof of Lemma 3). Third, we need to develop a new concentration bound to characterize the error term caused by the distribution shift when training samples are collected by $\varepsilon$-greedy policy (see $I_3$ in the proof of Lemma 3).

**(P3.) The convergence analysis of Algorithm 1.** When the initialization is not far away from $\boldsymbol{W}^\star$, the initialization lies in the local convex region of $\boldsymbol{W}^\star$ for the population risk function. When we have enough samples $N$ and a large enough $\varepsilon_t$, we can guarantee that the distance between the $g_t$ and the gradient of the population risk function is small enough such that the iterations following $g_t$ converges to a point nearby $\boldsymbol{W}^\star$ as well. However, if $\varepsilon_t$ is too large, the convergent point nearby $\boldsymbol{W}^\star$ can be even worse than the initial point. To avoid this issue, we have an upper bound for selecting $\varepsilon_t$, and the upper bound decreases as $\|\boldsymbol{W}^{(t,0)} - \boldsymbol{W}^\star\|$ decreases over $t$. Therefore, we build the convergence analysis of Algorithm 1.

# A Definitions and Notations

In this section, we implement the details of algorithms described in Algorithm 1, and some important notations are defined to simplify the presentation of the proof.

## A.1 Definition of the Empirical Risk Function and Its Corresponding Notations

Recall that the goal of $Q$-learning is to find the $Q^\star$-function to minimize (6). Therefore, we have

$$Q^\star(\boldsymbol{s}, a) = r(\boldsymbol{s}, a) + \gamma \cdot \mathbb{E}_{\boldsymbol{s}'|\boldsymbol{s},a} \max_{a' \in \mathcal{A}} Q^\star(\boldsymbol{s}', a') \quad for \quad (\boldsymbol{s}, a) \sim \mu^\star. \tag{29}$$

Since $\boldsymbol{W}^\star$ is the global minimal to (6), we have

$$Q(\boldsymbol{W}^\star; \boldsymbol{s}, a) = r(\boldsymbol{s}, a) + \gamma \cdot \mathbb{E}_{\boldsymbol{s}'|\boldsymbol{s},a} \max_{a' \in \mathcal{A}} Q(\boldsymbol{W}^\star; \boldsymbol{s}', a'). \tag{30}$$

Therefore, the population risk function is defined as

$$\begin{aligned} f(\boldsymbol{W}) &= \mathbb{E}_{(\boldsymbol{s},a)\sim\mu^\star} \big[ Q(\boldsymbol{W}; \boldsymbol{s}, a) - r(\boldsymbol{s}, a) - \gamma \cdot \mathbb{E}_{\boldsymbol{s}'|\boldsymbol{s},a} \max_{a' \in \mathcal{A}} Q(\boldsymbol{W}^\star; \boldsymbol{s}', a') \big]^2 \\ &= \mathbb{E}_{(\boldsymbol{s},a)\sim\mu^\star} \big[ Q(\boldsymbol{W}; \boldsymbol{s}, a) - Q(\boldsymbol{W}^\star; \boldsymbol{s}, a) \big]^2, \end{aligned} \tag{31}$$

where $\mu^*$ is the distribution of the sampled data following the optimal policy $\pi^\star$.

The gradient of the (31) is

$$\begin{aligned} \nabla_{\boldsymbol{W}} f(\boldsymbol{W}) &= \mathbb{E}_{\boldsymbol{x}\sim\mu^\star} \big( Q(\boldsymbol{W}; \boldsymbol{x}) - r(\boldsymbol{x}) - \gamma \cdot \mathbb{E}_{\boldsymbol{s}'\sim p_{\boldsymbol{s},\boldsymbol{s}'}^a} \max_{a' \in \mathcal{A}} Q^\star(\boldsymbol{s}', a') \big) \cdot \nabla_{\boldsymbol{W}} Q(\boldsymbol{W}; \boldsymbol{x}) \\ &= \mathbb{E}_{\boldsymbol{x}\sim\mu^\star, \boldsymbol{s}'\sim p_{\boldsymbol{s},\boldsymbol{s}'}^a} \big( Q(\boldsymbol{W}; \boldsymbol{x}) - r(\boldsymbol{x}) - \gamma \cdot \max_{a' \in \mathcal{A}} Q(\boldsymbol{W}^\star; \boldsymbol{s}', a') \big) \cdot \nabla_{\boldsymbol{W}} Q(\boldsymbol{W}; \boldsymbol{x}). \end{aligned} \tag{32}$$

As $\boldsymbol{W}^\star$ is one of the ground truths to $f(\boldsymbol{W})$, i.e., $f(\boldsymbol{W}^\star)$ achieves the minimum value as $f(\boldsymbol{W}^\star) = 0 \leq f(\boldsymbol{W})$ for any other $\boldsymbol{W}$. Given $f$ is a smooth function, we have the gradient of $f$ with respect to any $\boldsymbol{W}_\ell$ at the ground truth $\boldsymbol{W}^\star$ equals to zero, namely,

$$\nabla_\ell f(\boldsymbol{W}^\star) := \nabla_{\boldsymbol{W}_\ell} f(\boldsymbol{W}^\star) = \boldsymbol{0}, \qquad \forall \ell \in [L]. \tag{33}$$

In addition, without special descriptions, $\boldsymbol{\alpha} = [\boldsymbol{\alpha}_1^\top, \boldsymbol{\alpha}_2^\top, \cdots, \boldsymbol{\alpha}_K^\top]^\top$ stands for any unit vector that in $\mathbb{R}^{K_\ell K_{\ell-1}}$ with $\boldsymbol{\alpha}_j \in \mathbb{R}_{\ell-1}^K$ ($K_0 = d$). Therefore, we have

$$\begin{aligned} \|\nabla_\ell h\|_2 &= \max_{\boldsymbol{\alpha}} \|\boldsymbol{\alpha}^\top \nabla_\ell h\|_2 = \max_{\boldsymbol{\alpha}} \Big| \sum_{j=1}^K \boldsymbol{\alpha}_j^\top \frac{\partial h}{\partial \boldsymbol{w}_{\ell,j}} \Big|, \\ \|\nabla_\ell^2 h\|_2 &= \max_{\boldsymbol{\alpha}} \|\boldsymbol{\alpha}^\top \nabla_\ell^2 h \, \boldsymbol{\alpha}\|_2 = \max_{\boldsymbol{\alpha}} \Big( \sum_{j=1}^K \boldsymbol{\alpha}_j^\top \frac{\partial h}{\partial \boldsymbol{w}_{\ell,j}} \Big)^2. \end{aligned} \tag{34}$$

## A.2 Notations in Algorithm 1

Recall that the gradient in the $t$-th loop is

$$\begin{aligned} g_t(\boldsymbol{W}) &= \frac{1}{|\mathcal{D}_t^{(m)}|} \sum_{n \in \mathcal{D}_t^{(m)}} (Q(\boldsymbol{W}; \boldsymbol{x}_n) - y_n^{(t)}) \cdot \nabla_{\boldsymbol{W}} Q(\boldsymbol{W}; \boldsymbol{x}_n) \\ &= \frac{1}{N} \sum_{n=1}^N (Q(\boldsymbol{W}; \boldsymbol{x}_n) - r(\boldsymbol{x}_n) - \gamma \cdot \max_{a' \in \mathcal{A}} Q(\boldsymbol{W}^{(t-1)}; \boldsymbol{s}'_n, a')) \cdot \nabla_{\boldsymbol{W}} Q(\boldsymbol{W}; \boldsymbol{x}_n). \end{aligned} \tag{35}$$

Then, we define $g_t^{(m)}(\boldsymbol{W}_\ell; \boldsymbol{W})$ as the components of $g_t^{(m)}(\boldsymbol{W})$ with respect to $\boldsymbol{W}_\ell$. Recall that in (4) we have

$$\boldsymbol{W} = [\text{vec}(\boldsymbol{W}_1)^\top, \quad \text{vec}(\boldsymbol{W}_2)^\top, \quad \cdots, \quad \text{vec}(\boldsymbol{W}_L)^\top]^\top. \tag{36}$$

Then, with the definition of $g_t^{(m)}(\boldsymbol{W}_\ell; \boldsymbol{W})$, we have

$$g_t^{(m)}(\boldsymbol{W}) = [g_t^{(m)}(\boldsymbol{W}_1; \boldsymbol{W})^\top, \quad g_t^{(m)}(\boldsymbol{W}_2; \boldsymbol{W})^\top, \quad \cdots, \quad g_t^{(m)}(\boldsymbol{W}_L; \boldsymbol{W})^\top]^\top. \tag{37}$$

To simplify the analysis, the update of $\boldsymbol{W}^{(t,m)}$ is analyzed in the form of

$$\boldsymbol{W}_\ell^{(t,m+1)} = \boldsymbol{W}_\ell^{(t,m)} - \eta \cdot g_t^{(m)}(\boldsymbol{W}_\ell; \boldsymbol{W}^{(t,m)}) + \beta(\boldsymbol{W}_\ell^{(t,m)} - \boldsymbol{W}_\ell^{(t,m-1)}), \quad \forall \ell \in [L]. \tag{38}$$

One can see that (38) returns the same $\boldsymbol{W}^{(t,m+1)}$ as the gradient step at line 9 in Algorithm 1.

Table 3: Notations for the proofs

| | |
|---|---|
| $g_t(\boldsymbol{W})$ | The gradient function at point $\boldsymbol{W}$ in the $t$-th outer loop, defined in (7). |
| $g_t(\boldsymbol{W}_\ell; \boldsymbol{W})$ | The gradient function of $g_t(\boldsymbol{W})$ with respect to the components of $\boldsymbol{W}_\ell$. |
| $d$ | Dimension of the feature mappings of the state-action pair $(\boldsymbol{s}, a) \in \mathcal{S} \times \mathcal{A}$. |
| $K$ | Number of neurons in the hidden layer. |
| $L$ | Number of hidden layers. |
| $\boldsymbol{W}^\star$ | The desired Weights for approximating the optimal Q function. |
| $\boldsymbol{W}^{(t,m)}$ | Model returned by Algorithm 1 at $t$-th outer loop and $m$-th inner loop. |
| $f$ | The population risk function defined in (31). |
| $\nabla_W f(\boldsymbol{W}^\star)$ | The full gradient of a function $f$ at point $\boldsymbol{W}^\star$. |
| $\nabla_\ell f(\boldsymbol{W}^\star)$ | The gradient of a function $f$ with respect to the components of $\boldsymbol{W}_\ell$ at point $\boldsymbol{W}^\star$. |
| $\nabla_\ell^2 f(\boldsymbol{W}^\star)$ | The Hessian matrix of a function $f$ with respect to the components of $\boldsymbol{W}_\ell$ at point $\boldsymbol{W}^\star$. |
| $n$ | The dimension of $\boldsymbol{W}$. |
| $n_\ell$ | The dimension of vectorized $\boldsymbol{W}_\ell$. |
| $\boldsymbol{h}^{(\ell)}(\boldsymbol{W})$ | The input to the $\ell$-th layer, defined in (39). |
| $K_\ell$ | The dimension of $\boldsymbol{h}^{(\ell)}$. |
| $\mathcal{J}_\ell(\boldsymbol{W})$ | A function in $\mathbb{R}^n \longrightarrow \mathbb{R}^K$, defined in (42). |
| $\varepsilon_t$ | The value of $\varepsilon$ in the behavior policy at $t$-th outer loop. |
| $C_t$ | The distribution shift between the optimal policy and behavior policy at iteration $t$. |
| $N$ | The size of the experience replay buffer. |
| $R_{\max}$ | The upper bound of the reward. |

## A.3 Notations for the Deep Neural Networks.

Let $n$ denote the dimension of $\boldsymbol{W}$ defined in (4). We denote $n_l$ as the dimension of the vectorized neuron weights in the $\ell$-th layer, namely, $n_\ell = \dim(\text{vec}(\boldsymbol{W}_\ell))$.

Then, let $h^{(\ell)}(\boldsymbol{W})$ denote the input in the $\ell$-th layer (or the output in the $(\ell-1)$-th layer) with respect the neuron weights as $\boldsymbol{W}$, and $h^{(1)} = (\boldsymbol{s}, a)$, where

$$\boldsymbol{h}^{(\ell)}(\boldsymbol{W}) = \phi(\boldsymbol{W}_{\ell-1}^\top \boldsymbol{h}^{(\ell-1)}) = \cdots = \phi\Big(\boldsymbol{W}_\ell^\top \phi\big(\boldsymbol{W}_{\ell-1} \cdots \phi(\boldsymbol{W}_1^\top \boldsymbol{x})\big)\Big). \tag{39}$$

$\boldsymbol{h}^{(\ell)}(\boldsymbol{W})$ may be shortened as $\boldsymbol{h}^{(\ell)}$ when the neuron weights are clear from the contexts. Then, we denote the dimension of $\boldsymbol{h}^{(\ell)}$ as $K_\ell$, where

$$K_\ell = \begin{cases} K, & if \quad \ell > 1 \\ d, & if \quad \ell = 1. \end{cases} \tag{40}$$

Then, $Q(\boldsymbol{W}; \boldsymbol{s}, a)$ can be written as

$$Q(\boldsymbol{W}; \boldsymbol{s}, a) = \frac{\mathbf{1}^\top}{K} \phi(\boldsymbol{w}_{L,k}^\top \boldsymbol{h}^{(L)}) = \frac{\mathbf{1}^\top}{K} \phi\big(\boldsymbol{W}_L^\top \phi(\boldsymbol{W}_{L-1}^\top \boldsymbol{h}^{(L-1)})\big), \tag{41}$$

where $\boldsymbol{w}_{\ell,k}$ denotes the $k$-th neuron weights in the $\ell$-th layer. Then, we define a group of functions $\mathcal{J}_\ell(\boldsymbol{W}) \in \mathbb{R}^n \longrightarrow \mathbb{R}^K$ such that

$$\mathcal{J}_\ell(\boldsymbol{W}) = \begin{cases} \big[\mathbf{1}^\top \phi'(\boldsymbol{W}_L^\top \boldsymbol{h}^{(L)})\boldsymbol{W}_L^\top \cdot \phi'(\boldsymbol{W}_{L-1}^\top \boldsymbol{h}^{(L-1)})\boldsymbol{W}_{L-1}^\top \cdots \phi'(\boldsymbol{W}_{\ell+1}^\top \boldsymbol{h}^{(\ell+1)})\boldsymbol{W}_{\ell+1}^\top\big]^\top & if \quad \ell > 1 \\ \mathbf{1} \quad if \quad \ell = 1. \end{cases}$$

$$\tag{42}$$

Then, the gradient of $Q$ can be represented as

$$\frac{\partial Q}{\partial \boldsymbol{w}_{\ell,k}}(\boldsymbol{W}) = \frac{1}{K} \mathcal{J}_{\ell,k}(\boldsymbol{W})\phi'\big(\boldsymbol{w}_{\ell,k}^\top \boldsymbol{h}^{(\ell)}(\boldsymbol{W})\big)\boldsymbol{h}^{(\ell)}(\boldsymbol{W}), \tag{43}$$

where $\mathcal{J}_{\ell,k}$ stands for the $k$-th component of $\mathcal{J}_\ell$.

### A.4 Notations for Order-wise Analysis

Without loss of generality, we consider the case that $d \gg K$. If $K \gg d$, we can always switch the order of $K$ and $d$ in the proof. Let $\sigma_i(L)$ denote the $i$-th largest singular value of $\boldsymbol{W}_L^\star$. In this paper, we consider the case that $\boldsymbol{W}_L^\star$ is will-conditioned and bounded, i.e., $\sigma_1(L)$ and $\sigma_1(L)/\sigma_K(L)$ can be viewed as the constant and will be ignored in the analysis. In addition, some constant numbers will be ignored in most steps. In particular, we use $h_1(z) \gtrsim$ (or $\lesssim, \approx)h_2(z)$ to denote there exists some positive constant $C$ such that $h_1(z) \geq$ (or $\leq, =)C \cdot h_2(z)$ when $z \in \mathbb{R}$ is sufficiently large.

## B Proof of Lemma 1 and Theorem 1

The main idea in proving Theorem 1 is to characterize the gradient descent term by the *Mean Value Theorem* (MVT) in Lemma 4 as shown in (47) and (48). The MVT is not directly applied in $g_t$ because it is not smooth. However, the population risk functions defined in (31), which are the expectations over random variables, are smooth. Lemma 2 characterizes the bounds of the Hessian matrix defined in (49). Lemma 3 characterizes the bounds of gradient differences between the population risk function defined in (31) and $g_t$ in (7) as shown in (60). Furthermore, according to Lemma 3, we know that the distance $\|\nabla_\ell f(\boldsymbol{W}) - \nabla_\ell f(\boldsymbol{W}^*)\|_2$ is upper bounded in the order of $\|\boldsymbol{W} - \boldsymbol{W}^*\|_2$ as shown in (60). Then, we can establish the connection between $\|\boldsymbol{W}^{(t,m+1)} - \boldsymbol{W}^*\|_2$ and $\|\boldsymbol{W}^{(t,m)} - \boldsymbol{W}^*\|_2$ as shown in (59). Then, by mathematical induction over $m$, one can characterize the iteration of $\{\|\boldsymbol{W}^{(t,0)} - \boldsymbol{W}^*\|_2\}_{t=1}^T$ as shown in (65), which completes the proof of Lemma 1. Finally, selecting $\varepsilon_t$ based on (68) for all $t \in [T]$, we derive the error bound of $\|\boldsymbol{W}^{(T,0)} - \boldsymbol{W}^\star\|_2$ by mathematical induction over $t$, which completes the proof of Theorem 1.

**Lemma 2.** *Given any $\boldsymbol{W} \in \mathbb{R}^n$, let $\boldsymbol{W}$ satisfy*

$$\|\boldsymbol{W} - \boldsymbol{W}^\star\|_2 \lesssim \frac{\rho \cdot c_I \cdot \sigma_K}{K} \tag{44}$$

*for some constant $c_I \in (0, 1)$. Then, for the $f$ defined in (31), we have*

$$\frac{(1 - c_I)\rho}{K^2} \preceq \nabla_\ell^2 f(\boldsymbol{W}) \preceq \frac{7}{K}. \tag{45}$$

**Lemma 3.** *Let $f$ be the function defined in (31). Let $g_t$ be the function defined in (7). Then, we have*

$$\|\nabla_\ell f(\boldsymbol{W}) - g_t(\boldsymbol{W}_\ell; \boldsymbol{W})\|_2 \lesssim \frac{2 - \varepsilon_t}{K}\sqrt{\frac{K_\ell \cdot \log q}{N}} \cdot \|\boldsymbol{W} - \boldsymbol{W}^\star\|_2$$
$$+ \frac{(1 - \varepsilon_t/2) \cdot \gamma}{K} \cdot \|\boldsymbol{W}^{(t,0)} - \boldsymbol{W}^\star\|_2 \tag{46}$$
$$+ C_d \cdot \big(C_t + (1 - C_t)\varepsilon\big) \cdot \frac{R_{\max}}{1 - \gamma}.$$

with probability at least $1 - q^{-K_\ell}$.

**Lemma 4** (Mean Value Theorem). *Let $U \subset \mathbb{R}^{d_1}$ be open and $\boldsymbol{f} : U \longrightarrow \mathbb{R}^{d_2}$ be continuously differentiable, and $\boldsymbol{x} \in U$, $\boldsymbol{h} \in \mathbb{R}^{d_1}$ vectors such that the line segment $\boldsymbol{x} + t\boldsymbol{h}$, $0 \le t \le 1$ remains in $U$. Then we have:*

$$\boldsymbol{f}(\boldsymbol{x} + \boldsymbol{h}) - \boldsymbol{f}(\boldsymbol{x}) = \left( \int_0^1 \nabla \boldsymbol{f}(\boldsymbol{x} + t\boldsymbol{h}) dt \right) \cdot \boldsymbol{h},$$

*where $\nabla \boldsymbol{f}$ denotes the Jacobian matrix of $\boldsymbol{f}$.*

*Proof of Theorem 1.* Let $\boldsymbol{W}_\ell$ denote the neuron weights in the $\ell$-th layer. From Algorithm 1 and (38), in the $s$-th iteration and $t$-th episode, we have

$$
\begin{aligned}
\boldsymbol{W}_\ell^{(t,m+1)} =\ & \boldsymbol{W}_\ell^{(t,m)} - \eta g_t^{(m)}(\boldsymbol{W}_\ell; \boldsymbol{W}^{(t,m)}) + \beta(\boldsymbol{W}_\ell^{(t,m)} - \boldsymbol{W}_\ell^{(t,m-1)}) \\
=\ & \boldsymbol{W}_\ell^{(t,m)} - \eta \nabla_\ell f(\boldsymbol{W}^{(t,m)}) + \beta(\boldsymbol{W}_\ell^{(t,m)} - \boldsymbol{W}_\ell^{(t,m-1)}) \\
& + \eta \cdot \left( \nabla_\ell f(\boldsymbol{W}^{(t,m)}) - g_t^{(m)}(\boldsymbol{W}_\ell; \boldsymbol{W}^{(t,m)}) \right).
\end{aligned}
\tag{47}
$$

From (31), we can see that $\boldsymbol{W}^\star$ is the global optimal to $f$ because $f(\boldsymbol{W}^\star)$ achieves the minimum value as 0. Therefore, we have $\nabla_\ell f_t(\boldsymbol{W}^\star) = 0$. Since $\nabla_\ell f$ is a smooth function $\boldsymbol{W}^\star$, from the *Mean Value Theorem* in Lemma 4, we have

$$
\begin{aligned}
\nabla_\ell f(\boldsymbol{W}^{(t,m)}) =\ & \nabla_\ell f(\boldsymbol{W}^{(t,m)}) - \nabla_\ell f(\boldsymbol{W}^\star) \\
=\ & \int_0^1 \nabla_\ell^2 f\left( \boldsymbol{W}^{(t,m)} + u \cdot (\boldsymbol{W}^{(t,m)} - \boldsymbol{W}^\star) \right) du \cdot (\boldsymbol{W}_\ell^{(t,m)} - \boldsymbol{W}_\ell^\star).
\end{aligned}
\tag{48}
$$

For notational convenience, we use $\boldsymbol{H}$ to denote the integration as

$$\boldsymbol{H} := \int_0^1 \nabla_\ell^2 f\left( \boldsymbol{W}^{(t,m)} + u \cdot (\boldsymbol{W}^{(t,m)} - \boldsymbol{W}^\star) \right) du. \tag{49}$$

Then, we have

$$
\begin{aligned}
\begin{bmatrix} \boldsymbol{W}^{(t,m+1)} - \boldsymbol{W}^\star \\ \boldsymbol{W}^{(t,m)} - \boldsymbol{W}^\star \end{bmatrix} =\ & \begin{bmatrix} \boldsymbol{I} - \eta \boldsymbol{H} & \beta \boldsymbol{I} \\ \boldsymbol{I} & \boldsymbol{0} \end{bmatrix} \begin{bmatrix} \boldsymbol{W}^{(t,m)} - \boldsymbol{W}^\star \\ \boldsymbol{W}^{(t,m-1)} - \boldsymbol{W}^\star \end{bmatrix} \\
& + \eta \begin{bmatrix} \nabla_\ell f(\boldsymbol{W}^{(t,m)}) - g_t^{(m)}(\boldsymbol{W}_\ell; \boldsymbol{W}^{(t,m)}) \\ \boldsymbol{0} \end{bmatrix}.
\end{aligned}
\tag{50}
$$

Let $\boldsymbol{H} = \boldsymbol{S \Lambda S}^T$ be the eigen-decomposition of $\boldsymbol{H}$. Then, we define

$$\boldsymbol{A}(\beta) := \begin{bmatrix} \boldsymbol{S}^\top & \boldsymbol{0} \\ \boldsymbol{0} & \boldsymbol{S}^\top \end{bmatrix} \boldsymbol{A}(\beta) \begin{bmatrix} \boldsymbol{S} & \boldsymbol{0} \\ \boldsymbol{0} & \boldsymbol{S} \end{bmatrix} = \begin{bmatrix} \boldsymbol{I} - \eta \boldsymbol{\Lambda} + \beta \boldsymbol{I} & \beta \boldsymbol{I} \\ \boldsymbol{I} & 0 \end{bmatrix}. \tag{51}$$

Since $\begin{bmatrix} \boldsymbol{S} & \boldsymbol{0} \\ \boldsymbol{0} & \boldsymbol{S} \end{bmatrix} \begin{bmatrix} \boldsymbol{S}^\top & \boldsymbol{0} \\ \boldsymbol{0} & \boldsymbol{S}^\top \end{bmatrix} = \begin{bmatrix} \boldsymbol{I} & \boldsymbol{0} \\ \boldsymbol{0} & \boldsymbol{I} \end{bmatrix}$, we know $\boldsymbol{A}(\beta)$ and $\begin{bmatrix} \boldsymbol{I} - \eta \boldsymbol{\Lambda} + \beta \boldsymbol{I} & \beta \boldsymbol{I} \\ \boldsymbol{I} & 0 \end{bmatrix}$ share the same eigenvalues. Let $\lambda_i^{(\boldsymbol{\Lambda})}$ be the $i$-th eigenvalue of $\boldsymbol{H}_t^{(\ell)}$, then the corresponding $i$-th eigenvalue of (51), denoted by $\lambda_i^{(\boldsymbol{A})}$, satisfies

$$(\lambda_i^{(\boldsymbol{A})}(\beta))^2 - (1 - \eta \lambda_i^{(\boldsymbol{\Lambda})} + \beta) \lambda_i^{(\boldsymbol{A})}(\beta) + \beta = 0. \tag{52}$$

By simple calculation, we have

$$|\lambda_i^{(\boldsymbol{A})}(\beta)| = \begin{cases} \sqrt{\beta}, & \text{if} \quad \beta \ge \left(1 - \sqrt{\eta \lambda_i^{(\boldsymbol{\Lambda})}}\right)^2, \\ \frac{1}{2}\left|(1 - \eta \lambda_i^{(\boldsymbol{\Lambda})} + \beta) + \sqrt{(1 - \eta \lambda_i^{(\boldsymbol{\Lambda})} + \beta)^2 - 4\beta}\right|, & \text{otherwise.} \end{cases} \tag{53}$$

Specifically, we have

$$\lambda_i^{(\boldsymbol{A})}(0) > \lambda_i^{(\boldsymbol{A})}(\beta), \quad \text{for} \quad \forall \beta \in \left(0, (1 - \eta \lambda_i^{(\boldsymbol{\Lambda})})^2\right), \tag{54}$$

and $\lambda_i^{(A)}$ achieves the minimum $\lambda_i^{(A)\star} = \left|1 - \sqrt{\eta\lambda_i^{(\Lambda)}}\right|$ when $\beta^\star = \left(1 - \sqrt{\eta\lambda_i^{(\Lambda)}}\right)^2$. From Lemma 2, for any $\boldsymbol{a} \in \mathbb{R}^d$ with $\|\boldsymbol{a}\|_2 = 1$, we have

$$
\begin{aligned}
\boldsymbol{a}^\top \nabla_\ell f(\boldsymbol{W}^{(t,m)})\boldsymbol{a} &= \int_0^1 \boldsymbol{a}^\top \nabla_\ell^2 f\left(\boldsymbol{W}^{(t,m)} + u \cdot (\boldsymbol{W}^{(t,m)} - \boldsymbol{W}^\star)\right)\boldsymbol{a} \cdot du \\
&\leq \int_0^1 \lambda_{\max}\|\boldsymbol{a}\|_2^2 du = \lambda_{\max}, \\
\boldsymbol{a}^\top \nabla_\ell f(\boldsymbol{W}^{(t,m)})\boldsymbol{a} &= \int_0^1 \boldsymbol{a}^\top \nabla_\ell^2 f\left(\boldsymbol{W}^{(t,m)} + u \cdot (\boldsymbol{W}^{(t,m)} - \boldsymbol{W}^\star)\right)\boldsymbol{a} \cdot du \\
&\geq \int_0^1 \lambda_{\min}\|\boldsymbol{a}\|_2^2 du = \lambda_{\min},
\end{aligned}
\tag{55}
$$

where $\lambda_{\max} \asymp \frac{1}{K}$, and $\lambda_{\min} \asymp \frac{\rho}{K^2}$. Therefore, we have

$$
\lambda_{\min}^{(\Lambda)} \asymp \frac{(1 - c_I)\rho}{K^2}, \quad \text{and} \quad \lambda_{\max}^{(\Lambda)} \asymp \frac{1}{K}.
\tag{56}
$$

Thus, when $\eta \leq \frac{1}{2\lambda_{\max}^{(\Lambda)}} \lesssim K$, $\|\boldsymbol{A}(\beta^\star)\|_2$ can be bounded by

$$
\|\boldsymbol{A}(\beta^\star)\|_2 = 1 - \sqrt{\eta \cdot \lambda_{\min}^{(\Lambda)}} \leq 1 - \sqrt{\frac{(1 - c_I)\eta\rho}{K^2}}.
\tag{57}
$$

Therefore, we have

$$
\begin{aligned}
\|\boldsymbol{W}_\ell^{(t,m+1)} - \boldsymbol{W}_\ell^\star\|_2 &\leq \left(1 - \sqrt{\frac{(1 - c_I)\eta\rho}{K^2}}\right) \cdot \|\boldsymbol{W}_\ell^{(t,m)} - \boldsymbol{W}_\ell^\star\|_2 \\
&\quad + \eta \cdot \|\nabla_\ell f(\boldsymbol{W}^{(t,m)}) - g_t^{(m)}(\boldsymbol{W}^{(t,m)})\|_2 \\
&\lesssim \left(1 - \left(1 - \frac{c_I}{2}\right)\sqrt{\frac{\eta\rho}{K^2}}\right) \cdot \|\boldsymbol{W}_\ell^{(t,m)} - \boldsymbol{W}_\ell^\star\|_2 \\
&\quad + \eta \cdot \|\nabla_\ell f(\boldsymbol{W}^{(t,m)}) - g_t^{(m)}(\boldsymbol{W}^{(t,m)})\|_2.
\end{aligned}
\tag{58}
$$

Take the sum of (58) from $\ell = 1$ to $\ell = L$, we have

$$
\begin{aligned}
\|\boldsymbol{W}^{(t,m+1)} - \boldsymbol{W}^\star\|_2 &\leq \left(1 - \left(1 - \frac{c_I}{2}\right)\sqrt{\frac{\eta\rho}{K^2}}\right) \cdot \|\boldsymbol{W}^{(t,m)} - \boldsymbol{W}^\star\|_2 \\
&\quad + \eta \cdot \sum_\ell^L \|\nabla_\ell f(\boldsymbol{W}^{(t,m)}) - g_t^{(m)}(\boldsymbol{W}^{(t,m)})\|_2.
\end{aligned}
\tag{59}
$$

From Lemma 3, we have

$$
\begin{aligned}
\left\|\nabla_\ell f(\boldsymbol{W}^{(t,m)}) - g_t^{(m)}(\boldsymbol{W}_\ell; \boldsymbol{W}^{(t,m)})\right\|_2 &\lesssim \frac{2 - \varepsilon_t}{K}\sqrt{\frac{K_\ell \log q}{N_t}} \cdot \|\boldsymbol{W}^{(t,m)} - \boldsymbol{W}^\star\|_2 \\
&\quad + \frac{(1 - \varepsilon_t/2)\gamma}{K} \cdot \|\boldsymbol{W}^{(t,0)} - \boldsymbol{W}^\star\|_2 \\
&\quad + C_d \cdot \left(C_t + (1 - C_t)\varepsilon\right) \cdot \frac{R_{\max}}{1 - \gamma}.
\end{aligned}
\tag{60}
$$

For some small constant $c_N \geq 0$, let

$$
\eta \cdot \frac{1}{K}\sqrt{\frac{K_\ell \log q}{N_t}} \leq \frac{c_N}{L}\sqrt{\frac{\eta\rho}{K^2}},
\tag{61}
$$

which requires

$$
\begin{aligned}
N_t &\gtrsim c_N^{-2} \cdot \rho^{-1} \cdot \eta^{-1} \cdot L^2 \cdot \max_\ell K_\ell \cdot \log q \\
&= c_N^{-2} \cdot \rho^{-1} \cdot L \cdot d \cdot \log q.
\end{aligned}
\tag{62}
$$

Then, the sample complexity

$$N = \sum_{t=1}^{T} N_t \gtrsim c_N^{-2} \cdot \rho^{-1} \cdot L \cdot d \cdot \log q \cdot T. \tag{63}$$

Therefore, we have

$$\begin{aligned}
\|\boldsymbol{W}^{(t,m+1)} - \boldsymbol{W}^{\star}\|_2 \leq & \left( 1 - \left(1 - (2 - \varepsilon_t)c_N - \frac{c_I}{2}\right)\sqrt{\frac{\rho}{TK^2}} \right) \cdot \|\boldsymbol{W}^{(t,m)} - \boldsymbol{W}^{\star}\|_2 \\
& + \sqrt{\eta} \cdot \frac{(1 - \varepsilon_t/2)\gamma}{K} \cdot \|\boldsymbol{W}^{(t,0)} - \boldsymbol{W}^{\star}\|_2 \\
& + \eta \cdot C_d \cdot \left(C_t + (1 - C_t)\varepsilon\right) \cdot \frac{R_{\max}}{1 - \gamma}.
\end{aligned} \tag{64}$$

By mathematical induction, when $M = \log \gamma^{-1}$ and $\eta = 1/T = 1/\Theta(N)$, we have

$$\begin{aligned}
& \|\boldsymbol{W}^{(t,M)} - \boldsymbol{W}^{\star}\|_2 \\
& \lesssim \sqrt{\frac{K^2}{N}} \cdot C_d \cdot \left(C_t + (1 - C_t)\varepsilon_t\right) \cdot \frac{R_{\max}}{1 - \gamma} + (1 - \varepsilon_t/2)\gamma \cdot \|\boldsymbol{W}^{(t,0)} - \boldsymbol{W}^{\star}\|_2 \\
& \leq \frac{c_N \cdot C_d \cdot \left(C_t + (1 - C_t) \cdot \varepsilon_t\right)}{K} \cdot \frac{R_{\max}}{1 - \gamma} + (1 - \varepsilon_t/2)\gamma \cdot \|\boldsymbol{W}^{(t,0)} - \boldsymbol{W}^{\star}\|_2 \\
& \leq \frac{c_N \cdot C_d \cdot \left(C_{\max} + (1 - C_{\max}) \cdot \varepsilon_t\right)}{K} \cdot \frac{R_{\max}}{1 - \gamma} + (1 - \varepsilon_t/2)\gamma \cdot \|\boldsymbol{W}^{(t,0)} - \boldsymbol{W}^{\star}\|_2.
\end{aligned} \tag{65}$$

From Algorithm 1, we know that $\boldsymbol{W}^{(t+1,0)} = \boldsymbol{W}^{(t,M)}$. To guarantee that iteration converge to the ground truth $\boldsymbol{W}^{\star}$, namely, $\|\boldsymbol{W}^{(t+1,0)} - \boldsymbol{W}^{\star}\|_2 < \|\boldsymbol{W}^{(t,0)} - \boldsymbol{W}^{\star}\|$, we need

$$\varepsilon_t \leq \frac{(1 - \gamma)^2 \cdot K \cdot \|\boldsymbol{W}^{(t,0)} - \boldsymbol{W}^{\star}\|_2}{(1 - C_t) \cdot c_N \cdot C_d \cdot R_{\max}} - \frac{C_t}{1 - C_t}. \tag{66}$$

To guarantee that $\varepsilon_T \geq 0$, then we have

$$\|\boldsymbol{W}^{(T,0)} - \boldsymbol{W}^{\star}\|_F \gtrsim \frac{C_T \cdot c_N \cdot C_d \cdot R_{\max}}{(1 - \gamma)^2 \cdot K}. \tag{67}$$

Specifically, let

$$\varepsilon_t = \frac{c_\varepsilon \cdot K \cdot \|\boldsymbol{W}^{(t,0)} - \boldsymbol{W}^{\star}\|_2}{(1 - C_t) \cdot c_N \cdot C_d \cdot R_{\max}} - \frac{C_t}{1 - C_t}, \tag{68}$$

we have

$$\begin{aligned}
\|\boldsymbol{W}^{(t+1,0)} - \boldsymbol{W}^{\star}\|_2 &\lesssim \gamma + c_\varepsilon(1 - \gamma) \cdot \|\boldsymbol{W}^{(t,0)} - \boldsymbol{W}^{\star}\|_2, \\
and \quad \|\boldsymbol{W}^{(T,0)} - \boldsymbol{W}^{\star}\|_2 &\lesssim \left[\gamma + c_\varepsilon(1 - \gamma)\right]^T \cdot \|\boldsymbol{W}^{(0,0)} - \boldsymbol{W}^{\star}\|_2,
\end{aligned} \tag{69}$$

which completes the proof.

$\square$

## C  Proof of Lemma 2

Lemma 2 provides the lower and upper bounds for the eigenvalues of the Hessian matrix of population risk function in (31). According to Weyl's inequality in Lemma 5, the eigenvalues of $\nabla_\ell^2 f(\cdot)$ at any fixed point $\boldsymbol{W}$ can be bounded in the form of (75). Therefore, we first provide the lower and upper bounds for $\nabla_\ell^2 f$ at the desired ground truth $\boldsymbol{W}^{\star}$. Then, the bounds for $\nabla_\ell^2 f$ at any other point $\boldsymbol{W}$ is bounded through (31) by utilizing the conclusion in Lemma 6. Lemma 6 illustrates the distance between the Hessian matrix of $f$ at $\boldsymbol{W}$ and $\boldsymbol{W}^*$. Lemma 7 provides the lower bound of $\mathbb{E}_{\boldsymbol{x}}\left(\sum_{j=1}^{K} \boldsymbol{\alpha}_j^\top \frac{\partial Q}{\partial \boldsymbol{w}_{\ell,k}}(\boldsymbol{W}^{\star})\right)^2$ when $\boldsymbol{x}$ belongs to sub-Gaussian distribution, which is used in proving the lower bound of the Hessian matrix in (76).

**Lemma 5** (Weyl's inequality, [5]). *Let $\boldsymbol{B} = \boldsymbol{A} + \boldsymbol{E}$ be a matrix with dimension $m \times m$. Let $\lambda_i(\boldsymbol{B})$ and $\lambda_i(\boldsymbol{A})$ be the $i$-th largest eigenvalues of $\boldsymbol{B}$ and $\boldsymbol{A}$, respectively. Then, we have*

$$|\lambda_i(\boldsymbol{B}) - \lambda_i(\boldsymbol{A})| \leq \|\boldsymbol{E}\|_2, \quad \forall \quad i \in [m]. \tag{70}$$

**Lemma 6.** *Let $f(\boldsymbol{W})$ be the population risk function defined in (31). If $\boldsymbol{W}$ is close to $\boldsymbol{W}^\star$ such that*

$$\|\boldsymbol{W} - \boldsymbol{W}^\star\|_2 \lesssim \frac{\rho}{K} \tag{71}$$

*we have*

$$\|\nabla_\ell^2 f(\boldsymbol{W}) - \nabla_\ell^2 f(\boldsymbol{W}^\star)\|_2 \lesssim \frac{1}{K} \cdot \|\boldsymbol{W} - \boldsymbol{W}^\star\|_2. \tag{72}$$

**Lemma 7.** *Suppose the following assumptions hold:*

1. *$\{\boldsymbol{w}_j\}_{j=1}^K \in \mathbb{R}^{K_\ell}$ are linear independent,*

2. *$p_H(\boldsymbol{h}) : \mathbb{R}^{K_\ell} \longrightarrow [0 \; 1]$ be the probability density for $\boldsymbol{h}$ such that $\mathbb{E}_{\boldsymbol{h}}\|\boldsymbol{h}\|_2^2 \leq +\infty$.*

*Let $\boldsymbol{\alpha} \in \mathbb{R}^{K_1 K_2}$ be the unit vector defined in (34), we have*

$$\rho := \min_{\|\boldsymbol{\alpha}\|_2=1} \int_{\mathcal{R}} \left( \sum_{j=1}^K \boldsymbol{\alpha}^\top \boldsymbol{h} \phi'(\boldsymbol{w}_{\ell,j}^\top \boldsymbol{h}) \right)^2 p_H(\boldsymbol{h}) \cdot d\boldsymbol{h} > 0, \tag{73}$$

*where $\mathcal{R} \subset \mathbb{R}^{K_\ell}$ with $\int_{\mathcal{R}} f_H(\boldsymbol{h}) > 0$. Moreover, if further assuming $\mathcal{P}$ is Gaussian distribution and $\mathcal{R} = \mathbb{R}^{K_\ell}$, we have $\rho > 0.091$.*

**Lemma 8.** *Let $\boldsymbol{h}^{(\ell)}(\boldsymbol{W})$ be the function defined in (39). When $\boldsymbol{W}$ is sufficiently close to $\boldsymbol{W}^\star$, i.e., $\|\boldsymbol{W} - \boldsymbol{W}^\star\|_2$ is smaller than some positive constant $c < 1$, we have*

$$\begin{aligned}
\|\boldsymbol{h}^{(\ell)}(\boldsymbol{W})\|_2 &\lesssim \|\boldsymbol{x}\|_2, \\
\|\boldsymbol{h}^{(\ell)}(\boldsymbol{W}) - \boldsymbol{h}^{(\ell)}(\boldsymbol{W}^\star)\|_2 &\lesssim \|\boldsymbol{W} - \boldsymbol{W}^\star\|_2 \cdot \|\boldsymbol{x}\|_2.
\end{aligned} \tag{74}$$

*Proof of Lemma 2.* Let $\lambda_{\max}(\boldsymbol{W})$ and $\lambda_{\min}(\boldsymbol{W})$ denote the largest and smallest eigenvalues of $\nabla_\ell^2 f(\boldsymbol{W})$ at a point $\boldsymbol{W}$, respectively. Then, from Lemma 5, we have

$$\begin{aligned}
\lambda_{\max}(\boldsymbol{W}) &\leq \lambda_{\max}(\boldsymbol{W}^\star) + \|\nabla_\ell^2 f(\boldsymbol{W}) - \nabla_\ell^2 f(\boldsymbol{W}^\star)\|_2, \\
\lambda_{\min}(\boldsymbol{W}) &\geq \lambda_{\min}(\boldsymbol{W}^\star) - \|\nabla_\ell^2 f(\boldsymbol{W}) - \nabla_\ell^2 f(\boldsymbol{W}^\star)\|_2.
\end{aligned} \tag{75}$$

Then, we provide the lower bound of the Hessian matrix of the population function at $\boldsymbol{W}^\star$. Let $\mathcal{P}$ be the distribution for $\boldsymbol{h}^{(\ell)}(\boldsymbol{W})$ when $\boldsymbol{x} \sim \mu_t$ with probability density function denoted as $p_H$. For any $\boldsymbol{\alpha} \in \mathbb{R}^{K_\ell K}$ defined in (34) with $\|\boldsymbol{\alpha}\|_2 = 1$, we have

$$\begin{aligned}
&\min_{\|\boldsymbol{\alpha}\|_2=1} \boldsymbol{\alpha}^\top \nabla_\ell^2 f(\boldsymbol{W}^\star) \boldsymbol{\alpha} \\
&= \frac{1}{K^2} \min_{\|\boldsymbol{\alpha}\|_2=1} \mathbb{E}_{\boldsymbol{h} \sim \mathcal{P}} \left( \sum_{j=1}^K \boldsymbol{\alpha}_j^\top \boldsymbol{h}^{(\ell)} \mathcal{J}_{\ell,k} \phi'(\boldsymbol{w}_{\ell,j}^{\star\top} \boldsymbol{h}^{(\ell)}) \right)^2 \\
&= \frac{1}{K^2} \min_{\|\boldsymbol{\alpha}\|_2=1} \int_{\mathbb{R}^{K_\ell-1}} \left( \sum_{j=1}^K \boldsymbol{\alpha}_j^\top \boldsymbol{h}^{(\ell)} \mathcal{J}_{\ell,k} \phi'(\boldsymbol{w}_{\ell,j}^{\star\top} \boldsymbol{h}^{(\ell)}) \right)^2 p_H(\boldsymbol{h}^{(\ell)}) \cdot d\boldsymbol{h}^{(\ell)} \\
&= \frac{1}{K^2} \min_{\|\boldsymbol{\alpha}\|_2=1} \int_{\{\boldsymbol{h}^{(\ell)} | \mathcal{J}_{\ell,k} \neq 0\}} \left( \sum_{j=1}^K \boldsymbol{\alpha}_j^\top \boldsymbol{h}^{(\ell)} \phi'(\boldsymbol{w}_{\ell,j}^{\star\top} \boldsymbol{h}^{(\ell)}) \right)^2 p_H(\boldsymbol{h}^{(\ell)}) \cdot d\boldsymbol{h}^{(\ell)} \\
&\gtrsim \frac{\rho}{K^2},
\end{aligned} \tag{76}$$

where the last inequality comes from Lemma 7, and Lemma 7 holds since $\boldsymbol{h}^{(\ell)}$ belongs to sub-Gaussian distribution and $\boldsymbol{W}_\ell$ is full rank.

Next, the upper bound of $\nabla_\ell^2 f$ can be bounded as

$$\max_{\|\boldsymbol{\alpha}\|_2=1} \boldsymbol{\alpha}^\top \nabla_\ell^2 f(\boldsymbol{W}^\star)\boldsymbol{\alpha}$$

$$=\frac{1}{K^2} \max_{\|\boldsymbol{\alpha}\|_2=1} \mathbb{E}_{\boldsymbol{x}}\Big(\sum_{j=1}^{K} \boldsymbol{\alpha}_j^\top \boldsymbol{h}^{(\ell)} \cdot \mathcal{J}_{\ell,k}\phi'(\boldsymbol{w}_{\ell,j}^{\star\top}\boldsymbol{h}^{(\ell)})\Big)^2$$

$$=\frac{1}{K^2} \max_{\|\boldsymbol{\alpha}\|_2=1} \mathbb{E}_{\boldsymbol{x}} \sum_{j_1=1}^{K}\sum_{j_2=1}^{K} \boldsymbol{\alpha}_{j_1}^\top \boldsymbol{h}^{(\ell)} \cdot \mathcal{J}_{\ell,k}\phi'(\boldsymbol{w}_{\ell,j_1}^{\star\top}\boldsymbol{h}^{(\ell)}) \cdot \boldsymbol{\alpha}_{j_2}^\top \boldsymbol{h}^{(\ell)} \cdot \mathcal{J}_{\ell,k}\phi'(\boldsymbol{w}_{\ell,j_2}^{\star\top}\boldsymbol{h}^{(\ell)})$$

$$=\frac{1}{K^2} \sum_{j_1=1}^{K}\sum_{j_2=1}^{K} \mathbb{E}_{\boldsymbol{x}}\boldsymbol{\alpha}_{j_1}^\top \boldsymbol{h}^{(\ell)} \cdot \mathcal{J}_{\ell,k}\phi'(\boldsymbol{w}_{\ell,j_1}^{\star T}\boldsymbol{h}^{(\ell)}) \cdot \boldsymbol{\alpha}_{j_2}^\top \boldsymbol{h}^{(\ell)} \cdot \mathcal{J}_{\ell,k}\phi'(\boldsymbol{w}_{\ell,j_2}^{\star\top}\boldsymbol{h}^{(\ell)})$$

$$\leq\frac{1}{K^2} \max_{\|\boldsymbol{\alpha}\|_2=1} \sum_{j_1=1}^{K}\sum_{j_2=1}^{K} \Big[\mathbb{E}_{\boldsymbol{x}}(\boldsymbol{\alpha}_{j_1}^\top \boldsymbol{h}^{(\ell)})^4 \cdot \mathbb{E}(\phi'(\boldsymbol{w}_{\ell,j_1}^{\star\top}\boldsymbol{h}^{(\ell)}))^4 \cdot \mathbb{E}_{\boldsymbol{x}}(\boldsymbol{\alpha}_{j_2}^\top \boldsymbol{h}^{(\ell)})^4 \cdot \mathbb{E}_{\boldsymbol{x}}(\phi'(\boldsymbol{w}_{\ell,j_2}^{\star\top}\boldsymbol{h}^{(\ell)}))^4\Big]^{1/4}$$

$$\leq\frac{1}{K^2} \max_{\|\boldsymbol{\alpha}\|_2=1} \sum_{j_1=1}^{K}\sum_{j_2=1}^{K} \Big[\mathbb{E}_{\boldsymbol{x}}(\boldsymbol{\alpha}_{j_1}^\top \boldsymbol{x})^4 \cdot \mathbb{E}_{\boldsymbol{x}}(\boldsymbol{\alpha}_{j_2}^\top \boldsymbol{x})^4\Big]^{1/4}$$

$$\leq\frac{3}{K^2} \sum_{j_1=1}^{K}\sum_{j_2=1}^{K} \|\boldsymbol{\alpha}_{j_1}\|_2 \cdot \|\boldsymbol{\alpha}_{j_2}\|_2 \leq \frac{6}{K^2} \sum_{j_1=1}^{K}\sum_{j_2=1}^{K} \frac{1}{2}\Big(\|\boldsymbol{\alpha}_{j_1}\|_2^2 + \|\boldsymbol{\alpha}_{j_2}\|_2^2\Big)$$

$$=\frac{6}{K}. \tag{77}$$

Therefore, we have

$$\lambda_{\max}(\boldsymbol{W}^\star) = \max_{\|\boldsymbol{\alpha}\|_2=1} \boldsymbol{\alpha}^\top \nabla_\ell^2 f(\boldsymbol{W}^\star;p)\boldsymbol{\alpha} \leq \frac{6}{K}. \tag{78}$$

Then, given (71), we have

$$\|\boldsymbol{W} - \boldsymbol{W}^\star\|_2 \lesssim \frac{2\rho}{K}. \tag{79}$$

Combining (79) and Lemma 6, we have

$$\|\nabla_\ell^2 f(\boldsymbol{W}) - \nabla_\ell^2 f(\boldsymbol{W}^\star)\|_2 \lesssim \frac{\rho}{K^2}. \tag{80}$$

Therefore, from (80) and (75), we have

$$\lambda_{\max}(\boldsymbol{W}) \leq \lambda_{\max}(\boldsymbol{W}^\star) + \|\nabla_\ell^2 f(\boldsymbol{W}) - \nabla_\ell^2 f(\boldsymbol{W}^\star)\|_2 \leq \frac{6}{K} + \frac{\rho}{2K^2} \leq \frac{7}{K},$$
$$\lambda_{\min}(\boldsymbol{W}) \geq \lambda_{\min}(\boldsymbol{W}^\star) - \|\nabla_\ell^2 f(\boldsymbol{W}) - \nabla_\ell^2 f(\boldsymbol{W}^\star)\|_2 \geq \frac{\rho}{K^2} - \frac{\rho}{2K^2} = \frac{\rho}{2K^2}, \tag{81}$$

which completes the proof. $\qquad\square$

## D   Proof of Lemma 3

Before illustrating the whole proof, we first introduce some preliminary lemmas and definitions. Lemma 9 is the concentration theorem for independent random matrices. The definitions of the sub-Gaussian and sub-exponential variables are summarized in Definitions 3 and 4, and it is easy to verify that any bounded variables belong to sub-Gaussian distribution. Lemmas 10 and 11 serve as the technical tools in bounding matrix norms under the framework of the confidence interval.

The error bound between $\|\nabla_\ell f - g_t\|_2$ is divided into bounding $I_1$, $I_2$, and $I_3$ as shown in (91). $I_1$ in (92) represent the deviation of the mean of several random variables to their expectation, which can be bounded through concentration inequality, i.e, Chernoff bound. $I_2$ in (93) come from the inconsistency of "noisy" label in (8) and the "ground truth" label in the population risk function (31). $I_3$ in (94) come from the data distribution shift defined in Definition 1.

**Lemma 9** ([72], Theorem 1.6). *Consider a finite sequence $\{\boldsymbol{Z}_k\}$ of independent, random matrices with dimensions $d_1 \times d_2$. Assume that such a random matrix satisfies*

$$\mathbb{E}(\boldsymbol{Z}_k) = 0 \quad and \quad \|\boldsymbol{Z}_k\| \le R \quad almost\ surely.$$

*Define*

$$\delta^2 := \max\left\{ \Big\| \sum_k \mathbb{E}(\boldsymbol{Z}_k \boldsymbol{Z}_k^\top) \Big\|, \Big\| \sum_k \mathbb{E}(\boldsymbol{Z}_k^\top \boldsymbol{Z}_k) \Big\| \right\}.$$

*Then for all $t \ge 0$, we have*

$$Prob\left\{ \Big\| \sum_k \boldsymbol{Z}_k \Big\| \ge t \right\} \le (d_1 + d_2) \exp\left( \frac{-t^2/2}{\delta^2 + Rt/3} \right).$$

**Definition 3** (Definition 5.7, [74]). *A random variable $X$ is called a sub-Gaussian random variable if it satisfies*

$$(\mathbb{E}|X|^p)^{1/p} \le c_1 \sqrt{p} \tag{82}$$

*for all $p \ge 1$ and some constant $c_1 > 0$. In addition, we have*

$$\mathbb{E}e^{s(X - \mathbb{E}X)} \le e^{c_2 \|X\|_{\psi_2}^2 s^2} \tag{83}$$

*for all $s \in \mathbb{R}$ and some constant $c_2 > 0$, where $\|X\|_{\phi_2}$ is the sub-Gaussian norm of $X$ defined as $\|X\|_{\psi_2} = \sup_{p \ge 1} p^{-1/2} (\mathbb{E}|X|^p)^{1/p}$.*

*Moreover, a random vector $\boldsymbol{X} \in \mathbb{R}^d$ belongs to the sub-Gaussian distribution if one-dimensional marginal $\boldsymbol{\alpha}^\top \boldsymbol{X}$ is sub-Gaussian for any $\boldsymbol{\alpha} \in \mathbb{R}^d$, and the sub-Gaussian norm of $\boldsymbol{X}$ is defined as $\|\boldsymbol{X}\|_{\psi_2} = \sup_{\|\boldsymbol{\alpha}\|_2=1} \|\boldsymbol{\alpha}^\top \boldsymbol{X}\|_{\psi_2}$.*

**Definition 4** (Definition 5.13, [74]). *A random variable $X$ is called a sub-exponential random variable if it satisfies*

$$(\mathbb{E}|X|^p)^{1/p} \le c_3 p \tag{84}$$

*for all $p \ge 1$ and some constant $c_3 > 0$. In addition, we have*

$$\mathbb{E}e^{s(X - \mathbb{E}X)} \le e^{c_4 \|X\|_{\psi_1}^2 s^2} \tag{85}$$

*for $s \le 1/\|X\|_{\psi_1}$ and some constant $c_4 > 0$, where $\|X\|_{\psi_1}$ is the sub-exponential norm of $X$ defined as $\|X\|_{\psi_1} = \sup_{p \ge 1} p^{-1} (\mathbb{E}|X|^p)^{1/p}$.*

**Lemma 10** (Lemma 5.2, [74]). *Let $\mathcal{B}(0,1) \in \{\boldsymbol{\alpha} \big| \|\boldsymbol{\alpha}\|_2 = 1, \boldsymbol{\alpha} \in \mathbb{R}^d\}$ denote a unit ball in $\mathbb{R}^d$. Then, a subset $\mathcal{S}_\xi$ is called a $\xi$-net of $\mathcal{B}(0,1)$ if every point $\boldsymbol{z} \in \mathcal{B}(0,1)$ can be approximated to within $\xi$ by some point $\boldsymbol{\alpha} \in \mathcal{B}(0,1)$, i.e., $\|\boldsymbol{z} - \boldsymbol{\alpha}\|_2 \le \xi$. Then the minimal cardinality of a $\xi$-net $\mathcal{S}_\xi$ satisfies*

$$|\mathcal{S}_\xi| \le (1 + 2/\xi)^d. \tag{86}$$

**Lemma 11** (Lemma 5.3, [74]). *Let $\boldsymbol{A}$ be an $d_1 \times d_2$ matrix, and let $\mathcal{S}_\xi(d)$ be a $\xi$-net of $\mathcal{B}(0,1)$ in $\mathbb{R}^d$ for some $\xi \in (0,1)$. Then*

$$\|\boldsymbol{A}\|_2 \le (1-\xi)^{-1} \max_{\boldsymbol{\alpha}_1 \in \mathcal{S}_\xi(d_1), \boldsymbol{\alpha}_2 \in \mathcal{S}_\xi(d_2)} |\boldsymbol{\alpha}_1^\top \boldsymbol{A} \boldsymbol{\alpha}_2|. \tag{87}$$

*Proof of Lemma 3.* From (7), we know that

$$
\begin{aligned}
&g_t(\boldsymbol{w}_{\ell,k}; \boldsymbol{W}) \\
&= \frac{1}{N} \sum_{n=1}^N \left( Q(\boldsymbol{W}; \boldsymbol{s}_n, a_n) - y_n^{(t)} \right) \cdot \frac{\partial Q(\boldsymbol{W}; \boldsymbol{s}_n, a_n)}{\partial \boldsymbol{w}_{\ell,k}} \\
&= \frac{1}{N} \sum_{n=1}^N \Big( Q(\boldsymbol{W}; \boldsymbol{s}_n, a_n) - Q(\boldsymbol{W}^\star; \boldsymbol{s}_n, a_n) + \gamma \cdot \max_a Q(\boldsymbol{s}_n, a; \boldsymbol{W}^\star) \\
&\qquad\qquad - \gamma \cdot \max_a Q(\boldsymbol{s}_n, a; \boldsymbol{W}^{(t,0)}) \Big) \cdot \frac{\partial Q(\boldsymbol{W}; \boldsymbol{s}_n, a_n)}{\partial \boldsymbol{w}_{\ell,k}} \\
&= \frac{1}{N} \sum_{n=1}^N \Big( Q(\boldsymbol{W}; \boldsymbol{s}_n, a_n) - Q(\boldsymbol{W}^\star; \boldsymbol{s}_n, a_n) \Big) \cdot \frac{\partial Q(\boldsymbol{W}; \boldsymbol{s}_n, a_n)}{\partial \boldsymbol{w}_{\ell,k}} \\
&\quad + \frac{1}{N} \sum_{n=1}^N \gamma \cdot \Big( \max_a Q(\boldsymbol{s}_n, a; \boldsymbol{W}^\star) - \max_a Q(\boldsymbol{s}_n, a; \boldsymbol{W}^{(t,0)}) \Big) \cdot \frac{\partial Q(\boldsymbol{W}; \boldsymbol{s}_n, a_n)}{\partial \boldsymbol{w}_{\ell,k}}.
\end{aligned}
\tag{88}
$$

From (31), we know that

$$\frac{\partial f}{\partial \boldsymbol{w}_{\ell,k}}(\boldsymbol{W}) = \mathbb{E}_{(\boldsymbol{s},a)\sim\mu^{\star}}\Big(Q(\boldsymbol{W};\boldsymbol{s},a) - Q(\boldsymbol{W}^{\star};\boldsymbol{s},a)\Big) \cdot \frac{\partial Q(\boldsymbol{W};\boldsymbol{s},a)}{\partial \boldsymbol{w}_{\ell,k}}. \tag{89}$$

Then, from (88) and (89), we have

$$g_t(\boldsymbol{w}_{\ell,k};\boldsymbol{W}) - \frac{\partial f}{\partial \boldsymbol{w}_{\ell,k}}(\boldsymbol{W}) = g_t(\boldsymbol{w}_{\ell,k};\boldsymbol{W}) - \mathbb{E}_{(\boldsymbol{s},a)\sim\mathcal{D}_t} g_t(\boldsymbol{w}_{\ell,k};\boldsymbol{W})$$
$$+ \mathbb{E}_{(\boldsymbol{s},a)\sim\mu_t} g_t(\boldsymbol{w}_{\ell,k};\boldsymbol{W}) - \frac{\partial f}{\partial \boldsymbol{w}_{\ell,k}}(\boldsymbol{W}), \tag{90}$$

where $\mathcal{D}_t$ and $\mu_t$ are equivalent because of Assumption 2. Then, we have

$$g_t(\boldsymbol{w}_{\ell,k};\boldsymbol{W}) - \frac{\partial f}{\partial \boldsymbol{w}_{\ell,k}}(\boldsymbol{W})$$
$$= \left[\frac{1}{N}\sum_{n=1}^{N}\Big(Q(\boldsymbol{W};\boldsymbol{s}_n,a_n) - Q(\boldsymbol{W}^{\star};\boldsymbol{s}_n,a_n)\Big) \cdot \frac{\partial Q(\boldsymbol{W};\boldsymbol{s}_n,a_n)}{\partial \boldsymbol{w}_{\ell,k}}\right.$$
$$\left. - \mathbb{E}_{(\boldsymbol{s},a)\sim\mu_t}\Big(Q(\boldsymbol{W};\boldsymbol{s},a) - Q(\boldsymbol{W}^{\star};\boldsymbol{s},a)\Big) \cdot \frac{\partial Q(\boldsymbol{W};\boldsymbol{s},a)}{\partial \boldsymbol{w}_{\ell,k}}\right] \tag{91}$$
$$+ \left[\frac{1}{N}\sum_{n=1}^{N}\gamma \cdot \Big(\max_a Q(\boldsymbol{s}_n,a;\boldsymbol{W}^{\star}) - \max_a Q(\boldsymbol{s}_n,a;\boldsymbol{W}^{(t,0)})\Big) \cdot \frac{\partial Q(\boldsymbol{W};\boldsymbol{s}_n,a_n)}{\partial \boldsymbol{w}_{\ell,k}}\right]$$
$$+ \mathbb{E}_{(\boldsymbol{s},a)\sim\mu_t} g_t(\boldsymbol{w}_{\ell,k};\boldsymbol{W}) - \frac{\partial f}{\partial \boldsymbol{w}_{\ell,k}}(\boldsymbol{W}).$$

For convenience, we define $\boldsymbol{I}_1$, $\boldsymbol{I}_2$, and $\boldsymbol{I}_3$ in the following ways with $\boldsymbol{x}_n := (\boldsymbol{s}_n, a_n)$ be the feature mapping of state-action pair $(\boldsymbol{s}_n, a_n)$.

Then, $\boldsymbol{I}_1$ is defined as

$$\boldsymbol{I}_1 := \frac{1}{N}\sum_{n=1}^{N}\Big(Q(\boldsymbol{W};\boldsymbol{s}_n,a_n) - Q(\boldsymbol{W}^{\star};\boldsymbol{s}_n,a_n)\Big) \cdot \frac{\partial Q(\boldsymbol{W};\boldsymbol{s}_n,a_n)}{\partial \boldsymbol{w}_{\ell,k}}$$
$$- \mathbb{E}_{(\boldsymbol{s},a)\sim\mathcal{D}_t}\Big(Q(\boldsymbol{W};\boldsymbol{s},a) - Q(\boldsymbol{W}^{\star};\boldsymbol{s},a)\Big) \cdot \frac{\partial Q(\boldsymbol{W};\boldsymbol{s},a)}{\partial \boldsymbol{w}_{\ell,k}}, \tag{92}$$

$\boldsymbol{I}_2$ is defined as

$$\boldsymbol{I}_2 := \frac{1}{N}\sum_{n=1}^{N}\gamma \cdot \Big(\max_a Q(\boldsymbol{s}_n',a;\boldsymbol{W}^{\star}) - \max_a Q(\boldsymbol{s}_n',a;\boldsymbol{W}^{(t,0)})\Big) \cdot \frac{\partial Q(\boldsymbol{W};\boldsymbol{s}_n,a_n)}{\partial \boldsymbol{w}_{\ell,k}}, \tag{93}$$

and $\boldsymbol{I}_3$ is defined as

$$\boldsymbol{I}_3 := \mathbb{E}_{(\boldsymbol{s},a)\sim\mu_t} g_t(\boldsymbol{w}_{\ell,k};\boldsymbol{W}) - \frac{\partial f}{\partial \boldsymbol{w}_{\ell,k}}(\boldsymbol{W}), \tag{94}$$

where

$$\frac{\partial Q(\boldsymbol{W};\boldsymbol{s}_n,\boldsymbol{a}_n)}{\partial \boldsymbol{w}_{\ell,k}} = \frac{1}{K}\mathcal{J}_{\ell,k}\phi'(\boldsymbol{w}_{\ell,k}^{\top}\boldsymbol{h}^{\ell})\boldsymbol{h}^{\ell}(\boldsymbol{W}) \tag{95}$$

from (43). Therefore, we have

$$\left\|g_t(\boldsymbol{w}_{\ell,k};\boldsymbol{W}) - \frac{\partial f_t}{\partial \boldsymbol{w}_{\ell,k}}(\boldsymbol{W})\right\|_2 \le \|\boldsymbol{I}_1\|_2 + \|\boldsymbol{I}_2\|_2 + \|\boldsymbol{I}_3\|_2. \tag{96}$$

Next, we will provide the bound for $\|\boldsymbol{I}_1\|_2$, $\|\boldsymbol{I}_2\|_2$, and $\|\boldsymbol{I}_3\|_2$.

**Bound of $\boldsymbol{I}_1$.** We first divide the data in $\mathcal{D}_t$ into two parts, namely, $\mathcal{D}_{t,1}$ and $\mathcal{D}_{t,2}$. $\mathcal{D}_{t,1}$ includes the state-action pair $(\boldsymbol{s},a)$ such that $a_n$ is randomly selected from action space $\mathcal{A}$, and $\mathcal{D}_{t,2}$ includes the state-action pair $(\boldsymbol{s},a)$ such that $a_n$ is selected based on the greedy policy with respect to $Q(\boldsymbol{W}^{(t,0)})$.

Then, we define a random variable $Z^{(\ell,1)} = \left(Q(\boldsymbol{x};\boldsymbol{W}) - Q(\boldsymbol{x};\boldsymbol{W}^\star)\right) \cdot \mathcal{J}_{\ell,k} \cdot \boldsymbol{\alpha}^T \boldsymbol{h}^{(\ell)}(\boldsymbol{W})$ with $\boldsymbol{x} \sim \mathcal{D}_{t,1}$ and $Z_n^{(\ell,1)} = \left(Q(\boldsymbol{x}_n;\boldsymbol{W}) - Q(\boldsymbol{x}_n;\boldsymbol{W}^\star)\right) \cdot \mathcal{J}_{\ell,k} \cdot \boldsymbol{\alpha}^T \boldsymbol{h}_n^{(\ell)}(\boldsymbol{W})$ as the realization of $Z_\ell^{(1)}$ for $n = 1, 2 \cdots, N$, where $\boldsymbol{\alpha} \in \mathbb{R}^d$ is any fixed unit vector with $\|\boldsymbol{\alpha}\|_2 \leq 1$. We know that $\boldsymbol{s}$ and $a$ are independent for $\boldsymbol{x} \sim \mathcal{D}_{t,1}$. Let $\Sigma_1$ denote the covariance matrix of $\boldsymbol{x} \sim \mathcal{D}_{t,1}$. Moreover, $\boldsymbol{x}(\boldsymbol{s},a)$ is bounded by 1, then we have $\|\Sigma_1\|_2 \leq 1$.

Similar to $Z^{(\ell,1)}$, we define a random variable $Z^{(\ell,2)} = \left(Q(\boldsymbol{x};\boldsymbol{W}) - Q(\boldsymbol{x};\boldsymbol{W}^\star)\right) \cdot \mathcal{J}_{\ell,k} \cdot \boldsymbol{\alpha}^T \boldsymbol{h}^{(\ell)}(\boldsymbol{W})$ with $\boldsymbol{x} \sim \mathcal{D}_{t,2}$ and $Z_n^{(\ell,2)} = \left(Q(\boldsymbol{x}_n;\boldsymbol{W}) - Q(\boldsymbol{x}_n;\boldsymbol{W}^\star)\right) \cdot \mathcal{J}_{\ell,k} \cdot \boldsymbol{\alpha}^T \boldsymbol{h}_n^{(\ell)}(\boldsymbol{W})$ as the realization of $Z^{(\ell,2)}$ for $n = 1, 2 \cdots, N$. Differ from $Z^{(\ell,1)}$, $\boldsymbol{s}$ and $a$ are dependent for $\boldsymbol{x} \sim \mathcal{D}_{t,2}$. Let $\Sigma_2$ denote the covariance matrix of $\boldsymbol{x} \sim \mathcal{D}_{t,1}$. Then, we have $\|\Sigma_2\|_2 \leq 1 + \max_j \rho_{x_j,a} \leq 2$, where $\rho_{x_j,a}$ denotes the correlation between $a$ and $x_j$.

According to the definition of (92), we can rewrite $\boldsymbol{I}_1$ as

$$
\begin{aligned}
\boldsymbol{I}_1 =& \frac{1}{K}\left[\frac{1}{N}\sum_{n=1}^{N}\left(Q(\boldsymbol{W};\boldsymbol{x}_n) - Q(\boldsymbol{W}^\star;\boldsymbol{x}_n)\right)\mathcal{J}_{\ell,k}\phi'(\boldsymbol{w}_{\ell,k}^\top\boldsymbol{h}_n^\ell)\boldsymbol{h}_n^\ell \right.\\
& \left. - \mathbb{E}_{\boldsymbol{x}\sim\mathcal{D}_t}\left(Q(\boldsymbol{W};\boldsymbol{x}) - Q(\boldsymbol{W}^\star;\boldsymbol{x})\right)\mathcal{J}_{\ell,k}\phi'(\boldsymbol{w}_{\ell,k}^\top\boldsymbol{h}^\ell)\boldsymbol{h}^\ell\right]\\
=& \frac{1}{K}\left[\frac{1}{N}\left(\sum_{n\in\mathcal{D}_{t,1}}\left(Q(\boldsymbol{W};\boldsymbol{x}_n) - Q(\boldsymbol{W}^\star;\boldsymbol{x}_n)\right)\mathcal{J}_{\ell,k}\phi'(\boldsymbol{w}_{\ell,k}^\top\boldsymbol{h}_n^\ell)\boldsymbol{h}_n^\ell\right.\right.\\
& \left.+ \sum_{n\in\mathcal{D}_{t,2}}\left(Q(\boldsymbol{W};\boldsymbol{x}_n) - Q(\boldsymbol{W}^\star;\boldsymbol{x}_n)\right)\mathcal{J}_{\ell,k}\phi'(\boldsymbol{w}_{\ell,k}^\top\boldsymbol{h}_n^\ell)\boldsymbol{h}_n^\ell\right)\\
& - \left(\varepsilon\mathbb{E}_{\boldsymbol{x}\sim\mathcal{D}_{t,1}}\left(Q(\boldsymbol{W};\boldsymbol{x}) - Q(\boldsymbol{W}^\star;\boldsymbol{x})\right)\mathcal{J}_{\ell,k}\phi'(\boldsymbol{w}_{\ell,k}^\top\boldsymbol{h}^\ell)\boldsymbol{h}^\ell\right.\\
& \left.\left.+ (1-\varepsilon)\mathbb{E}_{\boldsymbol{x}\sim\mathcal{D}_{t,2}}\left(Q(\boldsymbol{W};\boldsymbol{x}) - Q(\boldsymbol{W}^\star;\boldsymbol{x})\right)\mathcal{J}_{\ell,k}\phi'(\boldsymbol{w}_{\ell,k}^\top\boldsymbol{h}^\ell)\boldsymbol{h}^\ell\right)\right]\\
=& \frac{1}{K^2}\left[\varepsilon\cdot\left(\frac{1}{\varepsilon N}\sum_{n\in\mathcal{D}_{t,1}}\left(Q(\boldsymbol{W};\boldsymbol{x}_n) - Q(\boldsymbol{W}^\star;\boldsymbol{x}_n)\right)\mathcal{J}_{\ell,k}\phi'(\boldsymbol{w}_{\ell,k}^\top\boldsymbol{h}_n^\ell)\boldsymbol{h}_n^\ell\right.\right.\\
& \left.- \mathbb{E}_{\boldsymbol{x}\sim\mathcal{D}_{t,1}}\left(Q(\boldsymbol{W};\boldsymbol{x}) - Q(\boldsymbol{W}^\star;\boldsymbol{x})\right)\mathcal{J}_{\ell,k}\phi'(\boldsymbol{w}_{\ell,k}^\top\boldsymbol{h}^\ell)\boldsymbol{h}^\ell\right)\\
& + (1-\varepsilon)\left(\frac{1}{(1-\varepsilon)N}\sum_{n\in\mathcal{D}_{t,2}}\left(Q(\boldsymbol{W};\boldsymbol{x}_n) - Q(\boldsymbol{W}^\star;\boldsymbol{x}_n)\right)\mathcal{J}_{\ell,k}\phi'(\boldsymbol{w}_{\ell,k}^\top\boldsymbol{h}_n^\ell)\boldsymbol{h}_n^\ell\right.\\
& \left.\left.- \mathbb{E}_{\boldsymbol{x}\sim\mathcal{D}_{t,2}}\left(Q(\boldsymbol{W};\boldsymbol{x}) - Q(\boldsymbol{W}^\star;\boldsymbol{x})\right)\mathcal{J}_{\ell,k}\phi'(\boldsymbol{w}_{\ell,k}^\top\boldsymbol{h}^\ell)\boldsymbol{h}^\ell\right)\right]
\end{aligned}
\tag{97}
$$

Then, for any $p \in \mathbb{N}^+$, we have

$$
\begin{aligned}
\left(\mathbb{E}|Z^{(1)}|^p\right)^{1/p} =& \left(\mathbb{E}_{\boldsymbol{x}\sim\mathcal{D}_{t,1}}|Q(\boldsymbol{W};\boldsymbol{x}) - Q(\boldsymbol{W}^\star;\boldsymbol{x})|^p \cdot |\mathcal{J}_{\ell,k}\phi'(\boldsymbol{w}_{\ell,k}^\top\boldsymbol{x})| \cdot |\boldsymbol{\alpha}^T\boldsymbol{h}^\ell|^p\right)^{1/p}\\
\leq& \left(\mathbb{E}_{\boldsymbol{x}\sim\mathcal{D}_{t,1}}|Q(\boldsymbol{W};\boldsymbol{x}) - Q(\boldsymbol{W}^\star;\boldsymbol{x})|^p \cdot |\boldsymbol{\alpha}^T\boldsymbol{h}^\ell|^p\right)^{1/p}\\
\leq& \left(\mathbb{E}_{\boldsymbol{x}\sim\mathcal{D}_{t,1}}\left|\|\boldsymbol{W} - \boldsymbol{W}^\star\|_2 \cdot \|\boldsymbol{x}\|_2\right|^p \cdot |\boldsymbol{\alpha}^T\boldsymbol{x}|^p\right)^{1/p}\\
\leq& C_1 \cdot \|\boldsymbol{W} - \boldsymbol{W}^\star\|_2 \cdot p
\end{aligned}
\tag{98}
$$

where $C_1$ is a positive constant.

From Definition 4, we know that $Z^{(\ell,1)}$ belongs to sub-exponential distribution with $\|Z^{(\ell,1)}\|_{\psi_1} \leq C_1\|\boldsymbol{W} - \boldsymbol{W}^\star\|_2$. Therefore, by Chernoff inequality, we have

$$
\mathbb{P}\left\{\left|\frac{1}{N}\sum_{n=1}^{N}Z_n^{(\ell,1)}(j) - \mathbb{E}Z^{(\ell,1)}(j)\right| < t\right\} \leq 1 - \frac{e^{-C(C_1\|\boldsymbol{W} - \boldsymbol{W}^\star\|_2)^2 \cdot Ns^2}}{e^{Nst}}
\tag{99}
$$

for some positive constant $C$ and any $s \in \mathbb{R}$.

Let $t = C_1 \|\boldsymbol{W} - \boldsymbol{W}^\star\|_2 \sqrt{\frac{d \log q}{N}}$ and $s = \frac{2}{C \|\boldsymbol{W} - \boldsymbol{W}^\star\|_2} \cdot t$ for some large constant $q > 0$. Then, we have

$$\left| \frac{1}{N} \sum_{n=1}^{N} Z_n^{(\ell,1)}(j) - \mathbb{E} Z^{(\ell,1)}(j) \right| \lesssim C_1 \|\boldsymbol{W} - \boldsymbol{W}^\star\|_2 \cdot \sqrt{\frac{d \log q}{N}} \tag{100}$$

with probability at least $1 - q^{-d}$.

Similar to (98), we have

$$\left( \mathbb{E} |Z^{(\ell,2)}|^p \right)^{1/p} \leq C_2 \cdot \|\boldsymbol{W} - \boldsymbol{W}^\star\|_2 \cdot p, \tag{101}$$

where $C_2 = 2 \cdot C_1$. Then, we have

$$\left| \frac{1}{N} \sum_{n=1}^{N} Z_n^{(\ell,2)}(j) - \mathbb{E} Z^{(\ell,2)}(j) \right| \lesssim 2C_1 \|\boldsymbol{W} - \boldsymbol{W}^\star\|_2 \cdot \sqrt{\frac{d \log q}{N}} \tag{102}$$

with probability at least $1 - q^{-d}$.

From Lemma 11 and (97), we have

$$\begin{aligned}
\|\boldsymbol{I}_1\|_2 \leq & 2 \cdot \frac{1}{K^2} \Bigg[ \varepsilon \cdot \left| \frac{1}{\varepsilon N} \sum_{n \in \mathcal{D}_{t,1}} Z_n^{(\ell,1)}(j) - \mathbb{E} Z^{(\ell,1)}(j) \right| \\
& + (1 - \varepsilon) \cdot \left| \frac{1}{(1 - \varepsilon)N} \sum_{n \in \mathcal{D}_{t,2}} Z_n^{(\ell,2)}(j) - \mathbb{E} Z^{(\ell,2)}(j) \right| \Bigg] \\
\lesssim & \frac{2 - \varepsilon}{K^2} \|\boldsymbol{W} - \boldsymbol{W}^\star\|_2 \cdot \sqrt{\frac{d \log q}{N}}
\end{aligned} \tag{103}$$

with probability at least $1 - |\mathcal{S}_{\frac{1}{2}}(d)| \cdot q^{-d}$.

From Lemma 10, we know that $|\mathcal{S}_{\frac{1}{2}}(d)| \leq 5^d$. Therefore, the probability for (103) holds is at least $1 - \left( \frac{q}{5} \right)^{-d}$. Because $q \gg 5$, we denote the probability as $1 - q^{-d}$ for convenience.

**Bound of $\boldsymbol{I}_2$.** Let $a_n^\star = \arg\max_{a \in \mathcal{A}} Q(\boldsymbol{W}^\star; \boldsymbol{s}_n', \boldsymbol{a})$. While for $Q(\boldsymbol{W})$, we have

$$\max_a Q(\boldsymbol{W}; \boldsymbol{s}_n', \boldsymbol{a}) \geq Q(\boldsymbol{W}; \boldsymbol{s}_n', \boldsymbol{a}^\star). \tag{104}$$

Then, we have

$$\begin{aligned}
\max_a Q(\boldsymbol{W}^\star; \boldsymbol{s}_n', \boldsymbol{a}) - \max_a Q(\boldsymbol{W}; \boldsymbol{s}_n', \boldsymbol{a}) = & Q(\boldsymbol{W}^\star; \boldsymbol{s}_n', \boldsymbol{a}_n^\star) - \max_a Q(\boldsymbol{W}; \boldsymbol{s}_n', \boldsymbol{a}) \\
\leq & Q(\boldsymbol{W}^\star; \boldsymbol{s}_n', \boldsymbol{a}_n^\star) - Q(\boldsymbol{W}; \boldsymbol{s}_n', \boldsymbol{a}_n^\star).
\end{aligned} \tag{105}$$

Similarly to (105), let us define $\tilde{a}_n^\star = \arg\max_a Q(\boldsymbol{W}; \boldsymbol{s}_n, \boldsymbol{a})$. Then, we have

$$\max_a Q(\boldsymbol{W}^\star; \boldsymbol{s}_n', \boldsymbol{a}) - \max_a Q(\boldsymbol{W}; \boldsymbol{s}_n', \boldsymbol{a}) \geq Q(\boldsymbol{W}^\star; \boldsymbol{s}_n', \tilde{a}_n^\star) - Q(\boldsymbol{W}; \boldsymbol{s}_n', \tilde{a}_n^\star). \tag{106}$$

Combining (105) and (106), we have

$$\left| \max_a Q(\boldsymbol{W}^\star; \boldsymbol{s}_n', \boldsymbol{a}) - \max_a Q(\boldsymbol{W}; \boldsymbol{s}_n', \boldsymbol{a}) \right| \leq \max_a \left| Q(\boldsymbol{W}^\star; \boldsymbol{s}_n', \boldsymbol{a}) - Q(\boldsymbol{W}; \boldsymbol{s}_n', \boldsymbol{a}) \right|. \tag{107}$$

Following the definition of $Z^{(\ell,1)}$ in (98), we define

$$Z^{(\ell,3)}(j) = \left( \max_a Q(\boldsymbol{W}^\star; \boldsymbol{s}_n', \boldsymbol{a}) - \max_a Q(\boldsymbol{W}; \boldsymbol{s}_n', \boldsymbol{a}) \right) \cdot \mathcal{J}_{\ell,k} \phi'(\boldsymbol{w}_{\ell,k}^\top \boldsymbol{h}^{(\ell)}) \cdot \boldsymbol{\alpha}^\top \boldsymbol{h}^{(\ell)}.$$

Therefore, from (105) and (106), we know

$$\begin{aligned}
(\mathbb{E}|Z^{(3)}|^p)^{1/p} \leq & \Bigg( \mathbb{E}_{\boldsymbol{x} \sim \mathcal{D}_t} \left| \max_a Q(\boldsymbol{W}^\star; \boldsymbol{s}_n', \boldsymbol{a}) - \max_a Q(\boldsymbol{W}; \boldsymbol{s}_n', \boldsymbol{a}) \right|^p \\
& \qquad \cdot \left| \mathcal{J}_{\ell,k} \phi'(\boldsymbol{w}_{\ell,k}^\top \boldsymbol{h}^{(\ell)}) \right|^p \cdot |\boldsymbol{\alpha}^\top \boldsymbol{h}_n^{(\ell)}|^p \Bigg)^{1/p} \\
\leq & \left( \mathbb{E}_{\boldsymbol{x} \sim \mathcal{D}_t} \max_a \left| Q(\boldsymbol{W}^\star; \boldsymbol{s}_n', \boldsymbol{a}) - Q(\boldsymbol{W}; \boldsymbol{s}_n', \boldsymbol{a}) \right|^p \cdot |\boldsymbol{\alpha}^\top \boldsymbol{h}_n^{(\ell)}|^p \right)^{1/p} \\
\lesssim & (2 - \varepsilon) \cdot \|\boldsymbol{W} - \boldsymbol{W}^\star\|_2 \cdot \log |\mathcal{A}| \cdot p.
\end{aligned} \tag{108}$$

Following the steps in (98) to (100), we have

$$
\begin{aligned}
\|\boldsymbol{I}_2\|_2 &\lesssim \frac{(1-\varepsilon/2)\gamma}{K} \cdot \left( \|\boldsymbol{W} - \boldsymbol{W}^\star\|_2 \cdot \sqrt{\frac{d \cdot \log q \cdot \log |\mathcal{A}|}{N}} + \mathbb{E}Z^{(\ell,3)} \right) \\
&\lesssim \frac{(1-\varepsilon/2)\gamma}{K} \cdot \left( \|\boldsymbol{W} - \boldsymbol{W}^\star\|_2 \cdot \left( \sqrt{\frac{d \cdot \log q \cdot \log |\mathcal{A}|}{N}} + C \right) \right) \\
&\lesssim \frac{(1-\varepsilon/2)\gamma}{K} \cdot \|\boldsymbol{W} - \boldsymbol{W}^\star\|_2
\end{aligned}
\tag{109}
$$

with probability at least $1 - q^{-d}$, where the last inequality holds when $N \gtrsim d \cdot \log q \cdot \log |\mathcal{A}|$.

**Bound of $\boldsymbol{I}_3$.** We have

$$
\begin{aligned}
&I_3 \\
=&\mathbb{E}_{(\boldsymbol{s},a)\sim\mu_t}\, g_t(\boldsymbol{w}_{\ell,k}; \boldsymbol{W}) - \frac{\partial f}{\partial \boldsymbol{w}_{\ell,k}}(\boldsymbol{W}) \\
=&\mathbb{E}_{(\boldsymbol{s},a)\sim\mu_t}\left( Q(\boldsymbol{W}; \boldsymbol{s}, a) - Q(\boldsymbol{W}^\star; \boldsymbol{s}, a) \right) \cdot \frac{\partial Q(\boldsymbol{W}; \boldsymbol{s}, a)}{\partial \boldsymbol{w}_{\ell,k}} \\
&- \mathbb{E}_{(\boldsymbol{s},a)\sim\mu^\star}\left( Q(\boldsymbol{W}; \boldsymbol{s}, a) - Q(\boldsymbol{W}^\star; \boldsymbol{s}, a) \right) \cdot \frac{\partial Q(\boldsymbol{W}; \boldsymbol{s}, a)}{\partial \boldsymbol{w}_{\ell,k}} \\
=&\mathbb{E}_{(\boldsymbol{s},a)\sim\mu_t}\left( Q(\boldsymbol{W}; \boldsymbol{s}, a) - r(\boldsymbol{s}, a) - \gamma \cdot \mathbb{E}_{\boldsymbol{s}'\sim p^a_{\boldsymbol{s},\boldsymbol{s}'}} \max_{a'} Q(\boldsymbol{W}^\star; \boldsymbol{s}', a') \right) \cdot \frac{\partial Q(\boldsymbol{W}; \boldsymbol{s}, a)}{\partial \boldsymbol{w}_{\ell,k}} \\
&- \mathbb{E}_{(\boldsymbol{s},a)\sim\mu^\star}\left( Q(\boldsymbol{W}; \boldsymbol{s}, a) - r(\boldsymbol{s}, a) - \gamma \cdot \mathbb{E}_{\boldsymbol{s}'\sim p^a_{\boldsymbol{s},\boldsymbol{s}'}} \max_{a'} Q(\boldsymbol{W}^\star; \boldsymbol{s}', a') \right) \cdot \frac{\partial Q(\boldsymbol{W}; \boldsymbol{s}, a)}{\partial \boldsymbol{w}_{\ell,k}} \\
=&\mathbb{E}_{(\boldsymbol{s},a)\sim\mu_t,\boldsymbol{s}'\sim p^a_{\boldsymbol{s},\boldsymbol{s}'}}\left( Q(\boldsymbol{W}; \boldsymbol{s}, a) - r(\boldsymbol{s}, a) - \gamma \cdot \max_{a'} Q(\boldsymbol{W}^\star; \boldsymbol{s}', a') \right) \cdot \frac{\partial Q(\boldsymbol{W}; \boldsymbol{s}, a)}{\partial \boldsymbol{w}_{\ell,k}} \\
&- \mathbb{E}_{(\boldsymbol{s},a)\sim\mu^\star,\boldsymbol{s}'\sim p^a_{\boldsymbol{s},\boldsymbol{s}'}}\left( Q(\boldsymbol{W}; \boldsymbol{s}, a) - r(\boldsymbol{s}, a) - \gamma \cdot \max_{a'} Q(\boldsymbol{W}^\star; \boldsymbol{s}', a') \right) \cdot \frac{\partial Q(\boldsymbol{W}; \boldsymbol{s}, a)}{\partial \boldsymbol{w}_{\ell,k}}
\end{aligned}
\tag{110}
$$

Then, we have

$$
\begin{aligned}
|I_3| =& \left| \int_{(\boldsymbol{s},a)} \int_{\boldsymbol{s}'} \left( Q(\boldsymbol{W}; \boldsymbol{s}, a) - r(\boldsymbol{s}, a) - \gamma \cdot \max_{a'} Q(\boldsymbol{W}^\star; \boldsymbol{s}', a') \right) \cdot \frac{\partial Q(\boldsymbol{W}; \boldsymbol{s}, a)}{\partial \boldsymbol{w}_{\ell,k}} \right. \\
& \left. \cdot \left( \mu^\star(d\boldsymbol{s}, da)\mathcal{P}(d\boldsymbol{s}'|\boldsymbol{s}, a) - \mu_t(d\boldsymbol{s}, da)\mathcal{P}(d\boldsymbol{s}'|\boldsymbol{s}, a) \right) \right| \\
\leq& \left| Q(\boldsymbol{W}; \boldsymbol{s}, a) - r(\boldsymbol{s}, a) - \gamma \cdot \max_{a'} Q(\boldsymbol{W}^\star; \boldsymbol{s}', a') \right| \cdot \left| \frac{\partial Q(\boldsymbol{W}; \boldsymbol{s}, a)}{\partial \boldsymbol{w}_{\ell,k}} \right| \\
& \cdot \left| \int_{(\boldsymbol{s},a)} \int_{\boldsymbol{s}'} \left( \mu^\star(d\boldsymbol{s}, da)\mathcal{P}(d\boldsymbol{s}'|\boldsymbol{s}, a) - \mu_t(d\boldsymbol{s}, da)\mathcal{P}(d\boldsymbol{s}'|\boldsymbol{s}, a) \right) \right| \\
=& \left| Q(\boldsymbol{W}; \boldsymbol{s}, a) - r(\boldsymbol{s}, a) - \gamma \cdot \max_{a'} Q(\boldsymbol{W}^\star; \boldsymbol{s}', a') \right| \cdot \left| \frac{\partial Q(\boldsymbol{W}; \boldsymbol{s}, a)}{\partial \boldsymbol{w}_{\ell,k}} \right| \\
& \cdot \left[ (1-\varepsilon) \cdot \left| \int_{(\boldsymbol{s},a)} \int_{\boldsymbol{s}'} \left( \mu^\star(d\boldsymbol{s}, da)\mathcal{P}(d\boldsymbol{s}'|\boldsymbol{s}, a) - \mu_{t,1}(d\boldsymbol{s}, da)\mathcal{P}(d\boldsymbol{s}'|\boldsymbol{s}, a) \right) \right| \right. \\
& \left. + \varepsilon \cdot \left| \int_{(\boldsymbol{s},a)} \int_{\boldsymbol{s}'} \left( \mu^\star(d\boldsymbol{s}, da)\mathcal{P}(d\boldsymbol{s}'|\boldsymbol{s}, a) - \mu_{t,2}(d\boldsymbol{s}, da)\mathcal{P}(d\boldsymbol{s}'|\boldsymbol{s}, a) \right) \right| \right] \\
\leq& \frac{R_{\max}}{1-\gamma} \cdot \left[ (1-\varepsilon) \cdot \left| \int_{(\boldsymbol{s},a)} \int_{\boldsymbol{s}'} \left( \mu^\star(d\boldsymbol{s}, da)\mathcal{P}(d\boldsymbol{s}'|\boldsymbol{s}, a) - \mu_{t,1}(d\boldsymbol{s}, da)\mathcal{P}(d\boldsymbol{s}'|\boldsymbol{s}, a) \right) \right| \right. \\
& \left. + \varepsilon \cdot \left| \int_{(\boldsymbol{s},a)} \int_{\boldsymbol{s}'} \left( \mu^\star(d\boldsymbol{s}, da)\mathcal{P}(d\boldsymbol{s}'|\boldsymbol{s}, a) - \mu_{t,2}(d\boldsymbol{s}, da)\mathcal{P}(d\boldsymbol{s}'|\boldsymbol{s}, a) \right) \right| \right].
\end{aligned}
\tag{111}
$$

Then, we have

$$
\begin{aligned}
&\left| \int_{(\boldsymbol{s},a)} \int_{\boldsymbol{s}'} \left( \mu^\star(d\boldsymbol{s}, da)\mathcal{P}(d\boldsymbol{s}'|\boldsymbol{s}, a) - \mu_{t,1}(d\boldsymbol{s}, da)\mathcal{P}(d\boldsymbol{s}'|\boldsymbol{s}, a) \right) \right| \\
&= \left| \int_{(\boldsymbol{s},a)} \int_{\boldsymbol{s}'} \left( \mathcal{P}^\star(d\boldsymbol{s})\pi^\star(da|\boldsymbol{s})\mathcal{P}(d\boldsymbol{s}'|\boldsymbol{s}, a) - \mathcal{P}_{t,1}(d\boldsymbol{s})\pi_{t,1}(da|d\boldsymbol{s})\mathcal{P}(d\boldsymbol{s}'|\boldsymbol{s}, a) \right) \right| \\
&\leq \left| \int_{(\boldsymbol{s},a)} \int_{\boldsymbol{s}'} \left( \mathcal{P}^\star(d\boldsymbol{s}) - \mathcal{P}_{t,1}(d\boldsymbol{s}) \right)\pi^\star(da|\boldsymbol{s})\mathcal{P}(d\boldsymbol{s}'|\boldsymbol{s}, a) \right| \\
&\quad + \left| \int_{(\boldsymbol{s},a)} \int_{\boldsymbol{s}'} \mathcal{P}_{t,1}(d\boldsymbol{s})\left( \pi_{t,1}(da|d\boldsymbol{s}) - \pi^\star(da|d\boldsymbol{s}) \right)\mathcal{P}(d\boldsymbol{s}'|\boldsymbol{s}, a) \right| \\
&\leq |\mathcal{A}| \cdot C_t.
\end{aligned}
\tag{112}
$$

Therefore, the bound of $I_3$ can be found as

$$
\begin{aligned}
|I_3| &\lesssim \frac{R_{\max}}{1-\gamma} \cdot |\mathcal{A}| \cdot \left( (1-\varepsilon)C_t + \varepsilon \cdot C_t \right) \\
&= C_d \cdot \left( C_t + (1-C_t)\varepsilon \right) \cdot \frac{R_{\max}}{1-\gamma},
\end{aligned}
\tag{113}
$$

where $C_d = |\mathcal{A}|$.

In conclusion, let $\boldsymbol{\alpha} \in \mathbb{R}^{Kd}$ and $\boldsymbol{\alpha}_j \in \mathbb{R}^d$ with $\boldsymbol{\alpha} = [\boldsymbol{\alpha}_1^T, \boldsymbol{\alpha}_2^T, \cdots, \boldsymbol{\alpha}_K^T]^T$, we have

$$
\begin{aligned}
&\|g_t(\boldsymbol{W}) - \nabla f_t(\boldsymbol{W})\|_2 \\
&= \left| \boldsymbol{\alpha}^T \left( g_t(\boldsymbol{W}) - \nabla f_t(\boldsymbol{W}) \right) \right| \\
&\leq \sum_{k=1}^K \left| \boldsymbol{\alpha}_k^T \left( g_t(\boldsymbol{w}_{\ell,k}; \boldsymbol{W}) - \frac{\partial f}{\partial \boldsymbol{w}_{\ell,k}}(\boldsymbol{W}) \right) \right| \\
&\leq \sum_{k=1}^K \left\| g_t(\boldsymbol{w}_{\ell,k}; \boldsymbol{W}) - \frac{\partial f}{\partial \boldsymbol{w}_{\ell,k}}(\boldsymbol{W}) \right\|_2 \cdot \|\boldsymbol{\alpha}_k\|_2 \\
&\leq \sum_{k=1}^K (\|\boldsymbol{I}_1\|_2 + \|\boldsymbol{I}_2\|_2 + \|\boldsymbol{I}_3\|_2) \cdot \|\boldsymbol{\alpha}_k\|_2 \\
&\leq \frac{2-\varepsilon}{K}\sqrt{\frac{d\log q}{N}} \cdot \|\boldsymbol{W} - \boldsymbol{W}^\star\|_2 + \frac{(1-\varepsilon/2)\gamma}{K} \cdot \|\boldsymbol{W}^{(t,0)} - \boldsymbol{W}^\star\|_2 \\
&\quad + C_d \cdot \left( C_t + (1-C_t)\varepsilon \right) \cdot \frac{R_{\max}}{1-\gamma}
\end{aligned}
\tag{114}
$$

with probability at least $1 - q^{-d}$. $\qquad\square$

# E   Additional proof of the lemmas in Appendix C

## E.1   Proof of Lemma 6

The distance of the second order derivatives of the population risk function $f(\cdot)$ at point $\boldsymbol{W}$ and $\boldsymbol{W}^\star$ can be converted into bounding $\boldsymbol{P}_1$, $\boldsymbol{P}_2$, which are defined in (116). The major idea in proving $\boldsymbol{P}_1$ is to connect the error bound to the angle between $\boldsymbol{W}$ and $\boldsymbol{W}^\star$ given $\boldsymbol{h}^{(\ell)}$ belongs to the sub-Gaussian distribution.

*Proof of Lemma 6.* From the definition of $f$ in (31), we have

$$
\begin{aligned}
\frac{\partial^2 f}{\partial \boldsymbol{w}_{\ell,j_1} \partial \boldsymbol{w}_{\ell,j_2}}(\boldsymbol{W}^\star) &= \frac{1}{K^2}\mathbb{E}_{\boldsymbol{x}} \mathcal{J}_{\ell,k}\phi'(\boldsymbol{w}_{j_1}^{\star\top}\boldsymbol{h}) \cdot \mathcal{J}_{\ell,k}\phi'(\boldsymbol{w}_{j_2}^{\star\top}\boldsymbol{h}) \cdot \boldsymbol{h}^\star \boldsymbol{h}^{\star\top}, \\
\text{and} \quad \frac{\partial^2 f}{\partial \boldsymbol{w}_{\ell,j_1} \partial \boldsymbol{w}_{\ell,j_2}}(\boldsymbol{W}) &= \frac{1}{K^2}\mathbb{E}_{\boldsymbol{x}} \phi' \mathcal{J}_{\ell,k}^\star(\boldsymbol{w}_{\ell,j_1}^\top\boldsymbol{h}) \cdot \mathcal{J}_{\ell,k}^\star\phi'(\boldsymbol{w}_{\ell,j_2}^\top\boldsymbol{h}) \cdot \boldsymbol{h}\boldsymbol{h}^\top,
\end{aligned}
\tag{115}
$$

where $\boldsymbol{h} = \boldsymbol{h}^{(\ell)}(\boldsymbol{W})$ and $\boldsymbol{h}^\star = \boldsymbol{h}^{(\ell)}(\boldsymbol{W}^\star)$.

Then, we have

$$\frac{\partial^2 f}{\partial w_{\ell,j_1} \partial w_{\ell,j_2}}(\boldsymbol{W}^*) - \frac{\partial^2 f}{\partial w_{\ell,j_1} \partial w_{\ell,j_2}}(\boldsymbol{W})$$

$$= \frac{1}{K^2} \mathbb{E}_{\boldsymbol{x}} \left[ \mathcal{J}^\star_{\ell,k} \phi'(\boldsymbol{w}^{\star T}_{\ell,j_1} \boldsymbol{h}^\star) \mathcal{J}^\star_{\ell,k} \phi'(\boldsymbol{w}^{\star T}_{\ell,j_2} \boldsymbol{h}^\star) \boldsymbol{h}^\star \boldsymbol{h}^{\star\top} - \mathcal{J}_{\ell,k} \phi'(\boldsymbol{w}^\top_{\ell,j_1} \boldsymbol{h}) \mathcal{J}_{\ell,k} \mathcal{J}_{\ell,k} \phi'(\boldsymbol{w}^\top_{\ell,j_2} \boldsymbol{h}) \boldsymbol{h} \boldsymbol{h}^\top \right]$$

$$= \frac{1}{K^2} \mathbb{E}_{\boldsymbol{x}} \left[ \mathcal{J}^\star_{\ell,k} \phi'(\boldsymbol{w}^{\star T}_{\ell,j_1} \boldsymbol{h}^\star) \big( \mathcal{J}^\star_{\ell,k} \phi'(\boldsymbol{w}^{\star T}_{\ell,j_2} \boldsymbol{h}^\star) \boldsymbol{h}^\star \boldsymbol{h}^{\star\top} - \mathcal{J}_{\ell,k} \phi'(\boldsymbol{w}^\top_{\ell,j_2} \boldsymbol{h}) \boldsymbol{h} \boldsymbol{h}^\top \big) \right.$$

$$\left. + \mathcal{J}_{\ell,k} \phi'(\boldsymbol{w}^\top_{\ell,j_2} \boldsymbol{h}) \big( \mathcal{J}^\star_{\ell,k} \phi'(\boldsymbol{w}^{\star T}_{\ell,j_1} \boldsymbol{h}) \boldsymbol{h}^\star \boldsymbol{h}^{\star\top} - \mathcal{J}_{\ell,k} \phi'(\boldsymbol{w}^\top_{\ell,j_1} \boldsymbol{h}) \boldsymbol{h} \boldsymbol{h}^\top \big) \right]$$

$$:= \frac{1}{K^2} (\boldsymbol{P}_1 + \boldsymbol{P}_2). \tag{116}$$

For any $\boldsymbol{a} \in \mathbb{R}^{K_\ell}$ with $\|\boldsymbol{a}\|_2 = 1$, we have

$$\boldsymbol{a}^\top \boldsymbol{P}_1 \boldsymbol{a} = \mathbb{E}_{\boldsymbol{x}} \mathcal{J}^\star_{\ell,k} \phi'(\boldsymbol{w}^{\star T}_{\ell,j_1} \boldsymbol{h}^\star) \Big( \mathcal{J}^\star_{\ell,k} \phi'(\boldsymbol{w}^{\star T}_{\ell,j_2} \boldsymbol{h}^\star)(\boldsymbol{a}^\top \boldsymbol{h}^\star)^2 - \mathcal{J}_{\ell,k} \phi'(\boldsymbol{w}^\top_{\ell,j_2} \boldsymbol{h})(\boldsymbol{a}^\top \boldsymbol{h})^2 \Big). \tag{117}$$

Then, we have

$$|\boldsymbol{a}^\top \boldsymbol{P}_1 \boldsymbol{a}| = \left| \mathbb{E}_{\boldsymbol{x}} \mathcal{J}^\star_{\ell,k} \phi'(\boldsymbol{w}^{\star T}_{\ell,j_1} \boldsymbol{h}^\star) \Big( \mathcal{J}^\star_{\ell,k} \phi'(\boldsymbol{w}^{\star T}_{\ell,j_2} \boldsymbol{h}^\star)(\boldsymbol{a}^\top \boldsymbol{h}^\star)^2 - \mathcal{J}_{\ell,k} \phi'(\boldsymbol{w}^\top_{\ell,j_2} \boldsymbol{h})(\boldsymbol{a}^\top \boldsymbol{h})^2 \Big) \right|$$

$$\leq \mathbb{E}_{\boldsymbol{x}} \left| \mathcal{J}^\star_{\ell,k} \phi'(\boldsymbol{w}^{\star T}_{\ell,j_2} \boldsymbol{h}^\star)(\boldsymbol{a}^\top \boldsymbol{h}^\star)^2 - \mathcal{J}_{\ell,k} \phi'(\boldsymbol{w}^\top_{\ell,j_2} \boldsymbol{h})(\boldsymbol{a}^\top \boldsymbol{h})^2 \right|$$

$$\leq \mathbb{E}_{\boldsymbol{x}} \left| \mathcal{J}^\star_{\ell,k} \phi'(\boldsymbol{w}^{\star T}_{\ell,j_2} \boldsymbol{h}^\star)(\boldsymbol{a}^\top \boldsymbol{h}^\star)^2 - \mathcal{J}^\star_{\ell,k} \phi'(\boldsymbol{w}^{\star T}_{\ell,j_2} \boldsymbol{h}^\star)(\boldsymbol{a}^\top \boldsymbol{h})^2 \right|$$

$$+ \mathbb{E}_{\boldsymbol{x}} \left| \mathcal{J}^\star_{\ell,k} \phi'(\boldsymbol{w}^{\star\top}_{\ell,j_2} \boldsymbol{h}^\star)(\boldsymbol{a}^\top \boldsymbol{h})^2 - \mathcal{J}_{\ell,k} \phi'(\boldsymbol{w}^{\star\top}_{\ell,j_2} \boldsymbol{h})(\boldsymbol{a}^\top \boldsymbol{h})^2 \right|$$

$$+ \mathbb{E}_{\boldsymbol{x}} \left| \mathcal{J}_{\ell,k} \phi'(\boldsymbol{w}^{\star\top}_{\ell,j_2} \boldsymbol{h})(\boldsymbol{a}^\top \boldsymbol{h})^2 - \mathcal{J}_{\ell,k} \phi'(\boldsymbol{w}^\top_{\ell,j_2} \boldsymbol{h})(\boldsymbol{a}^\top \boldsymbol{h})^2 \right|$$

$$\lesssim \|\boldsymbol{W} - \boldsymbol{W}^\star\|_2 + \|\boldsymbol{W} - \boldsymbol{W}^\star\|_2$$

$$+ \mathbb{E}_{\boldsymbol{x}} \left| \big( \phi'(\boldsymbol{w}^{\star\top}_{\ell,j_2} \boldsymbol{h}) - \phi'(\boldsymbol{w}^{\star\top}_{\ell,j_2} \boldsymbol{h}) \big) \cdot (\boldsymbol{a}^\top \boldsymbol{h})^2 \right|$$

$$\lesssim \|\boldsymbol{W} - \boldsymbol{W}^\star\|_2 + \mathbb{E}_{\boldsymbol{x}} \left| \big( \phi'(\boldsymbol{w}^\top_{\ell,j_2} \boldsymbol{h}) - \phi'(\boldsymbol{w}^{\star\top}_{\ell,j_2} \boldsymbol{h}) \big) \cdot (\boldsymbol{a}^\top \boldsymbol{h})^2 \right|. \tag{118}$$

Utilizing the Gram-Schmidt process, we can demonstrate the existence of a set of normalized orthonormal vectors denoted as $\mathcal{B} = \{\boldsymbol{a}, \boldsymbol{b}, \boldsymbol{c}, \boldsymbol{a}_4^\perp, \cdots, \boldsymbol{a}_d^\perp\} \in \mathbb{R}^d$. This set forms an orthogonal and normalized basis for $\mathbb{R}^d$, wherein the subspace spanned by $\boldsymbol{a}, \boldsymbol{b}, \boldsymbol{c}$ includes $\boldsymbol{a}, \boldsymbol{w}_{\ell,j_2}$, and $\boldsymbol{w}^*_{\ell,j_2}$. Then, for any $\boldsymbol{x} \in \mathbb{R}^d$, we have a unique $\boldsymbol{z} = [z_1, z_2, \cdots, z_d]^\top$ such that

$$\boldsymbol{h} = z_1 \boldsymbol{a} + z_2 \boldsymbol{b} + z_3 \boldsymbol{c} + \cdots + z_d \boldsymbol{a}_d^\perp.$$

Because (i) $\boldsymbol{a}, \boldsymbol{w}_{\ell,j_2}$, and $\boldsymbol{w}^*_{\ell,j_2}$ belongs to the subspace spanned by vectors $\{\boldsymbol{a}, \boldsymbol{b}, \boldsymbol{c}\}$ and (ii) $\boldsymbol{a}_4^\perp, \cdots, \boldsymbol{a}_d^\perp, \cdots$ are orthogonal to $\boldsymbol{a}, \boldsymbol{b}$, and $\boldsymbol{c}$. Then, we know that

$$\boldsymbol{w}^{\star\top}_{\ell,j_2} \boldsymbol{h} = \boldsymbol{w}^{\star\top}_{\ell,j_2} (z_1 \boldsymbol{a} + z_2 \boldsymbol{b} + z_3 \boldsymbol{c} + \cdots + z_d \boldsymbol{a}_d^\perp)$$

$$= z_1 \boldsymbol{w}^{\star\top}_{\ell,j_2} \boldsymbol{a} + z_2 \boldsymbol{w}^{\star\top}_{\ell,j_2} \boldsymbol{b} + z_3 \boldsymbol{w}^{\star\top}_{\ell,j_2} \boldsymbol{c} + \cdots + z_d \boldsymbol{w}^{\star\top}_{\ell,j_2} \boldsymbol{a}_d^\perp$$

$$= z_1 \boldsymbol{w}^{\star\top}_{\ell,j_2} \boldsymbol{a} + z_2 \boldsymbol{w}^{\star\top}_{\ell,j_2} \boldsymbol{b} + z_3 \boldsymbol{w}^{\star\top}_{\ell,j_2} \boldsymbol{c} + 0 \tag{119}$$

$$= \boldsymbol{w}^{\star\top}_{\ell,j_2} (z_1 \boldsymbol{a} + z_2 \boldsymbol{b} + z_3 \boldsymbol{c})$$

$$:= \boldsymbol{w}^{\star\top}_{\ell,j_2} \widetilde{\boldsymbol{h}}.$$

where $\widetilde{\boldsymbol{h}} = z_1 \boldsymbol{a} + z_2 \boldsymbol{b} + z_3 \boldsymbol{c}$. Similar to (119), we have $\boldsymbol{w}^\top_{\ell,j_2} \boldsymbol{h} = \boldsymbol{w}^\top_{\ell,j_2} \widetilde{\boldsymbol{h}}$ and $\boldsymbol{a}^\top \boldsymbol{h} = \boldsymbol{a}^\top \widetilde{\boldsymbol{h}}$.

Then, we define $I_4$ as

$$I_4 := \mathbb{E}_{\boldsymbol{h}} \left| \big( \phi'(\boldsymbol{w}^{\star\top}_{\ell,j_2} \boldsymbol{h}) - \phi'(\boldsymbol{w}^\top_{\ell,j_2} \boldsymbol{h}) \big) \cdot (\boldsymbol{a}^\top \boldsymbol{h}) \right|$$

$$= \int_{\mathcal{R}_{\boldsymbol{h}}} |\phi'(\boldsymbol{w}^\top_{\ell,j_2} \boldsymbol{h}) - \phi'(\boldsymbol{w}^{\star T}_{\ell,j_2} \boldsymbol{h})| \cdot |\boldsymbol{a}^\top \boldsymbol{h}|^2 \cdot f_H(\boldsymbol{h}) d\boldsymbol{h} \tag{120}$$

$$= \int_{\mathcal{R}_{\boldsymbol{z}}} |\phi'(\boldsymbol{w}^\top_{\ell,j_2} \boldsymbol{h}) - \phi'(\boldsymbol{w}^{\star T}_{\ell,j_2} \boldsymbol{h})| \cdot |\boldsymbol{a}^\top \boldsymbol{h}|^2 \cdot f_Z(\boldsymbol{z}) \cdot |\boldsymbol{J}_{\boldsymbol{h}}(\boldsymbol{z})| d\boldsymbol{z}$$

where $|\boldsymbol{J_h}(\boldsymbol{z})|$ is the determinant of the Jacobian matrix $\frac{\partial \boldsymbol{h}}{\partial \boldsymbol{z}}$. Since $\boldsymbol{z}$ is a representation of $\boldsymbol{h}$ based on an orthogonal and normalized basis, we have $|\boldsymbol{J_h}(\boldsymbol{z})| = 1$. According to (119), $I_4$ can be rewritten as

$$
\begin{aligned}
I_4 &= \int_{\mathcal{R}_z} |\phi'(\boldsymbol{w}_{\ell,j_2}^\top \widetilde{\boldsymbol{h}}) - \phi'(\boldsymbol{w}_{\ell,j_2}^{\star T} \widetilde{\boldsymbol{h}})| \cdot |\boldsymbol{a}^\top \widetilde{\boldsymbol{h}}|^2 \cdot f_Z(\boldsymbol{z}) d\boldsymbol{z} \\
&= \int_{\mathcal{R}_z} |\phi'(\boldsymbol{w}_{\ell,j_2}^\top \widetilde{\boldsymbol{h}}) - \phi'(\boldsymbol{w}_{\ell,j_2}^{\star T} \widetilde{\boldsymbol{h}})| \cdot |\boldsymbol{a}^\top \widetilde{\boldsymbol{h}}|^2 \cdot f_Z(z_1, z_2, z_3) dz_1 dz_2 dz_3
\end{aligned}
\tag{121}
$$

where in the last equality we abuse $f_Z(z_1, z_2, z_3)$ to represent the probability density function of $(z_1, z_2, z_3)$ defined in region $\mathcal{R}_z$.

Next, we show that $\boldsymbol{z}$ is rotational invariant over $\mathcal{R}_z$. Let $\boldsymbol{R} = [\boldsymbol{a}\ \boldsymbol{b}\ \boldsymbol{c}\ \cdots\ \boldsymbol{a}_d^\perp]$, we have $\boldsymbol{h} = \boldsymbol{R}\boldsymbol{z}$. For any $\boldsymbol{z}^{(1)}$ and $\boldsymbol{z}^{(2)}$ with $\|\boldsymbol{z}^{(1)}\|_2 = \|\boldsymbol{z}^{(2)}\|_2$. We define $\boldsymbol{h}^{(1)} = \boldsymbol{R}\boldsymbol{z}^{(1)}$ and $\boldsymbol{h}^{(2)} = \boldsymbol{R}\boldsymbol{z}^{(2)}$. Since $\boldsymbol{x}$ is rotational invariant and $\|\boldsymbol{h}^{(1)}\|_2 = \|\boldsymbol{h}^{(2)}\|_2 = \|\boldsymbol{z}^{(1)}\|_2 = \|\boldsymbol{z}^{(2)}\|_2$, then we know $\boldsymbol{h}^{(1)}$ and $\boldsymbol{h}^{(2)}$ has the same distribution density. Then, $\boldsymbol{z}^{(1)}$ and $\boldsymbol{z}^{(2)}$ has the same distribution density as well. Therefore, $\boldsymbol{z}$ is rotational invariant over $\mathcal{R}_z$.

Then, we consider spherical coordinates with $z_1 = R\cos\phi_1, z_2 = R\sin\phi_1\sin\phi_2, z_3 = R\sin\phi_1\cos\phi_2$. Hence, we have

$$
I_4 = \int |\phi'(\boldsymbol{w}_{\ell,j_2}^\top \widetilde{\boldsymbol{h}}) - \phi'(\boldsymbol{w}_{\ell,j_2}^{\star\top} \widetilde{\boldsymbol{h}})| \cdot |R\cos\phi_1|^2 \cdot f_Z(R, \phi_1, \phi_2) \cdot R^2 \sin\phi_1 \cdot dR d\phi_1 d\phi_2.
\tag{122}
$$

Since $\boldsymbol{z}$ is rotational invariant, we have that

$$
f_Z(R, \phi_1, \phi_2) = f_Z(R).
\tag{123}
$$

Then, we have

$$
\begin{aligned}
I_4 &= \int |\phi'(\boldsymbol{w}_{\ell,j_2}^\top (\widetilde{\boldsymbol{h}}/R)) - \phi'(\boldsymbol{w}_{\ell,j_2}^{\star T}(\widetilde{\boldsymbol{h}}/R))| \cdot |R\cos\phi_1|^2 \cdot f_Z(R) R^2 \sin\phi_1 dR d\phi_1 d\phi_2 \\
&= \int_0^\infty R^4 f_z(R) dR \int_0^{\psi_1(R)} \int_0^{\psi_2(R)} |\cos\phi_1|^2 \cdot \sin\phi_1 \\
&\quad \cdot |\phi'(\boldsymbol{w}_{\ell,j_2}^\top (\widetilde{\boldsymbol{h}}/R)) - \phi'(\boldsymbol{w}_{\ell,j_2}^{\star T}(\widetilde{\boldsymbol{h}}/R))| d\phi_1 d\phi_2 \\
&\leq \int_0^\infty R^4 f_z(R) dR \int_0^\pi \int_0^{2\pi} \sin\phi_1 \cdot |\phi'(\boldsymbol{w}_{\ell,j_2}^\top \bar{\boldsymbol{x}}) - \phi'(\boldsymbol{w}_{\ell,j_2}^{\star T} \bar{\boldsymbol{x}})| d\phi_1 d\phi_2,
\end{aligned}
\tag{124}
$$

where the first equality holds because $\phi'(\boldsymbol{w}_{i,,j_2}^\top \boldsymbol{h})$ only depends on the direction of $\boldsymbol{h}$, and $\bar{\boldsymbol{x}} := \boldsymbol{h}/R = (\cos\phi_1, \sin\phi_1\sin\phi_2, \sin\phi_1\cos\phi_2)$ in the last inequality.

Because $\boldsymbol{z}$ belongs to the sub-Gaussian distribution, we have $F_z(R) \geq 1 - 2e^{-\frac{R^2}{\sigma^2}}$ for some constant $\sigma > 0$. Then, the integration of $R$ can be represented as

$$
\begin{aligned}
\int_0^\infty R^4 f_Z(R) dR &= \int_0^\infty R^4 d(1 - F_z(R)) \\
&\leq \int_0^\infty 4R^3 (1 - F_z(R)) dR \\
&\leq \int_0^\infty 8R^3 e^{-\frac{R^2}{\sigma^2}} dR \\
&\leq \frac{32}{\sqrt{2\pi}}\sigma \int_0^\infty R^2 e^{-\frac{R^2}{\sigma^2}} dR \\
&= 32\sigma^2 \int_0^\infty R^2 \frac{1}{\sqrt{2\pi\sigma^2}} e^{-\frac{R^2}{\sigma^2}} dR,
\end{aligned}
\tag{125}
$$

where the last inequality comes from the calculation that

$$
\begin{aligned}
\int_0^\infty 2R^2 e^{-\frac{R^2}{\sigma^2}} dR &= \sqrt{2\pi}\sigma^3, \\
\int_0^\infty 2R^3 e^{-\frac{R^2}{\sigma^2}} dR &= 4\sigma^4.
\end{aligned}
\tag{126}
$$

Then, we define $\widetilde{\boldsymbol{x}} \in \mathbb{R}^{K_\ell}$ belongs to Gaussian distribution as $\widetilde{\boldsymbol{x}} \sim \mathcal{N}(\mathbf{0}, \sigma^2 \boldsymbol{I})$. Therefore, we have

$$
\begin{aligned}
I_4 &\leq 32\sigma^2 \cdot \int_0^\infty R^2 \frac{1}{\sqrt{2\pi\sigma^2}} e^{-\frac{R^2}{\sigma^2}} dR \int_0^\pi \int_0^{2\pi} \sin\phi_1 \cdot |\phi'(\boldsymbol{w}_{\ell,j_2}^\top \bar{\boldsymbol{x}}) - \phi'(\boldsymbol{w}_{\ell,j_2}^{\star\top} \bar{\boldsymbol{x}})| d\phi_1 d\phi_2 \\
&= 32\sigma^2 \cdot \mathbb{E}_{z_1,z_2,z_3} |\phi'(\boldsymbol{w}_{\ell,j_2}^\top \widetilde{\boldsymbol{x}}) - \phi'(\boldsymbol{w}_{\ell,j_2}^{\star\top} \widetilde{\boldsymbol{x}})| \\
&\eqsim \mathbb{E}_{\widetilde{\boldsymbol{x}}} |\phi'(\boldsymbol{w}_{\ell,j_2}^\top \widetilde{\boldsymbol{x}}) - \phi'(\boldsymbol{w}_{\ell,j_2}^{\star T} \widetilde{\boldsymbol{x}})|,
\end{aligned}
\tag{127}
$$

where $\widetilde{\boldsymbol{x}}$ belongs to Gaussian distribution.

Therefore, the inequality bound over a sub-Gaussian distribution is bounded by the one over a Gaussian distribution. In the following contexts, we provide the upper bound of $\mathbb{E}_{\widetilde{\boldsymbol{x}}} |\phi'(\boldsymbol{w}_{\ell,j_2}^\top \widetilde{\boldsymbol{x}}) - \phi'(\boldsymbol{w}_{\ell,j_2}^{\star T} \widetilde{\boldsymbol{x}})|$.

Define a set $\mathcal{A}_1 = \{\boldsymbol{x} | (\boldsymbol{w}_{\ell,j_2}^{\star\top} \widetilde{\boldsymbol{x}})(\boldsymbol{w}_{\ell,j_2}^\top \widetilde{\boldsymbol{x}}) < 0\}$. If $\widetilde{\boldsymbol{x}} \in \mathcal{A}_1$, then $\boldsymbol{w}_{\ell,j_2}^{\star\top} \widetilde{\boldsymbol{x}}$ and $\boldsymbol{w}_{\ell,j_2}^\top \widetilde{\boldsymbol{x}}$ have different signs, which means the value of $\phi'(\boldsymbol{w}_{\ell,j_2}^\top \widetilde{\boldsymbol{x}})$ and $\phi'(\boldsymbol{w}_{\ell,j_2}^{\star\top} \widetilde{\boldsymbol{x}})$ are different. This is equivalent to say that

$$
|\phi'(\boldsymbol{w}_{\ell,j_2}^\top \widetilde{\boldsymbol{x}}) - \phi'(\boldsymbol{w}_{\ell,j_2}^{\star\top} \widetilde{\boldsymbol{x}})| = \left\{ \begin{array}{ll} 1, & \text{if } \widetilde{\boldsymbol{x}} \in \mathcal{A}_1 \\ 0, & \text{if } \widetilde{\boldsymbol{x}} \in \mathcal{A}_1^c \end{array} \right. .
\tag{128}
$$

Moreover, if $\widetilde{\boldsymbol{x}} \in \mathcal{A}_1$, then we have

$$
|\boldsymbol{w}_{\ell,j_2}^{\star T} \widetilde{\boldsymbol{x}}| \leq |\boldsymbol{w}_{\ell,j_2}^{\star T} \widetilde{\boldsymbol{x}} - \boldsymbol{w}_{\ell,j_2}^\top \widetilde{\boldsymbol{x}}| \leq \|\boldsymbol{w}_{\ell,j_2}^\star - \boldsymbol{w}_{\ell,j_2}\|_2 \cdot \|\widetilde{\boldsymbol{x}}\|_2.
\tag{129}
$$

Let us define a set $\mathcal{A}_2$ such that

$$
\begin{aligned}
\mathcal{A}_2 &= \left\{ \widetilde{\boldsymbol{x}} \Big| \frac{|\boldsymbol{w}_{\ell,j_2}^{\star T} \widetilde{\boldsymbol{x}}|}{\|\boldsymbol{w}_{\ell,j_2}^\star\|_2 \|\widetilde{\boldsymbol{x}}\|_2} \leq \frac{\|\boldsymbol{w}_{\ell,j_2}^\star - \boldsymbol{w}_{\ell,j_2}\|_2}{\|\boldsymbol{w}_{\ell,j_2}^\star\|_2} \right\} \\
&= \left\{ \theta_{\widetilde{\boldsymbol{x}}, \boldsymbol{w}_{\ell,j_2}^\star} \Big| |\cos\theta_{\widetilde{\boldsymbol{x}}, \boldsymbol{w}_{\ell,j_2}^\star}| \leq \frac{\|\boldsymbol{w}_{\ell,j_2}^\star - \boldsymbol{w}_{\ell,j_2}\|_2}{\|\boldsymbol{w}_{\ell,j_2}^\star\|_2} \right\}.
\end{aligned}
\tag{130}
$$

Hence, we have that

$$
\begin{aligned}
\mathbb{E}_{\widetilde{\boldsymbol{x}}} |\phi'(\boldsymbol{w}_{\ell,j_2}^\top \widetilde{\boldsymbol{x}}) - \phi'(\boldsymbol{w}_{\ell,j_2}^{\star T} \widetilde{\boldsymbol{x}})|^2 &= \mathbb{E}_{\widetilde{\boldsymbol{x}}} |\phi'(\boldsymbol{w}_{\ell,j_2}^\top \widetilde{\boldsymbol{x}}) - \phi'(\boldsymbol{w}_{\ell,j_2}^{\star T} \widetilde{\boldsymbol{x}})| \\
&= \text{Prob}(\widetilde{\boldsymbol{x}} \in \mathcal{A}_1) \\
&\leq \text{Prob}(\widetilde{\boldsymbol{x}} \in \mathcal{A}_2).
\end{aligned}
\tag{131}
$$

Since $\widetilde{\boldsymbol{x}} \sim \mathcal{N}(\mathbf{0}, \|\boldsymbol{a}\|_2^2 \boldsymbol{I})$, $\theta_{\widetilde{\boldsymbol{x}}, \boldsymbol{w}_{\ell,j_2}^\star}$ belongs to the uniform distribution on $[-\pi, \pi]$, we have

$$
\begin{aligned}
\text{Prob}(\widetilde{\boldsymbol{x}} \in \mathcal{A}_2) = \frac{\pi - \arccos \frac{\|\boldsymbol{w}_{\ell,j_2}^\star - \boldsymbol{w}_{\ell,j_2}\|_2}{\|\boldsymbol{w}_{\ell,j_2}^\star\|_2}}{\pi} &\leq \frac{1}{\pi} \tan(\pi - \arccos \frac{\|\boldsymbol{w}_{\ell,j_2}^\star - \boldsymbol{w}_{\ell,j_2}\|_2}{\|\boldsymbol{w}_{\ell,j_2}^\star\|_2}) \\
&= \frac{1}{\pi} \cot(\arccos \frac{\|\boldsymbol{w}_{\ell,j_2}^\star - \boldsymbol{w}_{\ell,j_2}\|_2}{\|\boldsymbol{w}_{\ell,j_2}^\star\|_2}) \\
&\leq \frac{2}{\pi} \frac{\|\boldsymbol{w}_{\ell,j_2}^\star - \boldsymbol{w}_{\ell,j_2}\|_2}{\|\boldsymbol{w}_{\ell,j_2}^\star\|_2} \\
&\leq \|\boldsymbol{W}_\ell^\star - \boldsymbol{W}_\ell\|_2
\end{aligned}
\tag{132}
$$

Hence, (124) and (132) suggest that

$$
\begin{aligned}
I_4 &\lesssim \|\boldsymbol{W}_i - \boldsymbol{W}_i^\star\|_2 \cdot \|\boldsymbol{a}\|_2^2, \\
\text{and} \qquad \|\boldsymbol{P}_1\|_2 &\leq \|\boldsymbol{W} - \boldsymbol{W}^\star\|_2 + I_4 \lesssim \|\boldsymbol{W} - \boldsymbol{W}^\star\|_2,
\end{aligned}
\tag{133}
$$

The same bound that is shown in (133) holds for $\boldsymbol{P}_2$ as well.

Therefore, we have

$$
\begin{aligned}
\|\nabla_\ell^2 f(\boldsymbol{W}^\star) - \nabla_\ell^2 f(\boldsymbol{W})\|_2 &= \max_{\|\boldsymbol{\alpha}\|_2 \le 1} \left| \boldsymbol{\alpha}^\top \left( \nabla_\ell^2 f(\boldsymbol{W}^\star) - \nabla_\ell^2 f(\boldsymbol{W}) \right) \boldsymbol{\alpha} \right| \\
&\le \frac{1}{K^2} \sum_{j_1=1}^{K} \sum_{j_2=1}^{K} \|\boldsymbol{P}_1 + \boldsymbol{P}_2\|_2 \cdot \|\boldsymbol{\alpha}_{j_1}\|_2 \cdot \|\boldsymbol{\alpha}_{j_2}\|_2 \\
&\lesssim \frac{1}{K^2} \cdot \sum_{j_1=1}^{K} \sum_{j_2=1}^{K} \|\boldsymbol{W} - \boldsymbol{W}^\star\|_2 \cdot \|\boldsymbol{\alpha}_{j_1}\|_2 \|\boldsymbol{\alpha}_{j_2}\|_2 \qquad (134) \\
&\lesssim \frac{1}{K^2} \cdot \sum_{j_1=1}^{K} \sum_{j_2=1}^{K} \|\boldsymbol{W} - \boldsymbol{W}^\star\|_2 \cdot \left( \frac{\|\boldsymbol{\alpha}_{j_1}\|_2^2 + \|\boldsymbol{\alpha}_{j_2}\|_2^2}{2} \right) \\
&\lesssim \frac{1}{K} \cdot \|\boldsymbol{W}^\star - \boldsymbol{W}\|_2,
\end{aligned}
$$

where $\boldsymbol{\alpha} \in \mathbb{R}^{Kd}$ and $\boldsymbol{\alpha}_j \in \mathbb{R}^{K_\ell}$ with $\boldsymbol{\alpha} = [\boldsymbol{\alpha}_1^\top, \boldsymbol{\alpha}_2^\top, \cdots, \boldsymbol{\alpha}_K^\top]^\top$. $\qquad\square$

## E.2  Proof of Lemma 7

We aim to prove that $\int_{\mathcal{R}} \left( \sum_{j=1}^{K} \boldsymbol{\alpha}^\top \boldsymbol{h} \phi'(\boldsymbol{w}_{\ell,j}^\top \boldsymbol{h}) \right)^2 p_H(\boldsymbol{h}) \cdot d\boldsymbol{h}$ is strictly greater than zero for any $\boldsymbol{\alpha}$. Therefore, the $\rho$ in (2) is strictly greater than zero. The proof is inspired by Theorem 3.1 in [22]. It is obviously that $\left( \sum_{j=1}^{K} \boldsymbol{\alpha}^\top \boldsymbol{h} \phi'(\boldsymbol{w}_{\ell,j}^\top \boldsymbol{h}) \right)^2$ is greater or equal to zero. Given $\left( \sum_{j=1}^{K} \boldsymbol{\alpha}^\top \boldsymbol{h} \phi'(\boldsymbol{w}_{\ell,j}^\top \boldsymbol{h}) \right)^2$ is continuous, we only need to show that $\alpha$ such that $\sum_{j=1}^{K} \boldsymbol{\alpha}^\top \boldsymbol{h} \phi'(\boldsymbol{w}_{\ell,j}^\top \boldsymbol{h}) \ne 0$ for any $\alpha$, namely, $\{\boldsymbol{h}\phi'(\boldsymbol{w}_{\ell,j}^\top \boldsymbol{h})\}_{j=1}^{K}$ are linear independent. Compared with Theorem 3.1 in [22], we need to address two challenges: (1) the neuron weights $\boldsymbol{w}$ is the random variable in [22] while the input $\boldsymbol{h}$ is the random variable in this paper and (2) the random variable belongs to Gaussian distribution in [22] while the random variable belongs to sub-Gaussian distribution in this paper.

*Proof of Lemma 7.* Let $\mathcal{H}$ be a Hilbert space on $\mathbb{R}^{K_\ell}$, and the inner product of $\mathcal{H}$ is defined as

$$
\langle f, g \rangle = \int_{\mathcal{R}} f(\boldsymbol{h})^\top g(\boldsymbol{h}) f_H(\boldsymbol{h}) \cdot d\boldsymbol{h}, \quad \forall f, g \in \mathcal{H}, \qquad (135)
$$

where the Lebesgue measure of $\mathcal{R}$ over $\mathbb{R}^{K_\ell}$ is non-zero. Instead of directly proving $\int_{\mathcal{R}} \left( \sum_{k=1}^{K} \boldsymbol{\alpha}^\top \boldsymbol{h} \phi'(\boldsymbol{w}_k^\top \boldsymbol{h}) \right)^2 f_H(\boldsymbol{h}) \cdot d\boldsymbol{h} > 0$ for any $\boldsymbol{\alpha}$, we note that it is sufficient to prove that $\{\boldsymbol{h}\phi'(\boldsymbol{w}_k^\top \boldsymbol{h})\}_{k \in [K]}$ are linear independent over the Hilbert space $\mathcal{H}$. Namely, if $\{\boldsymbol{h}\phi'(\boldsymbol{w}_k^\top \boldsymbol{h})\}_{k \in [K]}$ are linear independent, we have

$$
\boldsymbol{\alpha}^\top \boldsymbol{h} \phi'(\boldsymbol{w}_k^\top \boldsymbol{h}) \ne 0 \quad \textit{almost everywhere}. \qquad (136)
$$

Therefore, we can know that $\int_{\mathcal{R}} \left( \sum_{j=1}^{K} \boldsymbol{\alpha}^\top \boldsymbol{h} \phi'(\boldsymbol{w}_{\ell,j}^\top \boldsymbol{h}) \right)^2 p_H(\boldsymbol{h}) \cdot d\boldsymbol{h}$ is strictly greater than zero.

Next, we provide the whole proof for that $\{x\phi'(\boldsymbol{w}_k^\top \boldsymbol{h})\}_{k \in [K]}$ are linear independent over the Hilbert space $\mathcal{H}$.

We define a group of functions $\{\psi_j(\boldsymbol{h})\}_{j=1}^{K}$, where $\psi_j(\boldsymbol{h}) = \boldsymbol{h}\phi'(\boldsymbol{w}_j^\top \boldsymbol{h})$. From the assumption in Lemma 7, we can justify that $\mathbb{E}_{\boldsymbol{h} \sim \mathcal{D}} |\psi_j(\boldsymbol{h})|^2 \le \mathbb{E}_{\boldsymbol{h} \sim \mathcal{D}} |\boldsymbol{h}|^2 < \infty$.

Let $\mathcal{X}_i = \{\boldsymbol{h} \mid \boldsymbol{w}_i^\top \boldsymbol{h} = 0\}$ for any $i \in [K]$. For any fixed $k$, we can justify that $\mathcal{X}_k$ cannot be covered by other sets $\{\mathcal{X}_k\}_{j \ne k}$ as long as $\boldsymbol{w}_k$ does not parallel to any other weights $\boldsymbol{w}_j$ with $j \ne k$. Namely, $\mathcal{X}_k \not\subset \cup_{j \ne k} \mathcal{X}_j$. The idea of proving the claim above is that the intersection of $\mathcal{X}_j$ and $\mathcal{X}_k$ is only a hyperplane in $\mathcal{X}_k$. The union of finite many hyperplanes is not even a measurable space and thus cannot cover the original space. Formally, we provide the formal proof for this claim as follows.

Let $\lambda$ be the Lebesgue measure on $\mathcal{X}_k$, then $\lambda(\mathcal{X}_k) > 0$. When $\boldsymbol{w}_j$ does not parallel to $\boldsymbol{w}_k$, $\mathcal{X}_k \cap \mathcal{X}_j$ is only a hyperplane in $\mathcal{X}_k$ for $j \neq k$. Hence, we have $\lambda(\mathcal{X}_j \cap \mathcal{X}_k) = 0$. Next, we have

$$\lambda\big(\mathcal{X}_k \cap (\cup_{j \neq k} \mathcal{X}_k)\big) \leq \sum_{j \neq k} \lambda(\mathcal{X}_k \cap \mathcal{X}_j) = 0. \tag{137}$$

Therefore, we have

$$\lambda\big(\mathcal{X}_k / (\cup_{j \neq k} \mathcal{X}_k)\big) = \lambda(\mathcal{X}_k) - \lambda\big(\mathcal{X}_k \cap (\cup_{j \neq k} \mathcal{X}_k)\big) = \lambda(\mathcal{X}_k) > 0. \tag{138}$$

Therefore, we have $\mathcal{X}_k / (\cup_{j \neq k} \mathcal{X}_j)$ is not empty, which means that $\mathcal{X}_k \not\subset \cup_{j \neq k} \mathcal{X}_j$.

Next, Since $\mathcal{X}_k / (\cup_{j \neq k} \mathcal{X}_j)$ is not an empty set, there exists a point $\boldsymbol{z}_k \in \mathcal{X}_k / (\cup_{j \neq k} \mathcal{X}_j)$ and $r_0 > 0$ such that

$$\mathcal{B}(\boldsymbol{z}_k, r) \cap \mathcal{D}_j = \emptyset \quad with \quad \forall r \leq r_0 \ and \ j \neq k, \tag{139}$$

where $\mathcal{B}(\boldsymbol{z}_k, r)$ stands for a ball centered at $\boldsymbol{z}_k$ with a radius of $r$. Then, we divide $\mathcal{B}(\boldsymbol{z}_k, r)$ into two disjoint subsets such that

$$\begin{aligned} \mathcal{B}_r^+ &= \mathcal{B}(\boldsymbol{z}_k, r) \cap \{\boldsymbol{h} \mid \boldsymbol{w}_k^\top \boldsymbol{h} > 0\}, \\ \mathcal{B}_r^- &= \mathcal{B}(\boldsymbol{z}_k, r) \cap \{\boldsymbol{h} \mid \boldsymbol{w}_k^\top \boldsymbol{h} < 0\}. \end{aligned} \tag{140}$$

Because $\boldsymbol{z}_k$ is a boundary point of $\{\boldsymbol{h} | \boldsymbol{w}_k^\top \boldsymbol{h} = 0\}$, both $\mathcal{B}_r^+$ and $\mathcal{B}_r^-$ are non-empty.

Note that $\psi_j(\boldsymbol{h})$ is continuous at any point except for the ones in $\mathcal{X}_j$. Then, for any $j \neq k$, we know that $\phi_j(\boldsymbol{w}_k^\top \boldsymbol{h})$ is continuous at point $\boldsymbol{z}_k$ since $\boldsymbol{z}_k \notin \mathcal{X}_j$. Hence, it is easy to verify that

$$\lim_{r \to 0_+} \frac{1}{\lambda(\mathcal{B}_r^+)} \int_{\mathcal{B}_r^+} \psi_k(\boldsymbol{h}) d\boldsymbol{h} = \lim_{r \to 0_-} \frac{1}{\lambda(\mathcal{B}_r^-)} \int_{\mathcal{B}_r^+} \psi_k(\boldsymbol{h}) d\boldsymbol{h} = \psi_k(\boldsymbol{z}_k). \tag{141}$$

While for $\psi_k$, we know that $\psi_k(\boldsymbol{h}) \equiv 0$ for $\boldsymbol{h} \in \mathcal{B}_r^-$, (ii) $\psi_k(\boldsymbol{h}) = \boldsymbol{h}$ for $\boldsymbol{h} \in \mathcal{B}_r^+$. Hence, it is easy to verify that

$$\begin{aligned} \lim_{r \to 0_+} \frac{1}{\lambda(\mathcal{B}_r^+)} \int_{\mathcal{B}_r^+} \psi_k(\boldsymbol{h}) d\boldsymbol{h} &= \boldsymbol{z}_k \\ \lim_{r \to 0_-} \frac{1}{\lambda(\mathcal{B}_r^-)} \int_{\mathcal{B}_r^+} \psi_k(\boldsymbol{h}) d\boldsymbol{h} &= 0. \end{aligned} \tag{142}$$

Now let us proof that $\{\psi_j\}_{j=1}^K$ are linear independent by contradiction. Suppose $\{\psi_j\}_{j=1}^K$ are linear dependent, we have

$$\sum_{j=1}^K \alpha_j \psi_j(\boldsymbol{h}) \equiv 0, \quad \forall \boldsymbol{h}. \tag{143}$$

Then, we have

$$\begin{aligned} \lim_{r \to 0_+} \frac{1}{\lambda(\mathcal{B}_r^+)} \int_{\mathcal{B}_r^+} \sum_{j=1}^K \alpha_j \psi_j(\boldsymbol{h}) d\boldsymbol{h} &= 0 \\ \lim_{r \to 0_+} \frac{1}{\lambda(\mathcal{B}_r^-)} \int_{\mathcal{B}_r^+} \sum_{j=1}^K \alpha_j \psi_j(\boldsymbol{h}) d\boldsymbol{h} &= 0 \end{aligned} \tag{144}$$

Then, we have

$$\begin{aligned} 0 &= \lim_{r \to 0_+} \frac{1}{\lambda(\mathcal{B}_r^+)} \int_{\mathcal{B}_r^+} \sum_{j=1}^K \alpha_j \psi_j(\boldsymbol{h}) d\boldsymbol{h} - \lim_{r \to 0_+} \frac{1}{\lambda(\mathcal{B}_r^-)} \int_{\mathcal{B}_r^+} \sum_{j=1}^K \alpha_j \psi_j(\boldsymbol{h}) d\boldsymbol{h} \\ &= \alpha_k \boldsymbol{z}_k \end{aligned} \tag{145}$$

where the last equality comes from (141) and (142).

Note that $\boldsymbol{z}_k$ cannot be $\boldsymbol{0}$ because $\boldsymbol{z}_k \notin \mathcal{X}_j$. Therefore, we have $\alpha_k = 0$. Similarly to (145), we can obtain that $\alpha_j = 0$ by define $\boldsymbol{z}_j$ following the definition of $\boldsymbol{z}_k$ for any $j \in [K]$. Then, we know that (143) holds if and only if $\boldsymbol{\alpha} = \boldsymbol{0}$, which contradicts the assumption that $\{\psi_j\}_{j=1}^K$ are linear dependent.

In conclusion, we know that $\{\psi_j\}_{j=1}^K$ are linear independent, and $\int_{\mathcal{R}} \big( \sum_{j=1}^K \boldsymbol{\alpha}^\top \boldsymbol{h} \phi'(\boldsymbol{w}_{\ell,j}^\top \boldsymbol{h}) \big)^2 p_H(\boldsymbol{h}) \cdot d\boldsymbol{h}$ is strictly greater than zero. $\qquad \square$

### E.3 Proof of Lemma 8

*Proof of Lemma 8.* From the definition of (39), we have

$$
\begin{aligned}
&\|\boldsymbol{h}^{(\ell)}(\boldsymbol{W}) - \boldsymbol{h}^{(\ell)}(\boldsymbol{W}^\star)\|_2 \\
=&\|\phi\big(\boldsymbol{W}_{\ell-1}^\top \boldsymbol{h}^{(\ell-1)}(\boldsymbol{W})\big) - \phi\big(\boldsymbol{W}_{\ell-1}^{\star\top} \boldsymbol{h}^{(\ell-1)}(\boldsymbol{W}^\star)\big)\|_2 \\
=&\|\phi\big(\boldsymbol{W}_{\ell-1}^\top \boldsymbol{h}^{(\ell-1)}(\boldsymbol{W})\big) - \phi\big(\boldsymbol{W}_{\ell-1}^{\star\top} \boldsymbol{h}^{(\ell-1)}(\boldsymbol{W})\big) \\
&+ \phi\big(\boldsymbol{W}_{\ell-1}^{\star\top} \boldsymbol{h}^{(\ell-1)}(\boldsymbol{W})\big) - \phi\big(\boldsymbol{W}_{\ell-1}^{\star\top} \boldsymbol{h}^{(\ell-1)}(\boldsymbol{W}^\star)\big)\|_2 \\
\leq&\|\phi\big(\boldsymbol{W}_{\ell-1}^\top \boldsymbol{h}^{(\ell-1)}(\boldsymbol{W})\big) - \phi\big(\boldsymbol{W}_{\ell-1}^{\star\top} \boldsymbol{h}^{(\ell-1)}(\boldsymbol{W})\big)\|_2 \\
&+ \|\phi\big(\boldsymbol{W}_{\ell-1}^{\star\top} \boldsymbol{h}^{(\ell-1)}(\boldsymbol{W})\big) - \phi\big(\boldsymbol{W}_{\ell-1}^{\star\top} \boldsymbol{h}^{(\ell-1)}(\boldsymbol{W}^\star)\big)\|_2 \\
\leq&\|\boldsymbol{W}_{\ell-1} - \boldsymbol{W}_{\ell-1}^\star\|_2 \cdot \|\boldsymbol{h}^{(\ell-1)}(\boldsymbol{W})\|_2 + \|\boldsymbol{h}^{(\ell-1)}(\boldsymbol{W}) - \boldsymbol{h}^{(\ell-1)}(\boldsymbol{W}^\star)\|_2.
\end{aligned}
\tag{146}
$$

With the assumption in the Lemma 8 such that $\boldsymbol{W}$ is close enough to $\boldsymbol{W}^\star$, we have

$$
\|\boldsymbol{W}_i\|_2 \leq \|\boldsymbol{W}_i^\star\|_2 + \|\boldsymbol{W}_i - \boldsymbol{W}_i^\star\|_2 \lesssim 1.
\tag{147}
$$

Therefore, we have

$$
\|\boldsymbol{h}^{(i)}(\boldsymbol{W})\|_2 \leq \|\boldsymbol{W}_i\|_2 \cdots \|\boldsymbol{W}_1\|_2 \cdot \|\boldsymbol{x}\|_2 \lesssim \|\boldsymbol{x}\|_2.
\tag{148}
$$

Then, we have

$$
\begin{aligned}
&\|\boldsymbol{h}^{(\ell)}(\boldsymbol{W}) - \boldsymbol{h}^{(\ell)}(\boldsymbol{W}^\star)\|_2 \\
\leq&\|\boldsymbol{W}_{\ell-1} - \boldsymbol{W}_{\ell-1}^\star\|_2 \cdot \|\boldsymbol{x}\|_2 + \|\boldsymbol{h}^{(\ell-1)}(\boldsymbol{W}) - \boldsymbol{h}^{(\ell-1)}(\boldsymbol{W}^\star)\|_2 \\
\leq&\sum_{i=1}^{\ell-1} \|\boldsymbol{W}_i - \boldsymbol{W}_i^\star\|_2 \cdot \|\boldsymbol{x}\|_2 + \|\boldsymbol{h}^{(1)}(\boldsymbol{W}) - \boldsymbol{h}^{(1)}(\boldsymbol{W}^\star)\|_2 \\
=&\sum_{i=1}^{\ell-1} \|\boldsymbol{W}_i - \boldsymbol{W}_i^\star\|_2 \cdot \|\boldsymbol{x}\|_2 + \|\boldsymbol{x} - \boldsymbol{x}\|_2 \\
=&\sum_{i=1}^{\ell-1} \|\boldsymbol{W}_i - \boldsymbol{W}_i^\star\|_2 \cdot \|\boldsymbol{h}^{(i-1)}(\boldsymbol{W})\|_2 \\
\leq&\|\boldsymbol{W} - \boldsymbol{W}^\star\|_2 \cdot \|\boldsymbol{x}\|_2,
\end{aligned}
\tag{149}
$$

which completes the proof. □

## F  Additional experiments

In this section, we provide numerical justification that our theoretical findings are aligned with DDQN through the Atari Breakout game The neural network follows the same architecture as the one used in Section 5. The algorithm terminates if the average score over the recent 100 episodes does not improve or the algorithm reaches the maximum episode set as 200, which is around $4 \times 10^5$ training steps. The testing score is calculated based on a similar setup as the training process by fixing the maximum memory size $N$ as 2000 and greedy policy, i.e., $\varepsilon = 0$. Each point in the plot is averaged over 10 experiments with an error bar representing the standard deviation.

**Estimation errors with respect to the sample complexity** $N$**.** We follow the setup in Section 5 to use the expected cumulative reward as the estimation error of the learned model to the optimal Q-value function. The $\varepsilon_t$ in $\varepsilon$-greedy policy decreases geometrically from 1 to 0.01. We vary the number of samples in the replay buffer from 3000 to 10000. Figure 4 shows that the test error is almost linear in $1/\sqrt{N}$, which is consistent with our characterization in (20). In addition, experiments with a large $N$ have a shorter error bar indicating a more stable learning performance with a large sample complexity as shown in (12).

**Convergence with different selections of** $\varepsilon$**.** Figure 5 illustrates the convergence rate when $\varepsilon_t$ in the $\varepsilon$-greedy policy changes. For each point, $\varepsilon_0$ is selected as the value in the x-axis, and we decrease $\varepsilon_t$ geometrically as the iteration $t$ increases. Each point is averaged over 10 independent trials. We can see that the convergence rate is a linear function of $c_\varepsilon$, matching our findings in (19).

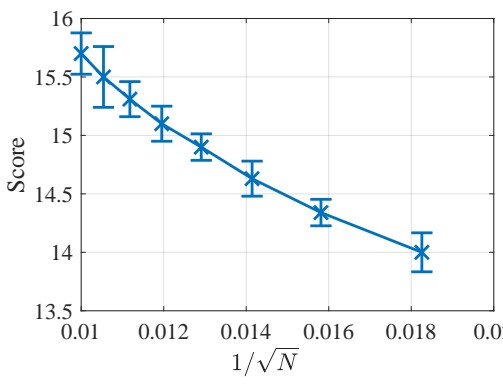

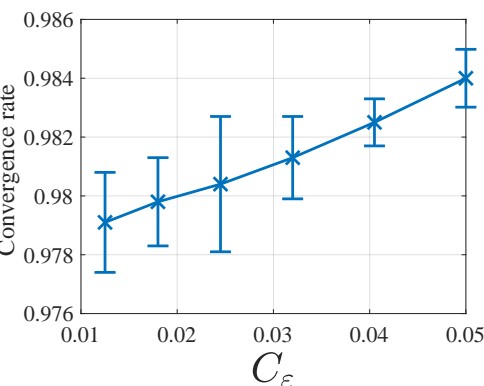

Figure 4: Test error in scores against the number of samples.

Figure 5: The convergence rate against the value of $c_\varepsilon$.

## G   Extension to non i.i.d. samples

**Assumption 3.** *At any fixed outer iteration $t$, the behavior policy $\pi_t$ and transition kernel $\mathcal{P}_t$ satisfy*

$$\sup_{\boldsymbol{s}\in\mathcal{S}} d_{TV}\big(\mathbb{P}(\boldsymbol{s}_\tau \in \cdot) \mid \boldsymbol{s}_0 = \boldsymbol{s}), \mathcal{P}_t\big) \leq \lambda\nu^\tau, \quad \forall\, \tau \geq 0 \tag{150}$$

*for some constant $\lambda > 0$ and $\nu \in (0,1)$, where $d_{TV}$ denotes the total-variation distance between the probability measures.*

Assumption 3 assumes the Markov chain $\{\boldsymbol{s}_t\}$ induced by the behavior policy, i.e., $\varepsilon_t$-greedy policy at $t$-th outer loop, is uniformly ergodic with the corresponding invariant measure $\mathcal{P}_t$. Compared with i.i.d. cases, we need to handle an additional error term when bounding the distance between the $g_t$ and $\nabla f$ as shown in (91). Therefore, the upper bound in Lemma 3 changes, which suggests an additional term in the final bound.

We present the major theoretical findings for non-i.i.d. samples in Theorem 2. The major proofs in this context follow similar steps to the proof of Theorem 1, with slight changes in the error bound between the sequences $g_t$ and $\nabla f$. In this section, we omit the details of the proof for Theorem 2 but provide the proof for Lemma 3 under the assumptions outlined in Assumption 2 to simplify the presentation.

**Theorem 2** (Convergence for non-i.i.d. case). *Suppose Assumption 1 and (143) hold, the buffer size $N$ satisfies (13). Let us define $C_{\max}$ be a constant that is larger than $C_t$ for $1 \leq t \leq T$ and $C_d = |\mathcal{A}| \cdot (1 + \log_\nu \lambda^{-1} + \frac{1}{1-\nu})$, when $\varepsilon_t$ satisfy*

$$\varepsilon_t = \frac{c_\varepsilon \cdot \Theta(\sqrt{N}) \cdot e_t}{(1 - C_{\max}) \cdot C_d \cdot R_{\max}} - \frac{C_{\max}}{1 - C_{\max}} \tag{151}$$

*for a fixed constant $c_\varepsilon \in (0, (1-\gamma)^2]$, and the initialization satisfies*

$$\|\boldsymbol{W}^{(0,0)} - \boldsymbol{W}^\star\|_F \leq \mathcal{O}\Big(1 - \frac{1 - c_\varepsilon}{\Theta(\sqrt{N})}\Big) \cdot \frac{\rho \cdot \|\boldsymbol{W}^\star\|_F}{K}. \tag{152}$$

*Then, with the high probability of at least $1 - T \cdot q^{-d}$, we have*

*(C1) The learned weights decay geometrically with*

$$\|\boldsymbol{W}^{(t+1,0)} - \boldsymbol{W}^\star\|_F \leq \big(\gamma + c_\varepsilon \cdot (1-\gamma)\big) \cdot \|\boldsymbol{W}^{(t,0)} - \boldsymbol{W}^\star\|_F + \frac{(2+\gamma)R_{\max}\tau^\star}{(1-\gamma)\Theta(N)}, \tag{153}$$

*(C2) the returned model $Q(\boldsymbol{W}^{(T,0)})$ exhibits an estimation error as*

$$\sup_{(s,a)} \big|Q(\boldsymbol{W}^{(T,0)}) - Q^\star\big| \leq \frac{C_{\max} \cdot C_d \cdot R_{\max}}{(1-\gamma)^2 \cdot \Theta(\sqrt{N \cdot T})} + \frac{(2+\gamma)R_{\max}\tau^\star}{(1-\gamma)\Theta(N \cdot T)}, \tag{154}$$

*where $\tau^\star = \min\{t \mid \lambda\nu^t \leq 1/(N \cdot T)\}$.*

*Proof of Lemma 3 under Assumption 2.* Recall that in (91), we have

$$
g_t(\boldsymbol{w}_{\ell,k}; \boldsymbol{W}) - \frac{\partial f}{\partial \boldsymbol{w}_{\ell,k}}(\boldsymbol{W})
$$

$$
= \left[ \frac{1}{N} \sum_{n=1}^{N} \Big( Q(\boldsymbol{W}; \boldsymbol{s}_n, a_n) - Q(\boldsymbol{W}^\star; \boldsymbol{s}_n, a_n) \Big) \cdot \frac{\partial Q(\boldsymbol{W}; \boldsymbol{s}_n, a_n)}{\partial \boldsymbol{w}_{\ell,k}} \right.
$$

$$
\left. - \mathbb{E}_{(\boldsymbol{s},a)\sim\mathcal{D}_t} \Big( Q(\boldsymbol{W}; \boldsymbol{s}, a) - Q(\boldsymbol{W}^\star; \boldsymbol{s}, a) \Big) \cdot \frac{\partial Q(\boldsymbol{W}; \boldsymbol{s}, a)}{\partial \boldsymbol{w}_{\ell,k}} \right]
$$

$$
+ \left[ \frac{1}{N} \sum_{n=1}^{N} \gamma \cdot \Big( \max_a Q(\boldsymbol{s}_n, a; \boldsymbol{W}^\star) - \max_a Q(\boldsymbol{s}_n, a; \boldsymbol{W}^{(t,0)}) \Big) \cdot \frac{\partial Q(\boldsymbol{W}; \boldsymbol{s}_n, a_n)}{\partial \boldsymbol{w}_{\ell,k}} \right] \tag{155}
$$

$$
+ \mathbb{E}_{(\boldsymbol{s},a)\sim\mu_t} g_t(\boldsymbol{w}_{\ell,k}; \boldsymbol{W}) - \frac{\partial f}{\partial \boldsymbol{w}_{\ell,k}}(\boldsymbol{W})
$$

$$
+ \mathbb{E}_{(\boldsymbol{s},a)\sim\mathcal{D}_t,\mathcal{P}} \big[ g_t(\boldsymbol{w}_{\ell,k}; \boldsymbol{W}) - \mathbb{E}_{(\boldsymbol{s},a)\sim\mu_t,\mathcal{P}} g_t(\boldsymbol{w}_{\ell,k}; \boldsymbol{W}) \big]
$$

$$
:= I_1 + I_2 + I_3 + I_4.
$$

**Bound of $I_1$ and $I_2$.** Compared with (91), the upper bound for $I_1$ and $I_2$ is the same as those shown in (103) and (109), respectively.

**Bound of $I_3$.** Following (111), the upper bound of $I_3$ can be characterized as

$$
\|I_3\|_2 \leq \frac{R_{\max}}{1-\gamma} \cdot \left[ (1-\varepsilon) \cdot \left| \int_{(\boldsymbol{s},a)} \int_{\boldsymbol{s}'} \big( \mu^\star(d\boldsymbol{s}, da)\mathcal{P}(d\boldsymbol{s}'|\boldsymbol{s},a) - \mu_{t,1}(d\boldsymbol{s}, da)\mathcal{P}(d\boldsymbol{s}'|\boldsymbol{s},a) \big) \right| \right.
$$

$$
\left. + \varepsilon \cdot \left| \int_{(\boldsymbol{s},a)} \int_{\boldsymbol{s}'} \big( \mu^\star(d\boldsymbol{s}, da)\mathcal{P}(d\boldsymbol{s}'|\boldsymbol{s},a) - \mu_{t,2}(d\boldsymbol{s}, da)\mathcal{P}(d\boldsymbol{s}'|\boldsymbol{s},a) \big) \right| \right]. \tag{156}
$$

and

$$
\left| \int_{(\boldsymbol{s},a)} \int_{\boldsymbol{s}'} \big( \mu^\star(d\boldsymbol{s}, da)\mathcal{P}(d\boldsymbol{s}'|\boldsymbol{s},a) - \mu_{t,1}(d\boldsymbol{s}, da)\mathcal{P}(d\boldsymbol{s}'|\boldsymbol{s},a) \big) \right|
$$

$$
= \left| \int_{(\boldsymbol{s},a)} \int_{\boldsymbol{s}'} \big( \mathcal{P}^\star(d\boldsymbol{s})\pi^\star(da|\boldsymbol{s})\mathcal{P}(d\boldsymbol{s}'|\boldsymbol{s},a) - \mathcal{P}_{t,1}(d\boldsymbol{s})\pi_{t,1}(da|d\boldsymbol{s})\mathcal{P}(d\boldsymbol{s}'|\boldsymbol{s},a) \big) \right| \tag{157}
$$

$$
\leq \left| \int_{(\boldsymbol{s},a)} \int_{\boldsymbol{s}'} \big( \mathcal{P}^\star(d\boldsymbol{s}) - \mathcal{P}_{t,1}(d\boldsymbol{s}) \big) \pi^\star(da|\boldsymbol{s})\mathcal{P}(d\boldsymbol{s}'|\boldsymbol{s},a) \right|
$$

$$
+ \left| \int_{(\boldsymbol{s},a)} \int_{\boldsymbol{s}'} \mathcal{P}_{t,1}(d\boldsymbol{s}) \big( \pi_{t,1}(da|d\boldsymbol{s}) - \pi^\star(da|d\boldsymbol{s}) \big) \mathcal{P}(d\boldsymbol{s}'|\boldsymbol{s},a) \right|.
$$

From Theorem 3.1 in [49], we know that

$$
\left| \int_{(\boldsymbol{s},a)} \big( \mathcal{P}^\star(d\boldsymbol{s}) - \mathcal{P}_{t,1}(d\boldsymbol{s}) \big) \right| \leq |\mathcal{A}|(\log_\nu \lambda^{-1} + \frac{1}{1-\nu})C_t \tag{158}
$$

$$
\text{and} \qquad \big\| \pi_{t,1}(da|d\boldsymbol{s}) - \pi^\star(da|d\boldsymbol{s}) \big\| \leq C_t.
$$

Therefore, the bound of $I_3$ can be found as

$$
\|I_3\|_2 \leq \frac{R_{\max}}{1-\gamma} \cdot |\mathcal{A}| \cdot \big( (1-\varepsilon)C_t + \varepsilon \cdot C_t \big) \cdot (1 + \log_\nu \lambda^{-1} + \frac{1}{1-\nu})
$$

$$
= C_d \cdot \big( C_t + (1-C_t)\varepsilon \big) \cdot \frac{R_{\max}}{1-\gamma}, \tag{159}
$$

where $C_d = |\mathcal{A}| \cdot (1 + \log_\nu \lambda^{-1} + \frac{1}{1-\nu})$.

**Bound of $I_4$.** $I_4$ is the bias of the data because the data $(\boldsymbol{s}, a)$ at iteration $t$ depends on the neural network parameters $\boldsymbol{W}$. Let us define $\bar{g}_t$ as

$$
\bar{g}_t(\boldsymbol{w}_{\ell,k}; \boldsymbol{W}) = \mathbb{E}_{\mu_t,\mathcal{P}} \, g_t(\boldsymbol{w}_{\ell,k}; \boldsymbol{W}) \tag{160}
$$

and

$$
\Delta_t = g_t(\boldsymbol{w}_{\ell,k}; \boldsymbol{W}) - \bar{g}_t(\boldsymbol{w}_{\ell,k}; \boldsymbol{W}). \tag{161}
$$

It is easy to verify that

$$\|g_t(\boldsymbol{w}_{\ell,k};\boldsymbol{W}) - g_t(\tilde{\boldsymbol{w}}_{\ell,k};\widetilde{\boldsymbol{W}})\| \leq (1+\gamma) \cdot \|\boldsymbol{W} - \widetilde{\boldsymbol{W}}\|,$$
$$\|\bar{g}_t(\boldsymbol{w}_{\ell,k};\boldsymbol{W}) - \bar{g}_t(\tilde{\boldsymbol{w}}_{\ell,k};\widetilde{\boldsymbol{W}})\| \leq (1+\gamma) \cdot \|\boldsymbol{W} - \widetilde{\boldsymbol{W}}\|, \tag{162}$$
$$and \qquad \|g_t\| \lesssim \frac{R_{\max}}{1-\gamma}.$$

Then, we have

$$\Delta_t(\boldsymbol{W}) - \Delta_t(\widetilde{\boldsymbol{W}}) \lesssim (1+\gamma) \cdot \|\boldsymbol{W} - \widetilde{\boldsymbol{W}}\|_2. \tag{163}$$

Therefore, we have

$$\Delta_t(\boldsymbol{W}^{(t,0)}) \leq \Delta_t(\boldsymbol{W}^{(t-\tau,0)}) + \frac{1+\gamma}{1-\gamma} \cdot R_{\max} \cdot \sum_{i=t-\tau}^{t-1} \eta_i. \tag{164}$$

Then, we need to bound $\delta_t(\boldsymbol{W}^{(t-\tau,0)})$.

Let us define the observed tuple $O_t(\boldsymbol{s}, a, s')$ as the collection of the state, action, and the next state at the $t$-th outer loop. Note that

$$\boldsymbol{W}^{(t-\tau,0)} \longrightarrow \boldsymbol{s}_{t-\tau} \longrightarrow \boldsymbol{s}_t \longrightarrow O_t \tag{165}$$

forms a Markov chain introduced by the policy $\pi_{t-\tau}$.

Let $\widetilde{\boldsymbol{W}}^{(t-\tau,0)}$ and $\widetilde{O}_t$ be independently drawn from the marginal distributions of $\widetilde{\boldsymbol{W}}^{(t-\tau,0)}$ and $O_t$, respectively.

With Lemma 9 in [4], we have

$$\mathbb{E}\,\Delta_t(\boldsymbol{W}^{(t-\tau,0)}, O_t) - \mathbb{E}\,\Delta_t(\widetilde{\boldsymbol{W}}^{(t-\tau,0)}, \widetilde{O}_t) \lesssim 2\sup_{\boldsymbol{w},O}|\Delta_t(\boldsymbol{W},O)| \cdot \lambda \cdot \nu^\tau. \tag{166}$$

By definition, we have $\mathbb{E}\,\Delta_t(\widetilde{\boldsymbol{W}}^{(t-\tau,0)}, \widetilde{O}_t) = 0$ and

$$|\Delta_t(\boldsymbol{W},O)| \leq \frac{2\,R_{\max}}{1-\gamma}. \tag{167}$$

Therefore, we have

$$\mathbb{E}\Delta_t(\boldsymbol{W}^{(t,0)}) \leq \mathbb{E}\Delta_t(\boldsymbol{W}^{(t-\tau,0)}) + \frac{1+\gamma}{1-\gamma} \cdot R_{\max} \cdot \sum_{i=t-\tau}^{t-1} \eta_i$$
$$\leq \frac{R_{\max}}{1-\gamma}\Big(\lambda \cdot \nu^\tau + (1+\gamma) \cdot \tau \cdot \eta_{t-\tau}\Big), \tag{168}$$

where the last inequality comes from the fact that the step size $\eta_t$ is non-increasing.

Choose $\tau^\star = \min\big\{t = 0,1,2,\cdots \mid \lambda\nu^\tau \leq \eta_T\big\}$. When $t \leq \tau^\star$, we choose $\tau = t$ and have

$$\mathbb{E}\Delta_t(\boldsymbol{W}^{(t,0)}) \leq \frac{R_{\max}}{1-\gamma} \cdot \tau^\star \cdot \eta_0. \tag{169}$$

When $t > \tau^\star$, we can choose $\tau = \tau^\star$ and obtain

$$\mathbb{E}\Delta_t(\boldsymbol{W}^{(t,0)}) \leq \frac{R_{\max}}{1-\gamma} \cdot (1+\gamma)\tau^\star \cdot \eta_{t-\tau^\star}. \tag{170}$$

Combining (169) and (170), we have

$$|I_4| \leq \frac{R_{\max}}{1-\gamma} \cdot (1+\gamma)\tau^\star \cdot \eta_{\max\{0,t-\tau^\star\}}, \tag{171}$$

where $\tau^\star = \min\{t \mid \lambda\nu^t \leq \eta_T\}$. $\qquad\square$

