$$Q(W; s, a) = \frac{\mathbf{1}^\top}{K}\phi(w_{L,k}^\top h^{(L)}) = \frac{\mathbf{1}^\top}{K}\phi\big(W_L^\top \phi(W_{L-1}^\top h^{(L-1)})\big), \tag{41}$$

where $w_{\ell,k}$ denotes the $k$-th neuron weights in the $\ell$-th layer. Then, we define a group of functions $\mathcal{J}_\ell(W) \in \mathbb{R}^n \longrightarrow \mathbb{R}^K$ such that

$$\mathcal{J}_\ell(W) = \begin{cases} \big[\mathbf{1}^\top \phi'(W_L^\top h^{(L)})W_L^\top \cdot \phi'(W_{L-1}^\top h^{(L-1)})W_{L-1}^\top \cdots \phi'(W_{\ell+1}^\top h^{(\ell+1)})W_{\ell+1}^\top\big]^\top & if \quad \ell > 1 \\ \mathbf{1} & if \quad \ell = 1. \end{cases} \tag{42}$$

Then, the gradient of $Q$ can be represented as

$$\frac{\partial Q}{\partial w_{\ell,k}}(W) = \frac{1}{K}\mathcal{J}_{\ell,k}(W)\phi'\big(w_{\ell,k}^\top h^{(\ell)}(W)\big)h^{(\ell)}(

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

}_{\ell,j_2}^{\star\top}\boldsymbol{a} + z_2\boldsymbol{w}_{\ell,j_2}^{\star\top}\boldsymbol{b} + z_3\boldsymbol{w}_{\ell,j_2}^{\star\top}\boldsymbol{c} + \cdots + z_d\boldsymbol{w}_{\ell,j_2}^{\star\top}\boldsymbol{a}_d^\perp$$

$$= z_1\boldsymbol{w}_{\ell,j_2}^{\star\top}\boldsymbol{a} + z_2\boldsymbol{w}_{\ell,j_2}^{\star\top}\boldsymbol{b} + z_3\boldsymbol{w}_{\ell,j_2}^{\star\top}\boldsymbol{c} + 0 \tag{119}$$

$$= \boldsymbol{w}_{\ell,j_2}^{\star\top}(z_1\boldsymbol{a} + z_2\boldsymbol{b} + z_3\boldsymbol{c})$$

$$:= \boldsymbol{w}_{\ell,j_2}^{\star\top}\widetilde{\boldsymbol{h}}.$$

where $\widetilde{\boldsymbol{h}} = z_1\boldsymbol{a} + z_2\boldsymbol{b} + z_3\boldsymbol{c}$. Similar to (119), we have $\boldsymbol{w}_{\ell,j_2}^\top\boldsymbol{h} = \boldsymbol{w}_{\ell,j_2}^\top\widetilde{\boldsymbol{h}}$ and $\boldsymbol{a}^\top\boldsymbol{h} = \boldsymbol{a}^\top\widetilde{\boldsymbol{h}}$.

Then, we define $I_4$ as

$$I_4 := \mathbb{E}_{\boldsymbol{h}}\Big| \big(\phi'(\boldsymbol{w}_{\ell,j_2}^{\star\top}\boldsymbol{h}) - \phi'(\boldsymbol{w}_{\ell,j_2}^\top\boldsymbol{h})\big)\cdot(\boldsymbol{a}^\top\boldsymbol{h})\Big|$$

$$= \int_{\mathcal{R}_{\boldsymbol{h}}} |\phi'(\boldsymbol{w}_{\ell,j_2}^\top\boldsymbol{h}) - \phi'(\boldsymbol{w}_{\ell,j_2}^{\star T}\boldsymbol{h})| \cdot |\boldsymbol{a}^\top\boldsymbol{h}|^2 \cdot f_H(\boldsymbol{h})d\boldsymbol{h} \tag{120}$$

$$= \int_{\mathcal{R}_{\boldsymbol{z}}} |\phi'(\boldsymbol{w}_{\ell,j_2}^\top\boldsymbol{h}) - \phi'(\boldsymbol{w}_{\ell,j_2}^{\star T}\boldsymbol{h})| \cdot |\boldsymbol{a}^\top\boldsymbol{h}|^2 \cdot f_Z(\boldsymbol{z}) \cdot |\boldsymbol{J}_{\boldsymbol{h}}(\boldsymbol{z})|d\boldsymbol{z}$$

where $|J_h(z)|$ is the determinant of the Jacobian matrix $\frac{\partial h}{\partial z}$. Since $z$ is a representation of $h$ based on an orthogonal and normalized basis, we have $|J_h(z)| = 1$. According to (119), $I_4$ can be rewritten as

$$I_4 = \int_{\mathcal{R}_z} |\phi'(w_{\ell,j_2}^\top \widetilde{h}) - \phi'(w_{\ell,j_2}^{\star T} \widetilde{h})| \cdot |a^\top \widetilde{h}|^2 \cdot f_Z(z) dz$$
$$= \int_{\mathcal{R}_z} |\phi'(w_{\ell,j_2}^\top \widetilde{h}) - \phi'(w_{\ell,j_2}^{\star T} \widetilde{h})| \cdot |a^\top \widetilde{h}|^2 \cdot f_Z(z_1, z_2, z_3) dz_1 dz_2 dz_3$$

(121)

where in the last equality we abuse $f_Z(z_1, z_2, z_3)$ to represent the probability density function of $(z_1, z_2, z_3)$ defined in region $\mathcal{R}_z$.

Next, we show that $z$ is rotational invariant over $\mathcal{R}_z$. Let $R = [a\ b\ c\ \cdots\ a_d^\perp]$, we have $h = Rz$. For any $z^{(1)}$ and $z^{(2)}$ with $\|z^{(1)}\|_2 = \|z^{(2)}\|_2$. We define $h^{(1)} = Rz^{(1)}$ and $h^{(2)} = Rz^{(2)}$. Since $x$ is rotational invariant and $\|h^{(1)}\|_2 = \|h^{(2)}\|_2 = \|z^{(1)}\|_2 = \|z^{(2)}\|_2$, then we know $h^{(1)}$ and $h^{(2)}$ has the same distribution density. Then, $z^{(1)}$ and $z^{(2)}$ has the same distribution density as well. Therefore, $z$ is rotational invariant over $\mathcal{R}_z$.

Then, we consider spherical coordinates with $z_1 = R cos\phi_1, z_2 = R sin\phi_1 sin\phi_2, z_3 = R sin\phi_1 cos\phi_2$. Hence, we have

$$I_4 = \int |\phi'(w_{\ell,j_2}^\top \widetilde{h}) - \phi'(w_{\ell,j_2}^{\star\top} \widetilde{h})| \cdot |R\cos\phi_1|^2 \cdot f_Z(R, \phi_1, \phi_2) \cdot R^2 \sin\phi_1 \cdot dRd\phi_1 d\phi_2. \quad (122)$$

Since $z$ is rotational invariant, we have that

$$f_Z(R, \phi_1, \phi_2) = f_Z(R). \quad (123)$$

Then, we have

$$I_4 = \int |\phi'(w_{\ell,j_2}^\top(\widetilde{h}/R)) - \phi'(w_{\ell,j_2}^{\star T}(\widetilde{h}/R))| \cdot |R\cos\phi_1|^2 \cdot f_Z(R) R^2 \sin\phi_1 dRd\phi_1 d\phi_2$$
$$= \int_0^\infty R^4 f_z(R) dR \int_0^{\psi_1(R)} \int_0^{\psi_2(R)} |\cos\phi_1|^2 \cdot \sin\phi_1$$
$$\cdot |\phi'(w_{\ell,j_2}^\top(\widetilde{h}/R)) - \phi'(w_{\ell,j_2}^{\star T}(\widetilde{h}/R))| d\phi_1 d\phi_2$$
$$\leq \int_0^\infty R^4 f_z(R) dR \int_0^\pi \int_0^{2\pi} \sin\phi_1 \cdot |\phi'(w_{\ell,j_2}^\top \bar{x}) - \phi'(w_{\ell,j_2}^{\star T} \bar{x})| d\phi_1 d\phi_2,$$

(124)

where the first equality holds because $\phi'(w_{i,,j_2}^\top h)$ only depends on the direction of $h$, and $\bar{x} :=