# OpenReview forum: "On the Convergence and Sample Complexity Analysis of Deep Q-Networks with $\epsilon$-Greedy Exploration"
_NeurIPS.cc/2023/Conference — NeurIPS 2023 poster_

### Official Review · Reviewer_fZ7E · 2023-07-05

**Soundness:** 2 fair
**Presentation:** 3 good
**Contribution:** 2 fair
**Rating:** 5
**Confidence:** 3

**Summary:**

This paper studies DQN (one of the most popular RL algorithms in the literature) with $\epsilon$-greedy behavior policy. Specifically, the authors provide the convergence and sample complexity guarantees and numerically verify the algorithm's performance.

**Strengths:**

This paper is well-written, and the appendix seems of high quality.

**Weaknesses:**

Major Comments:

(1) Corollary 1: It is unclear to me how to choose $\epsilon_t$ based on this result as $e_t$ is precisely the error of the algorithm, which is unknown to the agent.

(2) Theorem 1: The authors refer to $W^{T,0}$ as the convergent point of the algorithm, which is confusing because $W^{T,0}$ depends on the iteration number. In addition, it is unclear to me why Theorem 1 implies the almost sure convergence (or any other kind of asymptotic convergence) of the weights.

In Eq. (19), I saw $t$ on the LHS. and $T$ on the RHS. Is this a typo? The claim of geometric decay based on Eq. (19) is not entirely convincing because $\epsilon_t$ is not a constant but an implicit function of $t$. This also impacts the claim in Corollary 2. From my understanding, Corollary 2 is derived solely based on Eq. (20) because the transient term in Eq. (19) decays at a geometric rate, thereby does not contribute to any polynomial sample complexity.

(3) Corollary 3: The assumption in Eq. (22) is not completely rigorous. In general, one should not make any assumptions about the trajectory of the very algorithm that is being studied.

**Questions:**

When using linear function approximation, Q-learning can be divergent [1,2] even if the target network is used as in this paper. In addition, there are several impossibility results in the literature [3,4] stating that unless certain assumptions are made (such as Bellman completeness, or both $V^*$ and $Q^*$ belong to the function class), the sample complexity can be exponential. While this paper uses neural network approximation, it is similar to linear function approximation when there is only one layer (the difference is only about the activation function). Then, (i) why is the algorithm naturally stable? and (ii) why does this algorithm achieve polynomial sample complexity? It is important to investigate the fundamental reason behind these questions to provide a better understanding of the behavior of Q-learning with function approximation.

[1] Baird, L. (1995). Residual algorithms: Reinforcement learning with function approximation. In Machine Learning Proceedings 1995 (pp. 30-37). Morgan Kaufmann.

[2] Chen, Z., Clarke, J. P., & Maguluri, S. T. (2022). Target Network and Truncation Overcome The Deadly Triad in $ Q $-Learning. arXiv preprint arXiv:2203.02628.

[3] Zanette, A. (2021, July). Exponential lower bounds for batch reinforcement learning: Batch rl can be exponentially harder than online rl. In International Conference on Machine Learning (pp. 12287-12297). PMLR.

[4] Wang, R., Foster, D. P., & Kakade, S. M. (2020). What are the statistical limits of offline RL with linear function approximation?. arXiv preprint arXiv:2010.11895.

**Limitations:**

The comments in the weakness section (especially comment (3)) should be the theoretical limitations of this work.

---

> ### Author Rebuttal · Authors · 2023-08-09
>
> *Q1:  Regarding Corollary 1.*
>
> **A1**: Thanks for your insightful question! $e_t$ in Corollary 1 can be replaced by its upper bound, which is independent of $\epsilon_t$. See GR for details.
>
> *Q2: Regarding “W^T depends on the iteration number.” and “almost sure convergence of the weights”.*
>
> **A2**:  Thank you for bringing up the confusing parts, and we appreciate your feedback. In the following contents, we would like to clarify the key points regarding the revised Theorem 1, which can be found in the GR.
>
> **Firstly, there exists an upper bound for estimation error of the learned model weights that is independent of the iteration number.** We did not make that explicit in the original Theorem 1, and we revised Theorem 1 to reflect that. See (iii) in GR for details.
>
> **Secondly, we have revised (20) in Theorem 1 to indicate the convergence of the weights.** In our revised Theorem 1,  $C_\max$ is bounded above for any arbitrarily large $T$. In addition, as the left-hand side of (20) goes to zero as $T$ goes to infinity, indicating the convergence of our proposed algorithm. We apologize for the unclear description of the upper bound of the learned weights in our manuscript because of the dependence between the upper limit and iteration number.
>
> **Thirdly, we would like to clarify that Theorem 1 holds with a high probability of at least $1 - T\cdot q^{-d}$, which we regret was missing in the submitted manuscript.** We regret the oversight of points 2 and 3 that may have led to your confusion about “convergence of the weights.”
>
> *Q3: Typo in (19)?*
>
> **A3**: We apologize for the typo. It should be $t$ in both the LHS and RHS of (19), and please check GR for the updated Theorem 1.
>
> *Q4: Regarding Corollary 2.*
>
> **A4**:  Thank you for your insights. Although $\epsilon_t$ is not a constant, the convergence rate is $\gamma + c_\epsilon(1-\gamma)$ from (19), and $c_\epsilon$ is a constant, which is pre-determined and remains unchanged over time $t$, see the line 245 after (17).
>
> *Q5: Regarding Corollary 3.*
>
> **A5**: Thank you for your thoughtful feedback. We believe the raised concern may stem from a misunderstanding of the relationship between Theorem 1 and Corollary 3. We would like to clarify that Theorem 1 contributes to our primary conclusion and does not rely on the assumption presented in (22). We apologize for the improper presentation of Theorem 1 in the original draft, and we anticipate that the revised version of Theorem 1 in GR will help alleviate any such misunderstanding.
>
> To elaborate, our significant theoretical findings, outlined in Section 4.1,  are directly derived from Theorem 1 without any reliance on the assumption in (22). Namely, (i) the theoretical characterization of $\epsilon_t$ for convergence, (ii) the convergence analysis of DQN with bounded estimation error, and (iii) sample complexity for achieving a desired estimation error of the optimal Q-value function. These three major contributions do not rely on the assumption in (22).  Moreover,  we introduced Corollary 3 as an extension to our primary theorem, showcasing a tighter bound, which diminishes to zero, for the estimation error of the Q-value function under a stronger assumption (22), compared to the general constant upper bound in (20).
>
> We are acutely aware of your concerns about the rigor of Eq. (22), and such a concern is carefully addressed in our original draft by not incorporating the aforementioned assumption into our core theoretical findings.
>
> *Q6: “When using linear function approximation, Q-learning can be divergent [1,2].”*
>
> **A6**: Thank you for providing the relevant papers. We agree with your observation that Q-learning can be divergent without making certain assumptions. In [2], the authors demonstrate that using a target network together with truncation is sufficient to stabilize Q-learning. The truncation effectively restricts the learned function's weights from deviating too far away from a certain point, which could potentially be a ground truth or some other reference point. A similar idea, as seen in [84], involves restricting the weights within a local region to ensure a stable algorithm.
>
> Although our work does not restrict the update of the weights, the weights naturally move in a relatively small region in our proposed algorithm. To see this, we characterize a local convex region in our work. Using a proper step size and a good initialization, we prove that the learned weights in Algorithm 1 are guaranteed to remain within this local convex region. This result aligns with the idea presented in [2] for stabilizing Q-learning. We will include the relevant papers in our work and provide a clear connection with existing research.
>
> *Q7: “The sample complexity can be exponential in [3,4]."*
>
> **A7**: Thank you for providing the relevant papers. We want to clarify the distinction between our manuscript and [3, 4], as [3, 4] both address offline RL (or batch RL) scenarios where the agent cannot access the environment during the algorithm's execution. However, this manuscript focuses on an online setting. In online settings, during each outer loop of the algorithm, the agent can access the environment and actively collect data based on the updated policy. This fundamental difference in the setup allows the proposed approach to leverage real-time data and adapt the policy during the learning process, which can lead to more efficient and adaptive learning compared to offline RL methods. The ability to access the Markov transition matrix and actively collect new data enables the algorithm to make informed decisions based on up-to-date information, potentially leading to more effective policy learning. For example, polynomial sample complexity has been achieved for online Q-learning with function approximation in [19, 22, 31, 74] (citation number in submitted version). We will add a new paragraph in the related work to clarify the sample complexity differences between offline and online RL.

---

> > ### Comment · Reviewer_fZ7E · 2023-08-14
> > **Acknowledgement of the Rebuttal**
> >
> > Thank the authors for the detailed response.
> >
> > A2: I believe the authors misunderstood my comments. I am suggesting not to call $W^T$ the convergent point in Theorem 1 (probably call it the output) because $W^*$ is the convergent point.
> >
> > A5: I understand that Theorem 1 is the main result. However, for theoretical work, all results should be rigorous. In the current draft, Corollary 3 relies on an assumption that depends on the algorithm trajectory, which is not mathematically rigorous. In the next version, it might be a better idea to either remove Corollary 3 or revise the assumption in Eq. (22) so that it does not depend on the algorithm trajectory.
> >
> > I feel this paper has the potential of contributing to the RL community. However, since the main theoretical results need to be significantly revised (which is quite unfortunate), the current draft in my opinion is not ready for publication. Therefore, I will retain my score.

---

> > > ### Author Response · Authors · 2023-08-14
> > > **Repsonse to Reviewer fZ7E and additonal clarification of Rebuttal**
> > >
> > > We thank the reviewer very much for the follow-up response. While we understand the reviewer's concerns and preference to maintain the original score, we would like to provide further clarifications on Corollary 3 and Theorem 1 to prevent any miscommunications.
> > >
> > > Regarding the reviewer’s concern about major theoretical results being significantly revised, we believe it is a misunderstanding. We improved the statements in Theorem 1 in the revision to make them clearer. It is important to note that we only improved the result presentation but we did not change any proofs of Theorem 1 except for replacing $e_t$ with its upper bound in (62) to obtain an improved presentation in Theorem 1. Specifically, we only replace $C_t$ and $e_t$ in (15)-(17) of the original paper with its upper bound in (19-20) in our paper, because the upper bounds are time-independent. Our original intent in the original submission was to present the least restrictive conditions on  $\epsilon_t$ in (17). Following the insightful feedback from the reviewers, we replaced (15)-(17) with the upper bounds that are already developed in our paper in (19) and (20). The resulting condition in the revised Theorem 1 is slightly more restrictive but independent on $t$.  Therefore, we just changed the presentation and did not make any major technical changes.
> > >
> > > We agree with the reviewer that it is not ideal to make assumptions about the algorithm trajectory. The reason we keep Corollary 3 rather than remove it is that (22) reflects the policy improvement of the behavior policy via greedy policy and is expected to hold but is difficult to prove with the current techniques.  In fact,  **assumption (22) has been previously proposed in the existing study [Zou et al.2019] (see equation. (2)), published in NeurIPS**, and leveraged to obtain the major theoretical results, which are Theorems 1 and 2 in [Zou et al.2019]. Therefore, we exploit the a similar assumption in Corollary 3 in our paper.
> > >
> > > Finally, **we would like to highlight the major contributions of this paper outlined in the original paper**, containing (i) the theoretical characterization of $\epsilon_t$ for convergence, (ii) the convergence analysis of DQN with bounded estimation error, and (iii) sample complexity for achieving a desired estimation error of the optimal Q-value function, **do not rely on equation (22).** As elaborated in A5, while drawn upon (22) from prior research, we are aware of the concerns about the rigor of such an assumption. Consequently, we intend to avoid presenting the results derived from (22) as our major theoretical results. Instead, we illustrate the results built upon (22) as Corollary 3, an augmentative extension of Theorem 1.
> > >
> > > [Zou et al.2019] Zou, Shaofeng, Tengyu Xu, and Yingbin Liang. "Finite-sample analysis for sarsa with linear function approximation." Advances in neural information processing systems 32 (2019).

---

> > > > ### Comment · Reviewer_fZ7E · 2023-08-17
> > > >
> > > > Thank the authors for the detailed response. I agree with the authors that while the statement of the main theorem is significantly revised, the adjustment is mainly for presentation with the core idea remaining the same.
> > > >
> > > > Regarding Corollary 3, I would like to point out that Eq. (2) in [Zou et al. 2019] is fundamentally different from Eq. (22) in this work. [Zou et al. 2019] did not specify using $\epsilon$-greedy policy. Their Eq. (2) is an assumption on the policy update function, which can be easily satisfied using softmax update. Most importantly, their Eq. (2) is not an assumption on the algorithm trajectory.

---

> > > > > ### Author Response · Authors · 2023-08-18
> > > > > **Response to Official Comment by Reviewer fZ7E**
> > > > >
> > > > > We sincerely appreciate the follow-up response and valid feedback from the reviewer! We are pleased to hear that the reviewer agrees with our perspective that the adjustments made to Theorem 1 is mainly for presentation, with the core idea remaining unchanged.
> > > > >
> > > > >
> > > > > Additionally, we would like to elaborate more details on addressing your concerns about Corollary 3 using Equation.(2) from [Zou et al.2019]. We apology for our  confusing statement in the previous response.
> > > > >
> > > > >
> > > > > Firstly, [Zou et al., 2019] considers an $\epsilon$-greedy policy, as mentioned in the first paragraph of Section 3.1, where it is stated: “$\Gamma$ is a policy improvement operator, e.g., greedy, $\epsilon$-greedy, softmax, and mellowmax”.
> > > > >
> > > > >
> > > > > Secondly, we agree with the reviewer that (2) in [Zou et al., 2019] is different from our Equation (22). We apologize for the imprecise statement before. What we should have said is that (2) in [Zou et al., 2019] leads to (22) of this paper, i.e., **(2) in [Zou et al., 2019]  is a sufficient condition for (22).** To see this, Equation (22) can be interpreted as the difference between the integration of $\pi_t$ over the state distribution at iteration $t$ and the integration of $\pi^\star$ over the state distribution following the optimal policy. Note that (2) in [Zou et al., 2019] holds universally across the entire state space and is not time-dependent. Given that policy $\pi_t$ is Lipschitz continuous concerning $W_t$ (or $\theta_t$ in [Zou et al., 2019]), it can be verified that the state distribution under different policies also exhibits Lipschitz continuity with respect to weights. Such a result is also proved and used in Lemmas 3 & 4 [Zou et al., 2019]. Consequently, the overall difference between the integrations also maintains Lipschitz continuity, mirroring Equation (22) in our paper.
> > > > >
> > > > > Thirdly, based on the previous point,  Corollary 3 still holds if we replace assumption (22) in this paper with (2) in [Zou et al., 2019]. The reason we assume (22) rather than (2) in [Zou et al., 2019] is because (i) equation (2) is a stronger condition, although it is trajectory independent, and (ii) our proof specifically requires equation (22) rather than the stronger version (2), and (iii) equation (22) is more in line with the notation of $C_t$ for the sake of simplification. However, we completely understand and agree with the reviewer’s concern that making an assumption on the trajectory leads to doubt whether it can hold or not,  so we will add this footnote when assuming (22) in the paper,
> > > > >
> > > > >  “Although equation (22) depends on the algorithm’s trajectory, it can be derived from Equation (2) in [Zou et al., 2019], which holds universally across the entire state space and is not time-dependent.”

---

> > > > > > ### Comment · Reviewer_fZ7E · 2023-08-18
> > > > > >
> > > > > > Thank the authors for the prompt response. I carefully looked into [Zou et al 2019] Eq. (2) and believe it can never be satisfied when using either greedy or $ \epsilon $-greedy policy. To illustrate, since the argmax operator (regardless of the tie-breaking rule) is NOT a continuous operator, it clearly cannot be Lipschitz continuous. It is straightforward to construct an example where the $Q$-function parameter only needs to change very slightly to completely change its maximum component, resulting in two different induced policies. For example, consider the tabular setting (which is a special case of linear function approximation) with a single state and two actions. For any positive $\alpha$, when the $Q$-function changes from $(1-\alpha,1)$ to $(1+\alpha,1)$, the $\epsilon$-greedy policy changes from $(\epsilon/2,1-\epsilon/2)$ to $(1-\epsilon/2,\epsilon/2)$. The difference between these two policies is significant and is independent of $\alpha$. Is my reasoning here correct? Nevertheless, since the softmax operator is Lipschitz continuous, [Zou et al 2019] Eq. (2) holds when using softmax policies.
> > > > > >
> > > > > > Coming back to this work, if my reasoning in the previous paragraph is correct, since this paper explicitly assumed using $\epsilon$-greedy policies, Eq. (2) in [Zou et al 2019] does not hold, and clearly cannot imply Eq. (22) in this paper.

---

> > > > > > > ### Author Response · Authors · 2023-08-19
> > > > > > > **Response to Official Comments by Reviewer fZ7E**
> > > > > > >
> > > > > > > We appreciate the reviewer for the detailed and prompt response. We thank the reviewer for pointing out (2) in [Zou et al.2019] does not hold for greedy or epsilon-greedy policy. We previously focused on the connection between (2) and (22) but did not think carefully about the contradiction between (2) and greedy policy. After careful thought, we agree that (2) does not hold for greedy or epsilon-greedy policy except for some simple and trivial cases, but our arguments still hold with some changes.
> > > > > > >
> > > > > > > We want to clarify that the following condition ($\star$), which is a weaker version of (2) in [Zou et al. 2019], leads to (22) in our paper. In other words, **($\star$) is a sufficient condition for (22).**  In fact, we were previously thinking about ($\star$) when we refer to (2). We thank the reviewer for helping us clean our thoughts. Specifically, ($\star$) we are considering is
> > > > > > >
> > > > > > > $$|\pi\_W(s|a) - \pi^\star(s|a)|\le C||W-W^\star||_2,   \textrm{ for any } W \textrm{ near the ground truth } W^\star \textrm{as shown in (12)} ~~~~~~~~~~~~    (\star)$$
> > > > > > >
> > > > > > > ($\star$) is weaker than (2) in [Zou et al. 2019] because (2) in [Zou et al.2019] requires $|\pi\_\{W\_1\}(s|a) - \pi\_\{W\_2\}(s|a)\le C ||W\_1-W\_2||\_2$  must hold for all $W_1$ and $W_2$, while ($\star$) only requires $W_2$ to be the ground truth and $W_1$ to be some weights near the ground truth.
> > > > > > >
> > > > > > > While (2) in [Zou et al.2019] does not hold with epsilon-greedy,   **($\star$) can hold with some Q learning with epsilon-greedy.** As an example, if we make a slight adjustment to your example by setting the ground truth $Q^\star$ to (1, 2), changing the action would necessitate a Q value change of at least 1 from the optimal. This makes it evident that the equation above holds for values of $W$ near $W^\star$, e.g., ($\star$) holds.
> > > > > > >
> > > > > > > Next, we elaborate on the reason that ($\star$) results in (22). Given that a policy $\pi_t$ differs from $\pi^\star$ as a function of $||W_t-W^\star||$, it can be verified that the state distribution under these two policies is also a function of $||W_t-W^\star||$. Recall that (22) can be interpreted as the difference between integrating $\pi_t$ over the state distribution at iteration $t$ and integrating $\pi^\star$ over the state distribution following the optimal policy. Given that both the $|\pi_t-\pi^\star|$ and the data distribution shift between $\pi_t$ and $\pi^\star$ are all related with $||W_t-W^\star||$, we can derive that $|C_t - C_\star|$ is a function of $||W_1 - W_2||$.
> > > > > > >
> > > > > > > As you could see from the above argument, we only need $|\pi_W(s|a) - \pi^\star(s|a)|\le C||W-W^\star||_2$ to hold for any $W$ near the ground truth $W^\star$ as ($\star$) states. We do not need a strong requirement as (2) in [Zou et al. 2019] that must be held for all $W$. In fact, in our previous response on how (2) in [Zou et al. 2019] leads to (22), our argument is based on showing step (i) (2) in [Zou et al. 2019] leads to ($\star$), and step (ii) ($\star$) leads to (22). After cleaning our thoughts, we realized that step (i) is unnecessary.
> > > > > > >
> > > > > > > Therefore, we will change the footnote when assuming (22) in the paper to the following,
> > > > > > > “Although Equation (22) depends on the algorithm’s trajectory, it can be easily derived from
> > > > > > > $$|\pi_W(s|a) - \pi^\star(s|a)|\le C||W-W^\star||_2  (\star)  \textrm{ for any } W \textrm{ near the ground truth } W^\star \textrm{as shown in (12)} $$  Note that ($\star$) is not time-dependent. Moreover, ($\star$) is a weaker condition than Equation (2) in [Zou et al., 2019], which holds universally across the entire space and model parameter space.”

---

> > > > > > > > ### Comment · Reviewer_fZ7E · 2023-08-20
> > > > > > > >
> > > > > > > > Thank the authors for the detailed explanation. Most of my concerns are addressed so I have increased my score.
> > > > > > > >
> > > > > > > > I suggest imposing Eq. (*) directly as the assumption and adding a remark saying that it can be relaxed to the algorithm-trajectory-dependent assumption. In that way, this paper is more mathematically rigorous.

---

> > > > > > > > > ### Author Response · Authors · 2023-08-20
> > > > > > > > > **Thank you**
> > > > > > > > >
> > > > > > > > > We are encouraged to hear that most of the concerns have been addressed, and we thank the reviewer for taking the time to provide the review and engage in the discussion! Your active participation in the discussion has played a crucial role in clarifying our thoughts and elevating the overall rigor of our paper. We truly appreciate your suggestion regarding the inclusion of ($\star$) in the main text to enhance the mathematical rigor of Corollary 3, as well as a remark to further clarify the connection between ($\star$) and (22) based on our recent discussion!

---

### Official Review · Reviewer_5oZJ · 2023-07-06

**Soundness:** 3 good
**Presentation:** 3 good
**Contribution:** 3 good
**Rating:** 7
**Confidence:** 4

**Summary:**

The paper studies non-asymptotic behavior of deep Q-network where the inner loop uses momentum and the behavior policy is time-varying epsilon greedy policy. With nice choice of initialization and exploring strategy $\epsilon_t$, the authors prove that the DQN algorithm enjoys $1/\sqrt{N}$ sample complexity.

The analysis of inner loop iteration follows the spirit of bounding the error between the gradient of population risk function and its estimation, with a novel lower bound on the Hessian of population risk function.

**Strengths:**

1. The paper manages to derive sample complexity deep Q-network under epsilon-greedy policy, which has been difficult due to time-varying policy. The authors tackle this problem by defining $C_t$, a quantity that measures the difference of behavior policy and optimal policy. The theoretical contribution seems to be meaningful.

2. Moreover, the technique to lower bound the Hessian matrix, levereging on a series of functions that are linearly independent over a Hilbert space,  seems to be novel.

**Weaknesses:**

1. The exploration strategy $\epsilon_t$ requires prior knowledge of optimal policy, which is not possible. Moreover, the upper and lower bound for $\epsilon_t$ is not really intuitive.

2. The work assumes that mini-batch is sampled from time-varying stationary distribution. The transition matrix induced by behavior policy at every time step is assumed to be have stationary distribution.

3. The step-size depends on the number of outer loop iteartions, $T$.

**Questions:**

1. Please give reference or more explanation for the definition of local convex region in equation (11). It looks like norm, but what does it have to do with convexity? Why do we need to care about its lower bound?

2. To prove local convexity of $f(W)$ around $W^*$, the lower bound on the hessian of $f(W)$, Lemma 6 plays a key role, but I think the details are quite omitted. The coordinate transform to bound $I_4$ from line 856~861 is not really intuitive. Is this a standard techinque or have been used in other literature? Please give more details and explanation.

3. Regarding proof of Lemma 7, what is the intuition behind leveraging on a series of functions that are linearly independent over a Hilbert space?


- Miscellaneous

Regarding the footnote in page 6, which comments on the extension to non i.i.d. case, I think there is typo. It should be Appendix G not F. Moreover, what is the result on the non i.i.d. case? Mentioning about only the Lemma seems to be incomplete. Please add a full theorem or proposition about the non-i.i.d. case if to argue as in the footnote.

In Algorithm 1, how do we initialize the buffer and does it work like First-in-First-out queue or do we have to collect $N$ new samples whenever we start the inner loop?

 In line 713,  $M=\log \gamma$ does not make sense since $M$ should be positive while $\log \gamma$ is negative.

**Limitations:**

Even though, there are strict assumptions such as prior knowledge on optimal policy to select the exploration strategy $\epsilon_t$, stationarity of Markov chain for every time step, and strict conditions on the initialization, I think the paper paved a way for analysis of DQN algorithm, and theoretical contribution is substantial. Hence, I am leaning towards acceptance.

---

> ### Author Rebuttal · Authors · 2023-08-04
>
> ***Q1**: Regarding $\epsilon_t$.*
>
> **A1**: Thanks for your insightful question! $e_t$ in Corollary 1 can be replaced by its upper bound, which is independent of $\epsilon_t$. See GR for details.
>
> Moreover, we would like to clarify the insights from our upper and lower bound for $\epsilon_t$. The upper and lower bound indicates: as $e_t$ diminishes, the corresponding $\epsilon_t$ also decreases. This implies a strategy shift wherein greater exploration (characterized by a larger $\epsilon_t$) is preferred in instances of limited environmental knowledge ($e_t$ being larger, indicating a behavioral policy somewhat distant from optimality). Conversely, as our understanding of the environment deepens, a preference for exploitation emerges. Furthermore, As we scale up the number of samples $N$, the lower bound of $\epsilon_t$ becomes smaller while the upper bound becomes larger. This enlarged gap between the lower and upper bound signifies a broader spectrum of potential selections for $\epsilon_t$, aligned with the augmented sample size. This phenomenon underscores the advantages associated with employing a larger sample size.
>
>
> ***Q2**: Regarding extension to non i.i.d. cases.*
>
> **A2**:  Thank you for bringing our typo to our attention. Following your suggestion, we introduce Theorem 2 (see the pdf uploaded in GR due to the character limit), which serves as a variation of Theorem 1 tailored for the non-i.i.d. scenario. The analysis of non i.i.d. has yielded the following key distinctions:
>
> (i) Compared with Theorem 1, $|\mathcal{A}|$ is replaced with $C_d$ in equations (17) and (20). This substitution is drawn from the difference between the stationary distribution and collected data distribution when the revised Assumption 2 holds. The difference has been elaborated in the error bound of $I_3$, as outlined in (148) within the supplementary materials.
>
> (ii) Furthermore, Theorem 2 introduces an additional error term into equations (19) and (20) because of the dependence between the collected data and current neural network weights.  This distinction has been elaborated through the error bound of $I_4$, which can be found in (160) within the supplementary materials.
>
> ***Q3**:(11) implies local convexity.*
>
> **A3**: To establish the existence of a local convex region, a crucial requirement is to demonstrate that the Hessian matrix within this region is non-negative. Specifically, the smallest eigenvalue in this region is non-negative, which can be achieved with both (i) the Hessian function must be continuous, and (ii) the Hessian matrix at the ground truth must be strictly positive. Once these two conditions are satisfied simultaneously, we can ensure the presence of a neighboring region around the ground truth where the Hessian matrix is always non-negative. This region corresponds to the desired local convex region near the ground truth.
>
> Equation (11) relates to the smallest eigenvalue of the Hessian matrix at the ground truth. By establishing a strictly positive lower bound for (11), that makes condition (ii) hold, we guarantee the existence of the desired local convex region.
>
>
> ***Q4**: Details of $I_4$.*
>
> **A4**: We apologize for the omitted details. The concept conveyed in lines 856 to 861 aims to streamline the integration process by transitioning from a high-dimensional space to a more manageable three-dimensional space. We have elaborated on the details (see the pdf uploaded in GR due to the character limit) for a comprehensive understanding.
>
> ***Q5**: “Linearly independent over a Hilbert space”.*
>
> **A5**: The main objective of Lemma 7 is to establish the existence of a local convex region around the ground truth. As replied in A3, we need the Hessian matrix at the ground truth to be strictly positive. With this condition and the continuity of the Hessian matrix, we can guarantee the presence of a neighboring region around the ground truth, where the Hessian matrix is always positive. This region corresponds to the desired local convex region near the ground truth.
>
> To ensure the smallest eigenvalue of the Hessian matrix is strictly greater than zero, we analyze the expression for $$I:=\min_{||\alpha||2=1} \mathbb{E}_h \Big\( \sum\_\{j=1\}^K \alpha\_j^\top h \phi ( w\_j^\top h) \Big)\^2.$$
>
> Notably, $I$ is always greater than or equal to zero. To rule out the possibility of $I$ being exactly zero, we need to prove the linear independence among $\\{h \phi^{\prime}(w_{j}^{\top}h)\\}\_{j=1}\^K$, meaning that there should be no $\alpha$ that can make $\sum_{j=1}^K \alpha_j^\top h \phi^{\prime}(w_{j}^{\top}h)$ equal to zero. By demonstrating the linear independence of $\\{h \phi^{\prime}(w_{j}^{\top}h)\\}\_{j=1}\^K$, we establish the assurance that $I$ cannot be zero, leading to a strictly positive smallest eigenvalue for the Hessian matrix. This result, in turn, ensures the existence of the desired local convex region around the ground truth.
>
> ***Q6**: Initialization of the buffer: First-in-First-out (FIFO) queue or collect $N$ new samples?*
>
> **A6**: In each outer loop, it is necessary to collect $N$ new samples according to the theorem. However, to optimize computational efficiency, we implement the FIFO queue to manage the data collection for the buffer, along with a random sampling strategy in the inner loop, during numerical experiments.
>
> ***Q7**: Typo of $M=\log\gamma$.*
>
> **A7**: Thanks for pointing out our typo. $M$ should be $\log \gamma^{-1}$. To further clarify, $M$ should be exactly $\log_\nu \gamma = \frac{\log \gamma}{\log \nu}$, where $\nu$ is a constant that is less than 1. As $\log \nu$ is a negative constant, we can safely ignore it when determining the complexity. Thus, we can simplify the expression for $M$ to $\Theta(\log \gamma^{-1})$. We appreciate your careful observation, and we hope this clarification resolves any confusion.

---

> > ### Comment · Reviewer_5oZJ · 2023-08-16
> >
> > Thank you for the detailed response. Most of my concerns have been addressed. Even though there are restricitve conditions, e.g., nice initialization, choice of $\epsilon_t$, given the current literature regarding analysis of DQN with time varying eps-greedy policy, I believe the paper has substantial contriubtion. Even though I have not checked the proof in details, the revised Theorem 1 in GR looks solid. Overall, I have raised my score from 6 to 7.

---

> > > ### Author Response · Authors · 2023-08-17
> > > **Thank you**
> > >
> > > It is encouraging for us to see that our responses have addressed most of your previous concerns. Especially, we are glad to hear that the revised Theorem 1 appears solid to you. The insightful input from you and other reviewers regarding the selection of $\epsilon_t$ motivates us towards refining a more practical version of Theorem 1. We greatly appreciate your recognition of our contribution and the increase in your score. If you have any additional suggestions or questions, please do not hesitate to share them with us.
> > >
> > > Best,
> > >
> > > Authors

---

### Official Review · Reviewer_KLjC · 2023-07-07

**Soundness:** 4 excellent
**Presentation:** 3 good
**Contribution:** 4 excellent
**Rating:** 7
**Confidence:** 3

**Summary:**

This paper provides a theoretical analysis of the converge of DQN in deep reinforcement learning. Its contributions are mostly theoretical and concern the important setting of epsilon-greedy exploration DQN with general function approximation. The paper includes convergence and sample complexity analysis for DQN as well as bounds for varying the exploration epsilon. Its results extend prior work with more restrictive assumptions (e.g. no epsilon greedy exploration in Xu and Gu, 2020).

**Strengths:**

The paper is well written and clear. The main strength of the paper is that as far as we know, this is the first theoretical work to tackle convergence of the Bellman error under both neural approximation and epsilon greedy exploration, without overly restrictive assumptions (two-layer networks etc). We appreciated for instance the fact that even as the epsilon exploration bounds are somewhat loose (as shown in empirical sections), they qualitatively explain crucial phenomena known to RL practitioners such as the necessity to anneal the exploration epsilon. The clarity of exposition also contrasts with the technicity of proofs (e.g. using tools from sub-Gaussian concentration of measure). Overall we find the theoretical contribution to be valuable and significant.

**Weaknesses:**

As far as weaknesses go, there are minor orthographic typos that remain in the main text, and should be addressed in second reading.

**Questions:**

Question to authors: It is not immediately obvious to me which strong intuitions we should have about the Holder exponent of quantity C_t that intervenes in Corollary 3, or if we could attempt to measure a pathwise estimate empirically. Could we detail this point further, perhaps in supplementary ?

**Limitations:**

---

> ### Author Rebuttal · Authors · 2023-08-09
>
> We thank the reviewer's comprehensive reading of this paper and valuable suggestions. The point-to-point responses will be posted based on the reviewer’s questions.
>
> ***Q1**: As far as weaknesses go, there are minor orthographic typos that remain in the main text, and should be addressed in the second reading.*
>
> **A1**: Thank you for your valuable feedback. We appreciate your keen observation regarding the minor orthographic typos in the main text. We have diligently addressed the issues in the revised paper at our best (with some examples listed below), and we will be committed to ensuring that the revised manuscripts meet the highest standards of quality.
>
> * “A small positive constant with linearly dependence on $\epsilon_t$” —>“A small positive constant with a linear dependence on $\epsilon_t$”.
>
> * “denote the the minimum distance of” —> “denote the minimum distance of”.
>
> * “Guassian” —> “Gaussian”
>
> * “bellman function” —>”Bellman function”
>
> * “such as e.g., Gaussian distribution” —>”e.g., Gaussian distribution”
>
> ***Q2**: Question to authors: It is not immediately obvious to me which strong intuitions we should have about the Holder exponent of quantity C_t that intervenes in Corollary 3, or if we could attempt to measure a pathwise estimate empirically. Could we detail this point further, perhaps in supplementary?*
>
> **A2**: Thank you for your thought-provoking inquiries! Corollary 3 can be interpreted as
>
> $$|C_t - C^\star | \le |W^{(t,0)} - W^\star |_2^\alpha,$$
>
> where $C^\star=0$. While directly calculating the Holder exponent isn't feasible due to the unknown value of $C^\star$, we can estimate this exponent by identifying an appropriate $\alpha$ that satisfies the following inequality:
>
> $$ |C_j-C_i| \le C \cdot \|W^{(j)} - W^{(i)} \|_2^\alpha.$$
>
> This estimation involves selecting some random pairs of points $W^{(i)}$ and $W^{(j)}$. Should we identify a constant $C$ and an $\alpha$ that ensures the equation holds true for any combination of $W^{(j)}$ and $W^{(i)}$, it would consequently establish the validity of the same $\alpha$ for Corollary 3. We are fully committed to providing a detailed implementation of this process in the supplementary materials, further solidifying our theoretical findings.

---

> > ### Comment · Reviewer_KLjC · 2023-08-17
> > **RE : comment**
> >
> > Thanks for the clarification ! Yes indeed this can be estimated pathwise with Monte-Carlo (although an estimator based say on wavelet coefficients might be more desirable in practice). My question was more related about the intuition one should have about that Holder exponent, since C_t is a fraction of suboptimal actions, - maybe in relation to the Holder exponent of the sample Bellman error itself. Just an idea to keep in mind for further work. Regardless, thanks for your time in engaging. I have no further questions and am holding my score.

---

> > > ### Author Response · Authors · 2023-08-19
> > > **Response to Reviewer KLjC**
> > >
> > > Thanks for your clarification. Intuitively, the Bellman error of the samples should also be a good indicator to detect the value change of $C_t$. We truly value your insights and will give it a try!

---

### Official Review · Reviewer_2co6 · 2023-07-08

**Soundness:** 3 good
**Presentation:** 2 fair
**Contribution:** 2 fair
**Rating:** 4
**Confidence:** 3

**Summary:**

This paper analyzes the sample complexity of using $\epsilon$-greedy for exploration in the setting of general function approximation.

**Strengths:**

This paper provides an convergence analysis of using $\epsilon$-greedy in Q-learning.

**Weaknesses:**

In my opinion, this paper is studying optimization rather than sample complexity. Equation (18) is saying that the initial Q value is accurate. However, in the research of sample complexity, we are concerning that how to explore the environment even when the initial policy is nothing and we know nothing about the environment. Here we have a good initial point, which weaken the need of exploration.

**Questions:**

1. The definition of Theta in Equation 12 is not found in the main text.

2. Why is the requirement of the initial point in Equation 18 become higher as \rho grows bigger? What is the intuition behind this phenomenon?

**Limitations:**

See 'Weakness' Session

---

> ### Author Rebuttal · Authors · 2023-08-09
>
> ***Q1**: In my opinion, this paper is studying optimization rather than sample complexity.*
>
> **A1**: Thank you for sharing your thoughts. Your input is greatly appreciated. We find that the concept of "sample complexity" lacks a precise definition within the RL literature. To make a fair comparison with established practices, e.g., [74], [23], we follow the existing work to define our "sample complexity" as the number of samples needed to attain a specific accuracy, which has been studied in our manuscript. We recognize the importance of clarity in our terminology, and we intend to enhance the clarity of the "sample complexity" definition in our revised paper.
>
> ***Q2**:  Regarding initialization assumption as shown in Equation (18).*
>
> **A2**: Thanks for your insightful and constructive question. To provide a clear understanding, we would like to clarify that equation (18) does not assert the accuracy of the initial Q value, and such an assumption regarding initialization aligns with the state-of-the-art practice in the theoretical analysis of DQNs.
>
> **Firstly, (18) is not equivalent to assuming the accuracy of the initial Q value.** Equation (18) presumes an initialization near the ground truth within the range of $1/K$. Meanwhile, the resulting neuron weights by Algorithm 1 have an estimation error in the order of $1/\sqrt{N_s}$, with the potential to become considerably smaller than $1/K$ provided that the value of $N_s$ surpasses $K^2$ as shown in (21). This indicates a substantial enhancement over the initial assumption.
>
> **Secondly, our initialization assumption in (18) is the state-of-the-art practice in DQNs.** Due to the significant convexity of the DQN learning problem, existing DQN results with non-linear activations require various initialization or objective function landscape assumptions to prove convergence, as seen in (i) two-layer assumptions for characterizable Eluder dimension in [18, 17, 29, 53], (ii) the achievability assumption of global optimal in [23, 28, 41, 49], and (iii) the assumption of a good initialization near the ground truth in [10, 19, 48, 74, 76]. These assumptions do not hinder the valuable insights these papers offer. As mentioned in [R1] and [R2], analyzing non-linear neural networks is complex due to the highly non-convex optimization problem with numerous spurious local minima. This complexity prevents tracking iterations without any assumptions on the initialization or landscape of the objective function. Consider [74] and [23] as examples. [74] confines initial points within a radius of $K^{-1/2}L^{-9/4}$ (based on numbers of hidden neurons $K$ and layers $L$) near a good point (nearly as good as the ground truth $W^*$) for $1/\sqrt{T}$ convergence. In our work, we assume initialization near the ground truth with radius $1/K$, similar to [74] when $K$ is at the $L^4$ order. Notably, [R3] uses a 5-layer network with 512 hidden neurons, which satisfies $K\sim L^4$. Moreover, [23] directly assumes global optimum is attainable. In contrast, we detail how mini-batch stochastic gradient descent secures convergence systematically by characterizing the initialization region, providing a deeper understanding.
>
> **Thirdly, we would like to emphasize that our theoretical analysis informs the practice of DQNs.** Our manuscript (Section 5 and Appendix F in the supplementary) presents enlightening empirical findings that are obtained based on real-data experiments using the DQN algorithm with random initialization. We showed that these empirical insights are aligned with the theoretical analyses built upon the initialization assumption. Moreover, it is noteworthy that within the NTK framework for supervised learning using neural networks [R4, R5], a good convergence point—almost as good as the ground truth—is consistently found near random initializations. We believe this observation underscores the practical validation of equation (18).
>
>
> [R1] Liang, Shiyu, et al. "Understanding the loss surface of neural networks for binary classification." ICML, 2018.
>
> [R2] Safran, Itay, et al. ''Spurious local minima are common in two-layer relu neural networks.'' ICML, 2018.
>
> [R3] Mnih, Volodymyr, et al. "Playing atari with deep reinforcement learning." arXiv preprint arXiv:1312.5602 (2013).
>
> [R4] Allen-Zhu, et al. "Learning and generalization in overparameterized neural networks, going beyond two layers." NeurIPS, 2019.
>
> [R5] Arora, Sanjeev, et al. "Fine-grained analysis of optimization and generalization for overparameterized two-layer neural networks." In ICML, 2019.
>
>
> ***Q3** : The definition of Theta in Equation 12 is not found in the main text.*
>
> **A3**: We apologize for the lack of clarity in the notation. To clarify the notation, we use $\Theta$ to denote the order of a function. Specifically, if $f(N) = \Theta(g(N))$, it means there exist constants $C_1$ and $C_2$ such that $C_1 g(N) \leq f(N) \leq C_2 g(N)$.
>
> Returning to Equation 12, we can interpret it as follows: there exist some constants $c_1, c_2, C_1, C_2$ such that $1-\frac{1-c_1\cdot \epsilon_t}{ \sqrt{c_2\cdot N}}$ $\le 1-\frac{1-\Theta(\epsilon_t)}{\Theta(\sqrt{N})}$ $\le 1-\frac{1-C_1\cdot \epsilon_t}{\sqrt{C_2\cdot N}}$.
>
> ***Q4**: Regarding $\rho$ in (18).*
>
> **A4**: We apologize for the confusion caused by the typo in (18). The correct expression should have $\rho$ in the numerator instead of the denominator, as $e_t:=||W^{(t,0)}-W^\star||\_F\le\mathcal{O}\Big(1-\frac{1-\Theta(\epsilon_t)}{\Theta(\sqrt{N})}\Big)\cdot \frac{\rho\cdot ||W^\star||\_F}{K}$. As a result, the requirement for the initial point becomes *more relaxed* as $\rho$ increases in magnitude.
>
> To clarify, $\rho$ corresponds with the smallest eigenvalue of the Hessian matrix of the objective function in (6), which also affect the radius of the local convex region of the objective function. When we have a larger $\rho$, we have a larger region of the local convex region, resulting in a more relaxed requirement for the initial point.

---

> > ### Author Response · Authors · 2023-08-20
> > **Update on A1 & A2**
> >
> > As the discussion stage is close to the end and we have not heard from the reviewer, we would like to provide an update in A1 & 2 with additional details to address the concern that “this paper is studying optimization rather than sample complexity. We are concerning that how to explore the environment even when the initial policy is nothing and we know nothing about the environment.”
> >
> > **Firstly,  we would like to clarify that our initialization requirement in (12) pertains to the optimization analysis of the objective function rather than the initial policy. We only impose a very minor assumption on the initial policy, and we make no assumption on the environment.** In this paper, $C_0$ captures the policy difference between the initial policy and the optimal policy, and the value of $C_0$ is independent of (12) in our main theoretical results (Theorem 1). With a sufficiently large replay buffer, as shown in (13), $C_0$ can be arbitrarily close to zero. We only need to exclude the case when $C_0$ equals $1$, which suggests an extreme case that the initial policy is different from the optimal policy in ALL states. Therefore,  we only impose a very minor assumption on the initial policy.  Moreover, we did not assume we have any knowledge about the environment.
> >
> > **Secondly, we would like to clarify (12) is the start-of-the-art assumption for optimization analyses involving deep neural networks.** As the objective function of deep neural networks is highly non-convex, containing an infinite number of local minima, analyzing deep neural networks in either reinforcement learning or supervised learning necessitates certain assumptions. Taking RL as an example, [23] directly assumes that the objective function can converge to the ground truth, which is a stronger assumption than ours. [74] utilizes a similar assumption to ours by assuming that the initialization is close to the ground truth. A detailed comment can be found in the second point of A2.
> >
> > **Finally, as we only impose minor assumptions on the initial policy, our theoretical findings in (20) imply the sample complexity, which is the number of samples required for achieving the desired accuracy.**  Such a definition of sample complexity follows existing literature that studies 'finite-sample/time analysis,' which includes a large body of studies, e.g., [23], [74], [75], [84]. As mentioned in [84], one of the objectives of finite-sample/time analysis is to determine the rate of convergence for SARSA (or Q-learning) and the dependency of solution accuracy on the number of samples, referred to as sample complexity. For instance, Theorems 1 and 3 [84] outline the results related to sample complexity by specifying the required number of samples/iterations to achieve an accuracy level of $\delta$. Furthermore, although [84] examines a simpler scenario involving linear function approximation, it introduces an assumption concerning the local convexity region as presented in Assumption 2. This assumption indicates that the initial point and following iterations should be within the local convex region of the ground true parameter $\theta^\star$, similar to equation (12) where the initialization is near the ground true parameter and within its local convex region. Consequently, we contend that introducing an assumption regarding the initialization does not conflict with the analysis of sample complexity. Our findings in equation (20) can therefore be interpreted as a representation of sample complexity. In addition, we want to highlight that, different from existing works,  our work provides a pioneering study on the sample complexity of DQN with an epsilon-greedy policy, as elaborated in Table 1.
> >
> >
> > [23] Fan, Jianqing, Zhaoran Wang, Yuchen Xie, and Zhuoran Yang. "A theoretical analysis of deep Q-learning." In Learning for dynamics and control, pp. 486-489. PMLR, 2020.
> >
> > [74] Xu, Pan, and Quanquan Gu. "A finite-time analysis of Q-learning with neural network function approximation." In International Conference on Machine Learning, pp. 10555-10565. PMLR, 2020.
> >
> > [84] Zou, Shaofeng, Tengyu Xu, and Yingbin Liang. "Finite-sample analysis for sarsa with linear function approximation." Advances in neural information processing systems 32 (2019).

---

### Author Rebuttal · Authors · 2023-08-09

We thank all the reviewers' comprehensive reading of this paper and valuable suggestions, and the paper will be revised according to the reviewers’ comments. We will first provide our general response (GR) to capture the major modifications made, focusing particularly on our revision of Theorem 1. Furthermore, we will address the reviewers' specific inquiries concerning the dependence between $\epsilon_t$ and $e_t$ in GR, drawing from the insights provided by our revised Theorem 1.


**(Revision of Theorem 1.)** A revision of our Theorem 1 is presented in the following text while retaining the core insights outlined in Section 4.1. Namely, Theorem 1 still implies that (T1) Theoretical characterizations of   $\\{\epsilon_t\\}_{t=1}^{T}$ for convergence, (T2) Convergence to the optimal Q-value function $Q^\star$ with bounded estimation error, and (T3) Sample complexity for achieving a desired estimation error of the optimal Q-value function.

Here are several significant changes in Theorem 1:

(i) The notation in equation (20) requires a correction to be $N\cdot T$ instead of just $N$, as there was an incorrect use of notation in equation (59) within the supplementary materials. The revised equation (59) should read as follows: $N\cdot T \ge \sum_{t=1}^T N_t \ge c_{N}^{-2}\rho^{-1} Ld \log q \cdot T$ instead of $N \ge \sum_{t=1}^T N_t$.

(ii) Theorem 1 holds with a high probability of at least $1 - T\cdot q^{-d}$, which we regret was missing in the submitted manuscript.

(iii) $C_t$ in both (17) and (20) are replaced with $C_\max$, which serves as an upper bound of $C_t$ for $t\le T$. Recall that $C_t$ reflects the proportion of non-optimal state-action pairs during the $t$-th iteration, it logically follows that $C_\max$ is upper bounded by at least $1$. This bound ensures that $C_\max$ is upper bounded in (20) for any arbitrarily large $T$. As the left-hand side of (20), $\frac{ |\mathcal{A}|\cdot R_{\max}}{(1-\gamma)^2 \cdot \Theta(\sqrt{N\cdot T})}$, goes to zero as $T$ goes to infinity, indicating the convergence of our proposed algorithm.

(iv) $e_t$ is replaced by a characteriable upper bound in (17), which makes the selection of $\epsilon_t$ independent of $e_t$.

---
**Theorem 1** (Convergence to $Q^\star$)
Suppose Assumptions 1 and 2 holds, the buffer size $N$ satisfies (13). Let us define $C_{\max}$ be a upper bound of  $C_t$ for $1\le t\le T$,  when $\epsilon_t$ satisfy

$$\epsilon_t = \max\Big\\{ \frac{c_{\varepsilon}\cdot  \Theta(\sqrt{N})\cdot \big(\gamma + {c_\epsilon}\cdot(1-\gamma)\big)^t e_0 }{(1-C_{\max}) \cdot |\mathcal{A}|\cdot R_{\max}} -\frac{C_{\max}}{1-C_{\max}}, ~~~~
\frac{C_\max}{1-C_\max}\Big\\} ~~~~~~(16)$$


for a fixed constant $c_\epsilon\in (0, (1-\gamma)^2]$, and the initialization satisfies


$$||W^{(0,0)}-W^*||\_F \le \mathcal{O}\Big( 1 - \frac{1-c_\epsilon}{\Theta(\sqrt{N})} \big)\cdot \frac{\rho\cdot ||W^\star||_F}{K}.   ~~~(18)$$


Then, with the high probability of at least $1-T\cdot q^{-d}$, we have

   * the learned weights decay geometrically with

 $$|| W^{(t,0)} -W^\star ||\_F \le \big(\gamma + c_\epsilon \cdot(1-\gamma)\big)^t \cdot ||W^{(0,0)}-W^\star||\_F, ~~~ t \le T_0:=\log_{\gamma} (1/N), ~~~~~~~~(19)$$


   * the returned model $Q(W^{(T,0)})$ exhibits an  estimation error as

$$\sup_{(s,a)}\big|Q(W^{(T,0)})-Q^\star\big| \le \frac{{C_{\max}}\cdot |\mathcal{A}|\cdot R_{\max}}{(1-\gamma)^2 \cdot \Theta(\sqrt{N\cdot T})}. ~~~~~~~~~~~(20)$$

---

**(Theoretical derivation of the revised Theorem 1.)** We would like to clarify that the updated results in (iii) and (iv) can be derived by taking

$$\begin{align}&\frac{c_N\cdot C_d\cdot \big({C_t +(1-C_t)\cdot \epsilon_t}\big)}{K}
\cdot \frac{R_\max}{1-\gamma} +(1-\epsilon_t/2)\gamma \cdot ||W^{(t,0)}-W^\star ||\_2\\\\
\le &\frac{c_N\cdot C_d\cdot \big(C_\max +(1-C_\max)\cdot \epsilon_t\big)}{K}\cdot \frac{R_\max}{1-\gamma} +(1-\epsilon_t/2)\gamma \cdot \big(\gamma+ {c_\epsilon}\cdot(1-\gamma)\big)^t e_0.\end{align}$$

in equation (62) in the supplementary materials.

**(Selection of $\epsilon_t$ and revision of Corollary 1.)** In the revised Theorem 1, when $t<T_0$, $c_\epsilon$ in $\epsilon_t$ remains constant, controlling the upper bound of $e_t$, thus offering a way to upper bound the magnitude of $e_t$. This observed behavior empowers us to utilize the expression $\big(\gamma+c_\varepsilon\cdot (1-\gamma)\big)^t\cdot e_0$ to replace $e_t$ in the selection of $\epsilon_t$. When $t>T_0$, $\epsilon_t$ satisfies equation. (20), the upper bound of $\epsilon_t$ becomes a small constant. Therefore, the choice of $\epsilon_t$ gradually decreases following a geometric decay until it reaches a low threshold, beyond which it remains constant at this threshold level. Our numerical experiments also adhere to this standard, and we intend to incorporate a similar clarification regarding $e_t$ in the updated version of the paper.

Following similar steps to (19), the lower bound of $\epsilon_t$ in (15) can be derived as

$$\epsilon_t \ge \max\Big\\{  1-\Theta(\sqrt{N})\cdot \Big(1-\big(\gamma + {c_\epsilon}\cdot(1-\gamma)\big)^t e_0\Big), ~~~
1- \Big(\Theta(\sqrt{N})-\frac{{C_\max}\cdot |\mathcal{A}|\cdot R_\max}{(1-\gamma)^2 \cdot \Theta(\sqrt{T})}\Big) \Big\\}~~~~~~~~~~~~~(15)$$

and the upper bound of $\epsilon_t$ in (16) can be derived as

$$\epsilon_t \le \max\Big\\{ \frac{(1-\gamma)^2\cdot  \Theta(\sqrt{N})\cdot \big(\gamma + {c_\epsilon}\cdot(1-\gamma)\big)^t e_0 }{(1-C_{\max}) \cdot |\mathcal{A}|\cdot R_{\max}} -\frac{C_{\max}}{1-C_{\max}}, ~~~\frac{C_{\max}}{1-C_{\max}} \Big\\}  (16)$$
which are independent of the iteration number $t$. Similar to Remark 2 in the original manuscript, we can derive the conclusion that $\epsilon_t$ needs to decrease as $t$ decreases. Namely, more exploitation (a smaller $\epsilon_t$) is preferred as we gain knowledge of the environment (larger iteration number $t$).

---

### Decision · Program_Chairs · 2023-09-21

**Decision:**

Accept (poster)

**Comment:**

A new analysis for the eps-greedy exploration in deep Q learning is presented. Despite a negative average of ratings at the very beginning, authors did a great job clarifying a few concerns shared among reviewers (who reached a positive consensus). After checking the original submission, I realized that most such concerns (most of which are due to an unclear way of presenting and a lack of comparison) are valid but can be fixed by a thorough revision. On the other hand, the authors’ rebuttal made them clear. Although I recommended acceptance, I urge authors to incorporate the rebuttal material into the original paper (maybe as a discussion section), which will make it much more solid.